

# Higher categorical symmetries and gauging in two-dimensional spin systems

Clement Delcamp[1][*] and Apoorv Tiwari[2][†]

**1** Department of Physics and Astronomy, Ghent University,
Krijgslaan 281, 9000 Gent, Belgium
**2** Department of Physics, KTH Royal Institute of Technology, Stockholm, 106 91 Sweden

[*] clement.delcamp@ugent.be , [†] apoorvt@kth.se

## Abstract

We present a framework to systematically investigate higher categorical symmetries in two-dimensional spin systems. Though exotic, such generalised symmetries have been shown to naturally arise as dual symmetries upon gauging invertible symmetries. Our framework relies on an approach to dualities whereby dual quantum lattice models only differ in a choice of module 2-category over some input fusion 2-category. Given an arbitrary two-dimensional spin system with an ordinary symmetry, we explain how to perform the (twisted) gauging of any of its sub-symmetries. We then demonstrate that the resulting model has a symmetry structure encoded into the Morita dual of the input fusion 2-category with respect to the corresponding module 2-category. We exemplify this approach by specialising to certain finite group generalisations of the transverse-field Ising model, for which we explicitly define lattice symmetry operators organised into fusion 2-categories of higher representations of higher groups.

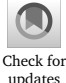

# 1 Introduction

*Global symmetries* have been playing a pivotal role in our understanding of quantum systems. Generally speaking, the existence of a global symmetry in a quantum system helps organise the spectrum of states and operators into representations of the symmetry. In addition, symmetry typically imposes strong constraints on the kinds of phases a quantum system can or cannot realise. These ideas have led for instance to Landau's classification scheme of phases of matter, and to organising principles for the particle content of the Standard Model. Despite its long and illustrious history, symmetry and its manifestations in quantum theory is very much an evolving story.

Conventionally, given a Hamiltonian model, symmetries are implemented by operators that act on all of space—i.e., *one-codimensional* operators in spacetime —commute with the Hamiltonian, and satisfy fusion rules representative of a *group*. In contrast, the modern perspective on global symmetries in quantum systems identifies the *topological invariance* of a symmetry operator within correlation functions as its defining property [72]. This perspective lends itself to numerous generalised notions of symmetry that have collectively come to be known as *global categorical symmetries* [63], in reference to the mathematical objects encoding them. Notably, these include symmetry structures whose topological operators may be supported on higher codimensional sub-manifolds and/or are not *invertible*, so that they do not obey fusion rules encoded into a group.

Relaxing the requirement that operators are one-codimensional has led to the concept of *higher-form* symmetry. Specifically, a $p$-form symmetry is defined with respect to topological operators with support on $(p+1)$-codimensional sub-manifolds and act by linking with $p$-dimensional operators [50, 74, 83]. These operators being invertible, fusion rules are still encoded into a group, but whenever $p > 0$, the corresponding group is necessarily *Abelian*.

Furthermore, it is possible to combine higher form symmetries of various degrees in a non-trivial way. The corresponding groups combine into categorifications of the notion of group known as *higher groups* [18], yielding the concept of higher group symmetries [10, 29, 49, 83]. Relaxing the requirement that operators are invertible has led to symmetries encoded into higher algebraic structures. For instance, given a (1+1)d system, (finite semisimple) non-invertible symmetries are encoded into *fusion 1-categories* [59], and the corresponding operators typically cannot be written as tensor products of local operators. More generally, given a $(d+1)$-dimensional system, it is possible to have non-invertible symmetry operators of varying degrees, in which case the corresponding algebraic structure is expected to be a fusion $d$-category [8, 23, 78]—a notion that remains partly elusive [47, 70].

As it turns out, though somewhat exotic, non-invertible symmetries are not rare in one-dimensional quantum models and have long been studied in the context of rational Conformal Field Theories (CFTs). There, topological operators go by the name of *Verlinde lines* [16, 104, 118] and exist in any rational CFT defined by a diagonal modular invariant. In particular, the fusion ring formed by the Verlinde lines corresponds to that of representations of the *chiral vertex algebra*, i.e., the underlying algebra of the given CFT, and is generically not group-like. A well-studied example is that of the diagonal *Ising* CFT, that hosts three Verlinde lines embodying the Ising fusion category. It includes in particular a non-invertible line known as the *Kramers-Wannier duality* defect [60, 61, 100, 101]. Guided in part by integrability, the sub-algebra of topological defects within rational CFTs was formalised for instance in ref. [20, 31, 64, 65]. Furthermore, it was already appreciated in this context that topological defects indeed embody a kind of symmetry structure within a quantum field theory (QFT), and thus it is sensible to consider notions of *'t Hooft anomalies* and *gauging* thereof [31, 62, 110].

Naturally, a prolific source of topological operators are *topological quantum field theories* (TQFTs) themselves. In fact, by definition, the entire spectrum of operators in a TQFT is topological. In particular, a large class of TQFTs host topological defects that obey non-invertible fusion rules. The most well-studied examples are provided by line defects in (2+1)d TQFTs, either in the continuum [107, 120] or in the discrete [26, 116]. Topological surface operators associated with *braided auto-equivalences* of the quantum invariant assigned by the theory to the circle have also been studied in this context [4, 22, 77, 113], but they are typically invertible. An important development was the remark that these surface operators in (2+1)d TQFTs could be constructed by condensing a suitable sub-algebras of topological line operators [37], suggesting a mechanism to generate a broader family of defects. This process was later formalised as the *condensation completion* of the category of line operators [47, 70]. The resulting (possibly) non-invertible *condensation defects* turn out to be rather ubiquitous in (2+1)d TQFTs [106].

In spite of these various developments, examples of non-invertible symmetry operators in higher dimensions have remained limited until recently. In the past year, various constructions of quantum systems with non-invertible symmetries have appeared that employ different kinds of generalised gauging procedures. Generally speaking, it is understood that given a theory with an invertible symmetry, gauging one of its sub-symmetries typically yields a theory with a different symmetry structure. Concretely, gauging a $p$-form symmetry in a $(d+1)$-dimensional theory yields a dual (gauged) model whose symmetry category contains $(d-p-1)$-dimensional topological operators labelled by irreducible representations of the corresponding group. Whenever the group is non-Abelian, these operators are in particular non-invertible [25, 46, 48, 52]. Moreover, gauging a (normal) sub-symmetry yields a theory possessing higher-group symmetries [110]. As it turns out, the symmetry structures resulting from these gauging procedures are even richer.

An early construction [82] of non-invertible defects in (3+1)d involved starting from a QFT with a 0-form and 1-form mixed anomaly and gauging the 1-form symmetry. It was shown that

this inevitably generates a non-invertible symmetry structure. Another class of examples were inspired by generalising the construction of the Kramers-Wannier duality defect in the (1+1)d Ising CFT to (3+1)d self-dual QFTs [27, 28, 94]. Yet another notable development pertained to the relation between gauging certain symmetry along sub-manifolds of spacetime and the condensation defects mentioned above [94, 106]. Most relevant to the present work were a series of papers [5, 6, 8, 9, 23, 24, 44] that considered starting from a (3+1)d theory with an invertible symmetry structure encoded into a higher-group and gauging one of its sub-symmetries. These works go beyond previous constructions in their analysis of the resulting symmetry structure in terms of so-called *higher representations* of higher groups [7, 56, 71]. Finally, these types of non-invertible symmetries have been further discussed in the context of various typical quantum field theories such as free field theories [99], pure gauge theories [2, 88], quantum electrodynamics [75], axion models [34] and within other physical contexts [30, 32, 33, 35, 69].

Notwithstanding the obvious recent interest in non-invertible symmetries, concrete lattice realisations of the corresponding topological operators have been largely unexplored, with some exceptions [80]. But, the lattice setting being concrete and tractable, it offers a welcome complimentary approach to understanding the most subtle aspects of these generalised categorical symmetries. Besides, it paves the way for exploring the implications of such symmetries on the phase diagram of familiar many-body systems. Furthermore, via the corresponding graphical calculus, the lattice setting is much closer related to the category theoretic framework underlying these symmetry structures.

Our paper aims at further bridging the gap between the *abstract* concept of a generalised categorical symmetry, as encoded into a higher mathematical structure, and its *concrete* realisation on a quantum theory. More specifically, we wish to address the question, what does it mean to have symmetry operators encoded into fusion 2-categories of higher categorical representations of higher groups? Guided by the *Morita theory* of fusion 2-categories [42, 44], we address this question by providing a framework that accomplishes—amongst other things— two tasks: Given an arbitrary (2+1)d spin system with an ordinary global symmetry, it allows for the systematic twisted gauging of one of its sub-symmetries, and the systematic identification of the resulting dual symmetry structure by constructing the corresponding topological lattice operators.

The framework we introduce in this manuscript is inspired by the study of dualities in one-dimensional quantum lattice models carried out in ref. [89, 90].[1] Within our framework, a duality class of models is specified by an algebra of operators that is generated by a set of (abstract) local operators. A representative of a duality class is obtained by choosing a Hilbert space and correspondingly explicit matrix representations for the local operators. Concretely, the algebras of operators we consider take as input data a finite group $G$—or rather, a fusion 2-category $2\mathrm{Vec}_G$ of $G$-graded 2-vector spaces—as well as a set of complex coefficients, which amounts to selecting certain linear combinations of local operators. These choices completely determine the physical properties of the duality class of models as encoded into their shared spectrum. Choosing a matrix representation then amounts to picking a so-called (indecomposable) *module 2-category* over $2\mathrm{Vec}_G$, i.e. a 2-category with a $G$-action. We think of the module 2-category as providing the physical degrees of freedom—which may satisfy *kinematical constraints*—on which the local operators act. It follows that Hamiltonian models that only differ in a choice of module 2-category are dual to one another.

In the framework described above, a duality operator amounts to a map between two module 2-categories, which provide matrix representations of the same local operators. For

---

[1]Throughout this manuscript, whenever we refer to two models as being dual to one another, we mean that there exists an operator performing the transmutation of the Hamiltonians into one another, which can be promoted to an isometry after addressing how the duality intertwines closed boundary conditions and charge sectors [90].

consistency, the action of this map is required to commute with the *G*-action resulting in the notion of *module 2-functor*. Similarly, a symmetry operator amounts to a module 2-endofunctor between a module 2-category and itself. More specifically, a module 2-endofunctor furnishes a topological surface operator that commutes with the Hamiltonian. There is also a notion of map between module 2-functors that are compatible with the *G*-action, namely *module natural 2-transformations*, which furnish topological lines at the interfaces of (possibly distinct) topological surfaces. These data can be organised into a 2-category. Crucially, given an indecomposable module 2-category $\mathcal{M}$, the composition of module 2-endofunctors endows this 2-category with a fusion structure. The resulting fusion 2-category $(2\mathsf{Vec}_G)^\star_\mathcal{M}$ is referred to the Morita dual of $2\mathsf{Vec}_G$ with respect to $\mathcal{M}$. This is the symmetry structure of the model obtained by choosing the Hilbert space associated with the module 2-category $\mathcal{M}$. Notice that we can make this statement without referring to a specific duality class of models. As emphasised in (1+1)d in ref. [89], this is because dualities are only sensitive to symmetry structures. Note that it is always possible to choose $2\mathsf{Vec}_G$ as a module 2-category over itself, in which case the symmetry fusion 2-category of the resulting model is again $2\mathsf{Vec}_G$. In other words, it is a model with an ordinary (0-form) *G*-symmetry. We can then show that choosing an alternative $2\mathsf{Vec}_G$-module 2-category has the interpretation of performing a twisted gauging of one of its sub-symmetries.

One merit of our approach is our ability to provide lattice operators accompanying these abstract statements, allowing to explicitly perform a twisted gauging in an arbitrary *G*-symmetric (2+1)d spin system and prove that the resulting model does have the expected symmetry structure by constructing the corresponding topological lattice operators. This ability extensively relies on the *tensor network* study of topological phases of matter where such symmetry operators first appeared in the form of matrix product operators in (1+1)d [40, 91] and projected entangled pair operators in (2+1)d [44]. In addition to providing a systematic recipe for generating new dual models, this framework explicitly provides lattice operators embodying symmetry structures related to higher representations of groups and categorifications thereof. Furthermore, we are also able to construct duality lattice operators performing the transmutation of the local symmetric operators.

We can offer a different perspective on our approach to dualities: It is understood that a three-dimensional TQFT as provided by the *Reshetikhin-Turaev* construction [108] possesses a *state-sum* description if and only if it admits a non-trivial gapped boundary [68]. These theories are of the *Turaev-Viro-Barrett-Westbury* type, whose input data are *spherical fusion 1-categories* [26, 116]. More specifically, given a choice of gapped boundary condition, which can be encoded into a module category over the input spherical fusion category, a state-sum can be obtained following the construction outlined for instance in ref. [17]. Distinct gapped boundary conditions yield distinct state-sums. The corresponding state spaces are then spanned by topological tensor network states that were defined in ref. [91]. In the same vein, we can construct a family of state-sums of the same four-dimensional topological *G* gauge theory indexed by module 2-categories over $2\mathsf{Vec}_G$ encoding various choices of gapped boundary conditions. The corresponding state spaces are then spanned by the topological tensor network states defined in ref. [44]. Importantly, it is possible to define distinct state sums of the same theory in different regions of spacetime. The operators intertwining these distinct lattice realisations then precisely correspond to the duality operators transmuting local symmetric operators of a given Hamiltonian into local symmetric operators of one of its duals, as considered in this manuscript.

We illustrate our approach with finite group generalisations of the transverse-field Ising model. For an arbitrary finite group, we consider the gauging of the whole invertible symmetry, revealing a dual symmetry structure in terms of *2-representations* of the group. Supposing that the input group is a semi-direct product, we also consider the gauging of its two constitutive

sub-symmetries in detail, revealing on the lattice dual symmetry structures in terms of 2-group-graded 2-vector spaces and 2-representations of 2-groups. Further specialising to the Klein four-group and the symmetric group of degree 3, we provide even more explicit expressions for the corresponding topological surfaces and topological lines in terms of spin operators, allowing us to confirm on the lattice their fusion and composition rules.

# Organisation of the paper

We begin in sec. 2 with an in-depth analysis of the symmetry structure resulting from gauging the global symmetry of the (2+1)d transverse-field Ising model. We emphasise in particular the appearance of non-invertible surface operators. Guided by this example, we present in sec. 3 a general framework to gauge invertible sub-symmetries of arbitrary (2+1)d quantum lattice models and construct the dual symmetry operators as encoded into the corresponding Morita dual fusion 2-category. A few specific scenarios are discussed in detail. Finally, we exemplify our approach in sec. 4 and 5 by specialising to finite group generalisations of the transverse-field Ising model for the Klein group and the symmetric group of degree 3, respectively.

# 2 Motivation: Transverse-field Ising model

*We set the stage by exploring the higher categorical symmetries that emerge from gauging the $\mathbb{Z}_2$ symmetry of the two-dimensional transverse-field Ising model.*

## 2.1 $\mathbb{Z}_2$-symmetric Hamiltonian

Let $\Sigma$ be a closed oriented two-dimensional surface endowed with a (fixed) triangulation $\Sigma_\triangle$ whose vertices, edges and plaquettes are denoted by $\mathsf{v}$, $\mathsf{e}$ and $\mathsf{p}$, respectively. Given an edge $\mathsf{e} \equiv (\mathsf{v}_1 \mathsf{v}_2)$ oriented from $\mathsf{v}_1$ to $\mathsf{v}_2$, we denote by $\mathsf{s}(\mathsf{e}) := \mathsf{v}_1$ and $\mathsf{t}(\mathsf{e}) := \mathsf{v}_2$ its source and target vertices, respectively. We consider a variant of the well-known *(2+1)d transverse-field Ising model*. As in the usual model, qubit degrees of freedom are assigned to vertices $\mathsf{v} \subset \Sigma_\triangle$. We identify such an assignment with a choice of 0-cochain $\mathfrak{m} \in C^0(\Sigma_\triangle, \mathbb{Z}_2)$ so the microscopic Hilbert space is provided by the tensor product $\bigotimes_\mathsf{v} \mathbb{C}[\mathbb{Z}_2] \simeq \bigotimes_\mathsf{v} \mathbb{C}^2$, where $\mathbb{Z}_2 = \langle r \,|\, r^2 = \mathbb{1}\rangle$. Moreover, we denote by $|\mathfrak{m}\rangle$ the state in the microscopic Hilbert space associated with 0-cochain $\mathfrak{m}$. Throughout this manuscript, we write basis elements of $\mathbb{C}[\mathbb{Z}_2]$ as $|0\rangle$ and $|1\rangle$, which are identified with the 'up' and 'down' state, respectively. Qubit degrees of freedom are governed by the Hamiltonian

$$\mathbb{H} = -J \sum_\mathsf{e} \sigma^z_{\mathsf{s}(\mathsf{e})} \sigma^z_{\mathsf{t}(\mathsf{e})} - J\kappa \sum_\mathsf{v} \sigma^x_\mathsf{v} - J\tilde{\kappa} \sum_\mathsf{v} \sigma^x_\mathsf{v} \prod_{(\mathsf{v}\mathsf{v}_1\mathsf{v}_2)} \exp\left(\frac{\mathrm{i}\pi}{4}(1 - \sigma^z_{\mathsf{v}_1}\sigma^z_{\mathsf{v}_2})\right), \tag{1}$$

where $\sigma^{x,z}_\mathsf{v}$ is the usual shorthand for $\mathrm{id} \otimes \cdots \otimes \mathrm{id} \otimes \sigma^{x,z}_\mathsf{v} \otimes \mathrm{id} \otimes \cdots \otimes \mathrm{id}$, with the tensor product being over all the vertices of the triangulation, and $\sigma^{x,z}_\mathsf{v}$ are Pauli operators.

The first term in the Hamiltonian describes a ferromagnetic interaction between qubits. The second term is the usual paramagnetic term. The third term is a topologically twisted variant of the paramagnetic term that includes phase factors associated with triangles $(\mathsf{v}\mathsf{v}_1\mathsf{v}_2)$ containing the vertex $\mathsf{v}$ [92]. One can readily check that the model has a global (0-form) $\mathbb{Z}_2$ symmetry implemented by *surface* operators[2] acting on all of $\Sigma_\triangle$

$$\mathcal{O}^0 = \prod_\mathsf{v} \mathrm{id}_\mathsf{v}, \quad \text{and} \quad \mathcal{O}^1 = \prod_\mathsf{v} \sigma^1_\mathsf{v}, \tag{2}$$

---

[2]Throughout this manuscript, we refer to operators that act on extended two-dimensional regions of $\Sigma$ as topological surface operators. These could act on all of $\Sigma$ or on a sub-region.

with $\mathbb{Z}_2$ fusion rules

$$\mathcal{O}^1 \odot \mathcal{O}^1 = \mathcal{O}^0 = \mathcal{O}^0 \odot \mathcal{O}^0, \quad \mathcal{O}^1 \odot \mathcal{O}^0 = \mathcal{O}^1 = \mathcal{O}^0 \odot \mathcal{O}^1. \tag{3}$$

Correspondingly, in the three extreme limits $1 \gg \kappa, \tilde{\kappa}$, $\kappa \gg 1, \tilde{\kappa}$ and $\tilde{\kappa} \gg 1, \kappa$, one obtains fixed-point Hamiltonians with ferromagnetic, paramagnetic and symmetry-protected topological (SPT) ground states, respectively. This Hamiltonian does not have any non-trivial 1-form symmetry as topological lines on either surface operator must be the identity line. Furthermore, it is not possible for the surface operator $\mathcal{O}^1$ to be open, i.e. to have support on a sub-region of $\Sigma_\triangle$. In other words, it is not possible to define a (topological) line interface between $\mathcal{O}^0$ and $\mathcal{O}^1$. We describe below how, upon *gauging* of the $\mathbb{Z}_2$ 0-form global symmetry, one inevitably lands on a dual model with more a intricate symmetry structure.

## 2.2 Gauging the $\mathbb{Z}_2$ symmetry

Although this was not immediately appreciated when the construction first appeared [81, 109], it is by now understood that gauging a 0-form $\mathbb{Z}_2$ symmetry yields a (2+1)d *dual* model hosting $\mathbb{Z}_2$ topological Wilson *lines* labelled by *representations* of the group. In modern terminology, this is the statement that the gauged model has a 1-form $\mathbb{Z}_2^\vee$ symmetry, with $\mathbb{Z}_2^\vee$ the *Pontrjagin dual* of $\mathbb{Z}_2$ [72]. However, it was recently pointed out that this is only part of the story [5, 24, 44]. Indeed, the 1-form $\mathbb{Z}_2^\vee$ symmetry is only a component of the symmetry structure of the gauged model in the sense that it does not encapsulate all possible topological operators.

In order to grasp the above statements, let us explicitly gauge the global $\mathbb{Z}_2$ symmetry in model (1). To do so, we begin by assigning additional qubit degrees of freedom to edges $e \subset \Sigma_\triangle$. We identify such an assignment with a choice of 1-cochain $\mathfrak{g} \in C^1(\Sigma_\triangle, \mathbb{Z}_2)$ so the model is now defined on the extended microscopic Hilbert space provided by the tensor product $\bigotimes_e \mathbb{C}[\mathbb{Z}_2] \bigotimes_v \mathbb{C}[\mathbb{Z}_2]$. Let us now promote generator $\mathcal{O}^1$ of the global $\mathbb{Z}_2$ symmetry to a local gauge transformation by defining *Gauß operators*

$$\mathbb{G}_v := \sigma_v^x \prod_{e \supset v} \sigma_e^x. \tag{4}$$

Since Gauß operators obey the multiplication rule in $\mathbb{Z}_2$ and $[\mathbb{G}_{v_1}, \mathbb{G}_{v_2}] = 0$, for any $v_1, v_2 \subset \Sigma_\triangle$, they are the generators of a $\mathbb{Z}_2$ gauge symmetry. Concretely, consider a basis state $|\mathfrak{g}, \mathfrak{m}\rangle$ in the extended microscopic Hilbert space. By definition, we have

$$\begin{aligned}
\sigma_v^z |\mathfrak{g}, \mathfrak{m}\rangle &= (-1)^{\mathfrak{m}[v]} |\mathfrak{g}, \mathfrak{m}\rangle, \\
\sigma_e^z |\mathfrak{g}, \mathfrak{m}\rangle &= (-1)^{\mathfrak{g}[e]} |\mathfrak{g}, \mathfrak{m}\rangle,
\end{aligned} \tag{5}$$

where $\mathfrak{m}[v]$ and $\mathfrak{g}[e]$ denote the restrictions of $\mathfrak{m}$ and $\mathfrak{g}$ to $v$ and $e$, respectively. One can now define a general Gauß operator indexed by a 0-cochain $\mathfrak{x} \in C^0(\Sigma_\triangle, \mathbb{Z}_2)$ which acts as

$$\mathbb{G}(\mathfrak{x}) := \prod_v \mathbb{G}_v^{\mathfrak{x}[v]} : |\mathfrak{g}, \mathfrak{m}\rangle \mapsto |\mathfrak{g} + d\mathfrak{x}, \mathfrak{m} + \mathfrak{x}\rangle. \tag{6}$$

The gauge symmetry is imposed *kinematically* so that we only consider *physical states* in the $+1$ eigenspace of $\mathbb{G}(\mathfrak{x})$ for any $\mathfrak{x} \in C^0(\Sigma_\triangle, \mathbb{Z}_2)$. We then require the gauged Hamiltonian to commute with Gauß operator $\mathbb{G}(\mathfrak{x})$. This can be accomplished by *minimally coupling* Hamiltonian (1) with the edge degrees of freedom:

$$\mathbb{H}_{g.} = -J \sum_e \sigma_{s(e)}^z \sigma_e^z \sigma_{t(e)}^z - J\kappa \sum_v \sigma_v^x - J\tilde{\kappa} \sum_v \sigma_v^x \prod_{(v v_1 v_2)} \exp\left(\frac{i\pi}{4}(1 - \sigma_{v_1}^z \sigma_{(v_1 v_2)}^z \sigma_{v_2}^z)\right) + \dots, \tag{7}$$

where '...' refers to other gauge invariant terms that can potentially be added in the process of gauging. A minimal example of such a term would be a product of $\sigma_e^z$ operators around closed loops in $\Sigma_\triangle$. For the sake of simplicity, we neglect these terms in what follows. We can readily confirm that $[\mathbb{H}_{\mathfrak{g}}, \mathbb{G}(\mathfrak{r})] = 0$ for any $\mathfrak{r} \in C^0(\Sigma_\triangle, Z_2)$.

At this point, it is crucial to notice that the microscopic Hilbert space splits into super-selection sectors labelled by eigenvalues of the operators $\prod_{e \subset (v_1 v_2 v_3)} \sigma_e^z$ associated with every triangle $(v_1 v_2 v_3) \subset \Sigma_\triangle$. As customary, we shall restrict to a single super-selection sector, namely that given by

$$\prod_{e \subset (v_1 v_2 v_3)} \sigma_e^z \overset{!}{=} \mathrm{id}, \quad \forall\, (v_1 v_2 v_3) \subset \Sigma_\triangle\,. \tag{8}$$

These conditions are also imposed kinematically enforcing $\mathfrak{g}$ to define a $\mathbb{Z}_2$ gauge field so that $d\mathfrak{g} = 0$. Finally, notice that we have

$$\left.\sigma_v^x\right|_{\mathbb{G}_v = \mathrm{id}} = \prod_{e \supset v} \left.\sigma_e^x\right|_{\mathbb{G}_v = \mathrm{id}}\,. \tag{9}$$

Upon enforcing this operator equality on the physical Hilbert space, the model becomes classical in the vertex degrees of freedom so they can be readily gauged away. Concretely, this operation is implemented by a unitary operator performing the basis rotation $|\mathfrak{g}, \mathfrak{m}\rangle \mapsto |\mathfrak{g} + d\mathfrak{m}, \mathfrak{m}\rangle$. Doing so delivers the dual Hamiltonian

$$\mathbb{H}^\vee = -J \sum_e \sigma_e^z - J\kappa \sum_v \prod_{e \supset v} \sigma_e^x - J\tilde{\kappa} \sum_v \prod_{e \supset v} \sigma_e^x \prod_{(v v_1 v_2)} \exp\left(\frac{i\pi}{4}\left(1 - \sigma_{(v_1 v_2)}^z\right)\right), \tag{10}$$

which acts on the physical Hilbert space $\mathcal{H}^\vee$ spanned by states $|\mathfrak{g}\rangle$, where $\mathfrak{g} \in Z^1(\Sigma_\triangle, \mathbb{Z}_2)$. Assuming for concreteness that $\Sigma_\triangle$ is the Poincaré dual of the honeycomb lattice, let us explicitly write the action of the various operators appearing in eq. (10). Firstly,

$$\sigma_e^z = \sum_{\mathfrak{g}} (-1)^{\mathfrak{g}[e]} |\mathfrak{g}\rangle\langle\mathfrak{g}|\,, \tag{11}$$

which measures the $\mathbb{Z}_2$ gauge field along the edge $e$. Secondly,

$$\prod_{e \supset v} \sigma_e^x = \sum_{\mathfrak{g}} |\mathfrak{g} + d\mathfrak{r}_v\rangle\langle\mathfrak{g}| \equiv \text{[hexagon diagram]}, \quad \text{with } \mathfrak{r}_v[v_1] = \begin{cases} 1, & \text{if } v = v_1\,, \\ 0, & \text{otherwise}, \end{cases} \tag{12}$$

which implements a $\mathbb{Z}_2$ gauge transformation at vertex $v$. Thirdly, denoting by $\bigotimes_v$ the hexagonal sub-complex centred around $v$ and $S := i^{\frac{1}{2}(1-\sigma^z)}$,

$$\prod_{e \supset v} \sigma_e^x \prod_{(v v_1 v_2)} \exp\left(\frac{i\pi}{4}\left(1 - \sigma_{(v_1 v_2)}^z\right)\right) \equiv \sum_{\mathfrak{g}} \exp\left(i\pi \mathrm{Bock}(\mathfrak{g})[\bigotimes_v]\right) |\mathfrak{g} + d\mathfrak{r}_v\rangle\langle\mathfrak{g}| \equiv \text{[hexagon diagram]}, \tag{13}$$

which implements a $\mathbb{Z}_2$ gauge transformation twisted by a sign depending on the number of 'up' states along $\partial \bigotimes_v$. In the above expression, Bock denotes the *Bockstein homomorphism*, a map of cohomology classes

$$\mathrm{Bock} : H^1(\Sigma_\triangle, \mathbb{Z}_2) \to H^2(\Sigma_\triangle, \mathbb{Z}_2)\,, \tag{14}$$

induced from the short exact sequence

$$1 \to \mathbb{Z}_2 \to \mathbb{Z}_4 \to \mathbb{Z}_2 \to 1. \tag{15}$$

Similar to the original model (1), the gauged model (7) also has three gapped phases. The case $1 \gg \kappa, \tilde{\kappa}$ corresponds to the confined phase, where the gauge fluctuations are energetically suppressed [66, 67]. Meanwhile, the $\kappa \gg 1, \tilde{\kappa}$ and $\tilde{\kappa} \gg 1, \kappa$ cases correspond to two topologically distinct deconfined phases. More precisely these are the two topological $\mathbb{Z}_2$ gauge phases whose renormalisation group fixed points are provided by the toric code and double semion model, respectively [51, 76, 95].

## 2.3 Symmetry operators

Let us now study the topological operators leaving the model $\mathbb{H}^\vee$ invariant. We distinguish two surface operators. These are the trivial operator or identity $\mathcal{U}^{\text{triv.}}$ and the non-trivial operator $\mathcal{U}^{\mathbb{Z}_2}$ defined as follows:[3]

$$\mathcal{U}^{\mathbb{Z}_2}[\Sigma_\triangle] := \sum_{\mathfrak{g} \in Z^1(\Sigma_\triangle, \mathbb{Z}_2)} \mathcal{Z}_{2d}(\mathfrak{g})[\Sigma_\triangle] |\mathfrak{g}\rangle\langle\mathfrak{g}| \equiv \frac{1}{2^{\#(\Sigma_\triangle)}} \sum_{\substack{\mathfrak{g} \in Z^1(\Sigma_\triangle, \mathbb{Z}_2) \\ \mathfrak{b} \in C^1(\Sigma_\triangle, \mathbb{Z}_2) \\ \mathfrak{n} \in C^0(\Sigma_\triangle, \mathbb{Z}_2)}} \exp\left( i\pi \int_{\Sigma_\triangle} \mathfrak{b} \smile (d\mathfrak{n} + \mathfrak{g}) \right) |\mathfrak{g}\rangle\langle\mathfrak{g}|,$$

where $\mathcal{Z}_{2d}(\mathfrak{g})[\Sigma_\triangle]$ is the partition function of a (1+1)d pure $\mathbb{Z}_2$ gauge theory coupled to background $\mathbb{Z}_2$ gauge field $\mathfrak{g}$ and $\#(\Sigma_\triangle) := |\Sigma_\triangle^0| + |\Sigma_\triangle^2|$ where $|\Sigma_\triangle^j|$ is the number of $j$-simplices in the triangulation $\Sigma_\triangle$ (see app. A for details). Henceforth, when there is no scope for confusion, we shall often omit specifying which sets the various cochains belong to for conciseness. The operator $\mathcal{U}^{\mathbb{Z}_2}[\Sigma_\triangle]$ commutes with the first term in the Hamiltonian as it acts diagonally in the $\sigma_e^z$ basis. It also commutes with the second and third terms in (10) by virtue of

$$\mathcal{Z}_{2d}(\mathfrak{g})[\Sigma_\triangle] = \mathcal{Z}_{2d}(\mathfrak{g} + d\mathfrak{r})[\Sigma_\triangle], \quad \text{and} \quad \text{Bock}(\mathfrak{g} + d\mathfrak{r}_v)[\otimes_v] = \text{Bock}(\mathfrak{g})[\otimes_v]. \tag{18}$$

Interestingly, this operator has *non-invertible* fusion rules [106]:

$$
\begin{aligned}
(\mathcal{U}^{\mathbb{Z}_2} \odot \mathcal{U}^{\mathbb{Z}_2})[\Sigma_\triangle] &= \frac{1}{2^{2\#(\Sigma_\triangle)}} \sum_{\substack{\mathfrak{g}, \mathfrak{b}, \mathfrak{n} \\ \mathfrak{g}', \mathfrak{b}', \mathfrak{n}'}} (-1)^{\int_{\Sigma_\triangle} \mathfrak{b} \smile (d\mathfrak{n} + \mathfrak{g}) + \mathfrak{b}' \smile (d\mathfrak{n}' + \mathfrak{g}')} |\mathfrak{g}\rangle\langle\mathfrak{g}|\mathfrak{g}'\rangle\langle\mathfrak{g}'| \\
&= \frac{1}{2^{2\#(\Sigma_\triangle)}} \sum_{\substack{\mathfrak{g}, \mathfrak{n} \\ \mathfrak{b}', \mathfrak{b}_+, \mathfrak{n}_+}} (-1)^{\int_{\Sigma_\triangle} \mathfrak{b}_+ \smile (d\mathfrak{n} + \mathfrak{g}) + d\mathfrak{n}_+ \smile \mathfrak{b}'} |\mathfrak{g}\rangle\langle\mathfrak{g}| \\
&= \left( \frac{1}{2^{\#(\Sigma_\triangle)}} \sum_{\mathfrak{b}', \mathfrak{n}_+} (-1)^{\int_{\Sigma_\triangle} \mathfrak{b}' \smile d\mathfrak{n}_+} \right) \frac{1}{2^{\#(\Sigma_\triangle)}} \sum_{\mathfrak{g}, \mathfrak{n}, \mathfrak{b}_+} (-1)^{\int_{\Sigma_\triangle} \mathfrak{b}_+ \smile (d\mathfrak{n} + \mathfrak{g})} |\mathfrak{g}\rangle\langle\mathfrak{g}| \\
&= \mathcal{Z}_{2d}[\Sigma_\triangle] \cdot \mathcal{U}^{\mathbb{Z}_2}[\Sigma_\triangle],
\end{aligned}
\tag{19}
$$

---

[3]Here d is the simplicial or lattice codifferential operator $d : C^n(\Sigma_\triangle, \mathbb{Z}_2) \to C^{n+1}(\Sigma_\triangle, \mathbb{Z}_2)$ such that for $\mathfrak{q} \in C^n(\Sigma_\triangle, \mathbb{Z}_2)$

$$d\mathfrak{q}[v_1 \dots v_{n+1}] = \sum_{j=1}^{n+1} (-1)^j \mathfrak{q}[v_1 \dots \hat{v}_j \dots v_{n+1}], \tag{16}$$

where $(v_1 \dots \hat{v}_j \dots v_{n+1})$ denotes the $n$-simplex with the vertex $v_j$ omitted. Further $\smile$ denotes the cup product $\smile : C^n(\Sigma_\triangle, \mathbb{Z}_2) \times C^m(\Sigma_\triangle, \mathbb{Z}_2) \to C^{n+m}(\Sigma_\triangle, \mathbb{Z}_2)$ such that

$$\mathfrak{q} \smile \mathfrak{p}[v_1 \cdots v_{n+m}] = \mathfrak{q}[v_1 \cdots v_n] \cdot \mathfrak{p}[v_n \cdots v_{n+m}], \tag{17}$$

where $\mathfrak{p} \in C^m(\Sigma_\triangle, \mathbb{Z}_2)$. Note that these notions can be readily generalised to other finite Abelian groups (see e.g. app. A of ref. [10]).

where the partition function $\mathcal{Z}_{\mathrm{2d}}[\Sigma_\triangle]$ of the pure $\mathbb{Z}_2$ gauge theory on $\Sigma_\triangle$ explicitly reads $\mathcal{Z}_{\mathrm{2d}}[\Sigma_\triangle] = 2^{b_1(\Sigma)-b_0(\Sigma)}$, where $b_j$ is the $j^{\mathrm{th}}$ *Betti number* of $\Sigma$.

Let us now attempt to rewrite the action of such a topological surface in terms of spin operators. To do so, it is instructive to first sum over $\mathfrak{b}$ in (18). Doing so delivers (see app. A)

$$\mathcal{U}^{\mathbb{Z}_2}[\Sigma_\triangle] = \frac{1}{2^{\chi(\Sigma)}} \sum_{\mathfrak{g},\mathfrak{n}} \delta_{\mathrm{d}\mathfrak{n},\mathfrak{g}} |\mathfrak{g}\rangle\langle\mathfrak{g}|, \tag{20}$$

where $\chi(\Sigma)$ is the Euler characteristic of $\Sigma$, $\mathrm{d}\mathfrak{n}[v_1 v_2] = \mathfrak{n}[v_1] + \mathfrak{n}[v_2]$, and $\delta_{\mathrm{d}\mathfrak{n},\mathfrak{g}} = \delta_{\mathrm{d}\mathfrak{n}+\mathfrak{g},0}$ is a $\mathbb{Z}_2$ Dirac delta function that imposes $\mathrm{d}\mathfrak{n} + \mathfrak{g} = 0 \bmod 2$. This operator can equivalently be expressed by first introducing 'virtual' qubit degrees of freedom at vertices so as to temporarily enlarge the physical Hilbert space from $\mathcal{H}^\vee$ to $\mathcal{H}^\vee \otimes \mathcal{H}^{\mathrm{virt.}}$ with $\mathcal{H}^{\mathrm{virt.}} = \bigotimes_v \mathbb{C}[\mathbb{Z}_2]$. Given $|\mathfrak{n}\rangle \in \mathcal{H}^{\mathrm{virt.}}$ with $\mathfrak{n} \in C^0(\Sigma_\triangle, \mathbb{Z}_2)$, we thus require an operator that projects onto the constraint subspace of states $|\mathfrak{g},\mathfrak{n}\rangle$ satisfying $\mathfrak{n}[v_1] = \mathfrak{g}[v_1 v_2] + \mathfrak{n}[v_2]$ at every edge $(v_1 v_2) \subset \Sigma_\triangle$, before performing a partial trace over $\mathcal{H}^{\mathrm{virt.}}$. In symbols,

$$\mathcal{U}^{\mathbb{Z}_2}[\Sigma_\triangle] = \frac{1}{2^{\chi(\Sigma)}} \mathrm{tr}_{\mathcal{H}^{\mathrm{virt.}}}\left( \prod_e \frac{1}{2}\left(\mathrm{id} + \sigma^z_{s(e)} \sigma^z_e \sigma^z_{t(e)}\right)\right). \tag{21}$$

Next, we ask, what are the line operators that commute with $\mathbb{H}^\vee$? In addition to the identity line, a line operator with support on any 1-cycle $\ell \in Z_1(\Sigma_\triangle, \mathbb{Z}_2)$ labelled by the non-trivial character in $\mathbb{Z}_2^\vee$ may be defined as

$$\prod_{e \subset \ell} \sigma^z_e = \sum_{\mathfrak{g}} \prod_{e \subset \ell} (-1)^{\mathfrak{g}[e]} |\mathfrak{g}\rangle\langle\mathfrak{g}|. \tag{22}$$

One can readily check that these line operators commute with $\mathbb{H}^\vee$. Moreover, they are topological by virtue of the kinematical constraints (8), so that the sign $\prod_{e \subset \ell} (-1)^{\mathfrak{g}[e]}$ only depends on the homology class of $\ell$ and is 1 whenever $\ell$ is a contractible cycle. More generally, any network of such lines can be assigned a cohomology class in $H^1(\Sigma_\triangle, \mathbb{Z}_2)$. Then the sign obtained by such a network of lines can be equivalently expressed via a representative cocycle $\mathfrak{f}$ in $H^1(\Sigma_\triangle, \mathbb{Z}_2)$ as

$$\mathcal{U}^{\mathrm{triv.}}(\mathfrak{f}) = \sum_{\mathfrak{g}} (-1)^{\int_{\Sigma_\triangle} \mathfrak{f} \smile \mathfrak{g}} |\mathfrak{g}\rangle\langle\mathfrak{g}|. \tag{23}$$

Consider for instance the following configuration:

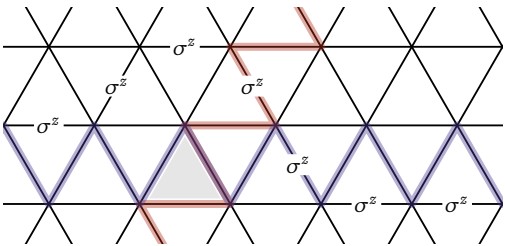

depicting a local patch of the triangular lattice $\Sigma_\triangle$ with a topological line operator (22) wrapping along one of the non-contractible cycles. The blue lines represent the only edges where the representative 1-cocycle $\mathfrak{f}$ evaluates to the non-trivial group element in $\mathbb{Z}_2$. Then for any choice of basis state $|\mathfrak{g}\rangle$ labelled by $\mathfrak{g} \in Z^1(\Sigma_\triangle, \mathbb{Z}_2)$,

$$\int_{\Sigma_\triangle} \mathfrak{f} \smile \mathfrak{g} = \prod_{e \subset \ell} \mathfrak{g}[e] \bmod 2. \tag{24}$$

For reference, the figure above also depicts a configuration $\mathfrak{g}$ which is non-trivial on the red lines and trivial elsewhere. The expressions on the left hand and right hand side of (24) evaluate to $-1$ for this choice of $\mathfrak{g}$ as the $\sigma^z$ operators only cross a single red line, and similarly a single plaquette (coloured in light grey) contributes to the cup product.

Summing over lines in (23) is equivalent to summing over $\mathfrak{f} \in H^1(\Sigma_\triangle, \mathbb{Z}_2)$:[4]

$$
\begin{aligned}
\frac{1}{|H^0(\Sigma_\triangle, \mathbb{Z}_2)|} \sum_{\mathfrak{f}} \mathcal{U}^{\text{triv.}}(\mathfrak{f}) &= \frac{1}{|H^0(\Sigma_\triangle, \mathbb{Z}_2)|} \sum_{\mathfrak{g}, \mathfrak{f}} (-1)^{\int_{\Sigma_\triangle} \mathfrak{f} \smile \mathfrak{g}} |\mathfrak{g}\rangle\langle\mathfrak{g}| \\
&= \frac{1}{2^{\#(\Sigma)}} \sum_{\mathfrak{g}, \mathfrak{b}, \mathfrak{n}} (-1)^{\int_{\Sigma_\triangle} \mathfrak{b} \smile (d\mathfrak{n} + \mathfrak{g})} |\mathfrak{g}\rangle\langle\mathfrak{g}| = \mathcal{U}^{\mathbb{Z}_2}[\Sigma_\triangle],
\end{aligned}
\tag{25}
$$

where, in going to second line, we have introduced a Lagrange multiplier field $\mathfrak{n}$, which when summed over imposes the cocycle condition on $\mathfrak{b} \in C^1(\Sigma_\triangle, \mathbb{Z}_2)$, recovering the first line. Interestingly, performing such a sum yields the surface operator $\mathcal{U}^{\mathbb{Z}_2}$ defined in eq. (18). As we shall comment later on, this is no mere coincidence. When gauging an Abelian 0-form symmetry, one obtains a dual model with topological line operators labelled by elements in the Pontrjagin dual, and more generally by representations of the group when it is non-Abelian. Additionally, one obtains topological surface operators, all of which can be understood by inserting networks (or condensing) suitable sub-algebras of topological lines. Such surface topological defects have been under scrutiny lately under the name of *condensation defects* [8, 28, 57, 70, 86, 106]. It follows immediately from the definition (23) that composition of such (networks of) lines within a surface operator $\mathcal{U}^{\text{triv.}}$ are given by

$$
\mathcal{U}^{\text{triv.}}(\mathfrak{f}_1 \circ \mathfrak{f}_2) = \mathcal{U}^{\text{triv.}}(\mathfrak{f}_1 + \mathfrak{f}_2).
\tag{26}
$$

Going back to definition (22), this is the statement that these line operators fuse like characters in $\mathbb{Z}_2^\vee$. Similarly, fusion rules of surface operators $\mathcal{U}^{\text{triv.}}$ with networks of lines inserted are given by

$$
\mathcal{U}^{\text{triv.}}(\mathfrak{f}_1) \odot \mathcal{U}^{\text{triv.}}(\mathfrak{f}_2) = \mathcal{U}^{\text{triv.}}(\mathfrak{f}_1 + \mathfrak{f}_2).
\tag{27}
$$

As suggested by our notation, we shall think of topological lines $\mathcal{U}^{\text{triv.}}(\mathfrak{f})$ as living on the trivial surface operator $\mathcal{U}^{\text{triv.}}$.

Next, we consider the operator $\mathcal{U}^{\mathbb{Z}_2}[\Sigma_\triangle]$ defined with a collection of lines inserted. Going back to definition (18) and given any 1-cycle $\ell \in Z_1(\Sigma_\triangle, \mathbb{Z}_2)$, such a line operator acts with the Pauli $\sigma^x$ operator on the virtual qubits, which are traced over, at the vertices $\mathsf{v} \subset \ell$. More generally, any network of such lines is found to be associated with a $\mathbb{Z}_2$-valued 1-cycle on the dual lattice $\Sigma_\triangle^\vee$, whose Poincaré dual is a 1-cocycle $\mathfrak{f} \in Z^1(\Sigma_\triangle, \mathbb{Z}_2)$ as before. The operator $\mathcal{U}^{\mathbb{Z}_2}(\mathfrak{f})[\Sigma_\triangle]$ with a network of such lines inserted has the form

$$
\begin{aligned}
\mathcal{U}^{\mathbb{Z}_2}(\mathfrak{f})[\Sigma_\triangle] &= \frac{1}{2^{\chi(\Sigma)}} \text{tr}_{\mathcal{H}^{\text{virt.}}} \left( \prod_e \frac{1}{2} (\text{id} + (-1)^{\mathfrak{f}[e]} \sigma^z_{s(e)} \sigma^z_e \sigma^z_{t(e)}) \right) \\
&= \frac{1}{2^{\#(\Sigma_\triangle)}} \sum_{\mathfrak{g}, \mathfrak{b}, \mathfrak{n}} (-1)^{\int_{\Sigma_\triangle} \mathfrak{b} \smile (d\mathfrak{n} + \mathfrak{f} + \mathfrak{g})} |\mathfrak{g}\rangle\langle\mathfrak{g}|.
\end{aligned}
\tag{28}
$$

---

[4] The choice of normalisation $|H^0(\Sigma_\triangle, \mathbb{Z}_2)|^{-1}$ is inherited from a convention in defining the partition function of $(d+1)$-dimensional finite group gauge theories, namely that the theory assigns a one-dimensional Hilbert space to a $d$-sphere for $d > 1$. Note that in eq. (25), we sum over cohomology classes, rather than cocycles, therefore the normalisation is $|H^0(\Sigma_\triangle, \mathbb{Z}_2)|$ instead of $|C^0(\Sigma_\triangle, \mathbb{Z}_2)|$.

Consider for instance the following configuration:

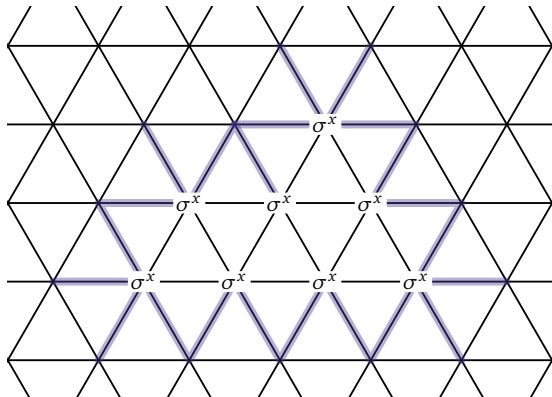

depicting such an operator, where as before blue lines represent the only edges where the corresponding 1-cocycle $\mathfrak{f}$ evaluates to the non-trivial element in $\mathbb{Z}_2$. It readily follows from the definition that composition rules of networks of lines within a surface operator $\mathcal{U}^{\mathbb{Z}_2}[\Sigma_\triangle]$ are given by

$$\mathcal{U}^{\mathbb{Z}_2}(\mathfrak{f}_1 \circ \mathfrak{f}_2)[\Sigma_\triangle] = \mathcal{U}^{\mathbb{Z}_2}(\mathfrak{f}_1 + \mathfrak{f}_2)[\Sigma_\triangle]. \tag{29}$$

Similarly, fusion rules of surface operators $\mathcal{U}^{\mathbb{Z}_2}$ with networks of lines inserted are given by

$$\big(\mathcal{U}^{\mathbb{Z}_2}(\mathfrak{f}_1) \odot \mathcal{U}^{\mathbb{Z}_2}(\mathfrak{f}_2)\big)[\Sigma_\triangle] = \mathcal{U}^{\mathbb{Z}_2}(\mathfrak{f}_1 + \mathfrak{f}_2)[\Sigma_\triangle]. \tag{30}$$

Finally, we would like to consider the possibility of defining a surface operator $\mathcal{U}^{\mathbb{Z}_2}$ with support on a sub-region of $\Sigma_\triangle$. This requires the existence of a topological line at the junction of topological surfaces $\mathcal{U}^{\mathbb{Z}_2}$ and $\mathcal{U}^{\text{triv.}}$. Such a line does exist and is simply obtained by restricting the definition (18) to an open sub-complex $\Xi_\triangle \subseteq \Sigma_\triangle$, i.e.

$$\mathcal{U}^{\mathbb{Z}_2}[\Xi_\triangle] = \sum_{\mathfrak{g} \in Z^1(\Sigma_\triangle, \mathbb{Z}_2)} \mathcal{Z}_{2\mathrm{d}}(\mathfrak{g})[\Xi_\triangle] |\mathfrak{g}\rangle\langle\mathfrak{g}|, \tag{31}$$

with Dirichlet boundary conditions, i.e., $\mathfrak{b}[\partial\Xi_\triangle] = 0$ imposed. We shall think of this operator as describing a line operator from a topological surface $\mathcal{U}^{\mathbb{Z}_2}$ to $\mathcal{U}^{\text{triv.}}$. Conversely, we shall think of the operator $\mathcal{U}^{\mathbb{Z}_2}[\Sigma_\triangle \backslash \Xi_\triangle]$ as describing a line operator from $\mathcal{U}^{\text{triv.}}$ to $\mathcal{U}^{\mathbb{Z}_2}$. Let us now consider the composition of the former line operator with the latter. Specifically, we consider a setup where $\Sigma$ is a two-torus or a cylinder endowed with a triangulation $\Sigma_\triangle$, and $\Xi_\triangle$ is an annular strip of width a single lattice spacing wrapping a non-contractible cycle:

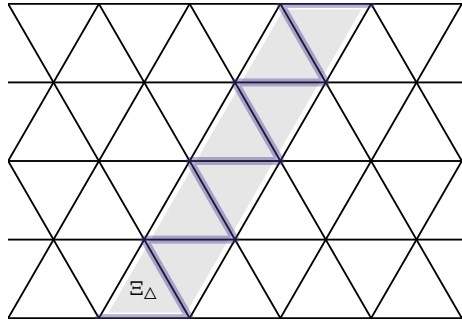

Then the composition of the lines is given by

$$\mathcal{U}^{\mathbb{Z}_2}[\Xi_\triangle] = \sum_{\mathfrak{g}} \mathcal{Z}_{2\mathrm{d}}[\Xi_\triangle] |\mathfrak{g}\rangle\langle\mathfrak{g}| = \sum_{\mathfrak{g}, \mathfrak{f}} (-1)^{\int_{\Xi_\triangle} \mathfrak{f} \smile \mathfrak{g}} |\mathfrak{g}\rangle\langle\mathfrak{g}| = \mathrm{id} + \prod_{\mathsf{e} \subset \ell} \sigma_\mathsf{e}^z, \tag{32}$$

where $\ell$ refers here to the non-contractible cycle wrapped by $\Xi_\triangle$. In the second equality, the sum is over $\mathfrak{f} \in H^1(\Xi_\triangle, \partial\Xi_\triangle, \mathbb{Z}_2) \cong \mathbb{Z}_2$, i.e., the *relative* cohomology group with Dirichlet boundary conditions imposed. Note also that the normalisation was implicitly modified from $1/|H^0(\Xi_\triangle, \mathbb{Z}_2)|$ to $1/|H^0(\Xi_\triangle, \partial\Xi_\triangle, \mathbb{Z}_2)| = 1$. The non-trivial class in $H^1(\Xi_\triangle, \partial\Xi_\triangle, \mathbb{Z}_2)$ corresponds to an $\mathfrak{f}$-defect wrapping the non-contractible cycle in $\Xi_\triangle$. For this choice of $\mathfrak{f}$, $\int_{\partial\Xi_\triangle} \mathfrak{f} \smile \mathfrak{g}$ evaluates to $\prod_{e \subset \ell} \mathfrak{g}[e]$ mod 2. As expected, this results in a line operator living on $\mathcal{U}^{\text{triv.}}$, which is labelled by the regular representation of $\mathbb{Z}_2$. The fusion rule of topological lines in (32) is closely related to the fusion rules of Kramers-Wannier duality defects in the (1+1)d transverse-field Ising model [27, 28].

Now let us compute the composition of the topological line between $\mathcal{U}^{\text{triv.}}$ and $\mathcal{U}^{\mathbb{Z}_2}$ by considering a thin annular strip of single lattice spacing width containing the identity operator $\mathcal{U}^{\text{triv.}}$, while the rest of the lattice $\Sigma_\triangle \backslash \Xi_\triangle$ containing $\mathcal{U}^{\mathbb{Z}_2}$. Let us specialise to the case where $\Sigma$ is a two-torus such that $\Sigma_\triangle \backslash \Xi_\triangle$ is path-connected. Let us denote the left and right boundaries of $\Xi_\triangle$ as $\partial_L \Xi_\triangle$ and $\partial_R \Xi_\triangle$, respectively. Then, the composition of lines is given by the operator

$$\mathcal{U}^{\mathbb{Z}_2}[\Sigma_\triangle \backslash \Xi_\triangle] = \frac{1}{2^{\#(\Sigma_\triangle \backslash \Xi_\triangle)}} \sum_{\mathfrak{g}, \mathfrak{b}, \mathfrak{n}} (-1)^{\int_{\Sigma_\triangle \backslash \Xi_\triangle} \mathfrak{b} \smile (d\mathfrak{n}+\mathfrak{g})} |\mathfrak{g}\rangle\langle\mathfrak{g}| = \frac{1}{2^{\chi(\Sigma_\triangle \backslash \Xi_\triangle)}} \sum_{\mathfrak{g}, \mathfrak{n}} \delta_{d\mathfrak{n}, \mathfrak{g}}^{\Sigma_\triangle \backslash \Xi_\triangle}, \qquad (33)$$

where in the first expression $\mathfrak{b} \in C^1(\Sigma_\triangle \backslash \Xi_\triangle, \mathbb{Z}_2)$ with the Dirichlet condition $\mathfrak{b}[\partial_L \Xi_\triangle] = \mathfrak{b}[\partial_R \Xi_\triangle] = 0$ imposed. Meanwhile $\mathfrak{n} \in C^0(\Sigma_\triangle, \mathbb{Z}_2)$[5] has no constraints imposed a priori. In the final expression, we sum over $\mathfrak{b}$, which imposes the cocycle condition $d\mathfrak{n} = \mathfrak{g}$ everywhere except within $\Xi_\triangle$, denoted by $\delta_{d\mathfrak{n}, \mathfrak{g}}^{\Sigma_\triangle \backslash \Xi_\triangle}$. Besides, note that the Euler characteristic $\chi(\Sigma_\triangle \backslash \Xi_\triangle) = 0$, since $\Sigma \backslash \Xi$ is a cylinder. Now pick a preferred edge in $\Xi_\triangle$. Naturally $d\mathfrak{n} = \mathfrak{g} + \mathfrak{s}$, where $\mathfrak{s}$ is valued in $\mathbb{Z}_2$, on this edge. It follows from conditions $d\mathfrak{g} = 0$ and $d\mathfrak{n} = \mathfrak{g}$ on $\partial\Xi_\triangle$ that fixing $\mathfrak{s}$ on any chosen edge in $\Xi_\triangle$ pins the configuration to the same value of $\mathfrak{s}$ for all other edges in $\Xi_\triangle$. Consider for instance the following configuration:

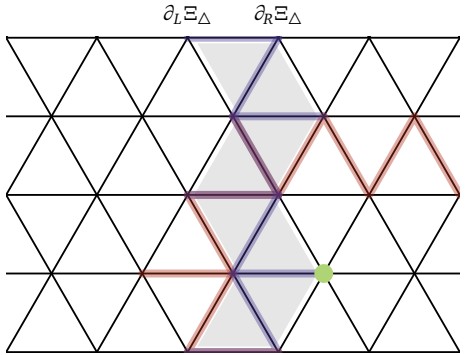

As before, the 1-cocycle $\mathfrak{g}$ is non-trivial only on the red edges. The condition $d\mathfrak{n} = \mathfrak{g}$ is satisfied everywhere in $\Sigma_\triangle \backslash \Xi_\triangle$, i.e., everywhere apart from the central region in grey. Then we consider $\mathfrak{n}$ to be fixed to a certain configuration on $\partial_L \Xi_\triangle$. Fixing the configuration of $\mathfrak{n}$ on a single vertex (for instance the one highlighted in green) on $\partial_R \Xi_\triangle$, pins the configuration on all other vertices on $\partial_R \Xi_\triangle$. The two choices at this vertex correspond to either the presence or absence of a line in $\text{Vec}_{\mathbb{Z}_2}$ traversing $\Xi_\triangle$. Therefore, defining a $\mathbb{Z}_2$ cocycle $\mathfrak{f}$ that evaluates to the non-trivial element in $\mathbb{Z}_2$ on every edge in $\Xi_\triangle$ (indicated in blue in the above diagram) and to the identity element elsewhere, we obtain

$$\mathcal{U}^{\mathbb{Z}_2}[\Sigma_\triangle \backslash \Xi_\triangle] = \mathcal{U}^{\mathbb{Z}_2}[\Sigma_\triangle] + \mathcal{U}^{\mathbb{Z}_2}(\mathfrak{f})[\Sigma_\triangle], \qquad (34)$$

which amounts to a line operator living on $\mathcal{U}^{\mathbb{Z}_2}[\Sigma_\triangle]$. This concludes our analysis of the symmetry structure of the gauged transverse-field Ising model.

---

[5]$\mathfrak{n}$ is a 0-cochain on $\Sigma_\triangle$ since $C_0(\Sigma_\triangle, \mathbb{Z}) = C_0(\Sigma_\triangle \backslash \Xi_\triangle, \mathbb{Z})$.

We showed in this section that starting from a (2+1)d lattice model with arguably the simplest kind of symmetry, namely a 0-form $\mathbb{Z}_2$ symmetry, gauging the symmetry results in a model with non-invertible surface operators. It turns out that the surface and line operators, together with their statistics, are organised into an algebraic structure referred to as the fusion 2-category of 2-representation of $\mathbb{Z}_2$. In the next section, we present a framework allowing for the systematic gauging of arbitrary invertible symmetries and analysis of the resulting symmetry structures in terms of higher representations of groups, and categorifications thereof.

# 3 Gauging and dual symmetries

*Motivated by the analysis of the (2+1)d transverse-field Ising model carried out above, we introduce in this section a systematic approach to gauging invertible symmetries in (2+1)d quantum lattice models and studying the resulting higher categorical symmetries.*

## 3.1 $G$-symmetric Hamiltonians

Throughout this manuscript, our starting point is always a two-dimensional quantum lattice model with a global 0-form $G$ symmetry, where $G$ is a finite (possibly non-Abelian) group. Concretely, it means that the Hamiltonian commutes with topological operators supported on the whole two-dimensional space, which are labelled by group elements of $G$, in such a way that the fusion of symmetry operators is governed by the multiplication rule of the group. By definition of a group, these symmetry operators are in particular *invertible*.

The modern approach to global symmetries in quantum field theories in terms of collections of *topological defects* invites us to organise symmetry operators and their properties into higher categories. More specifically, given a (2+1)d quantum theory we expect (finite semisimple) symmetries to correspond to *fusion 2-categories* in the sense of Douglas and Reutter [47], where objects label topological surface operators and (1-)morphisms label topological line operators at the junctions of surface operators. In this context, a $G$-symmetric Hamiltonian commutes with surface operators that form the so-called fusion 2-category of *$G$-graded 2-vector spaces*. Let us present this fusion 2-category in some detail.[6] First of all, let us define a *2-vector space* as a $\mathbb{C}$-linear, finite, semisimple category. We can then consider the 2-category 2Vec of 2-vector spaces, linear functors and natural transformations. It is a prototypical example of fusion 2-category, where the monoidal structure is given by the *Deligne* tensor product. Note that 2Vec has a unique equivalence class of simple objects, which is represented by the category Vec of complex vector spaces. Let us now consider the *2-groupoid*[7] $[G, \bullet, \bullet]$ with object-set $G$, no non-trivial 1-morphisms and no non-trivial 2-morphisms. Consider the category 2Fun($[G, \bullet, \bullet]$, 2Vec) of *pseudofunctors*, *pseudonatural transformations* and *modifications* between $[G, \bullet, \bullet]$ and 2Vec. By definition, an object V in 2Fun($[G, \bullet, \bullet]$, 2Vec) assigns to every $g \in G$ a 2-vector space $\mathsf{V}_g$ in 2Vec, and thus amounts to a $G$-graded 2-vector space of the form $\mathsf{V} = \boxplus_{g \in G} \mathsf{V}_g$. Pseudonatural transformations in 2Fun($[G, \bullet, \bullet]$, 2Vec) then correspond to grading preserving linear functors, and modifications to natural transformations. The convolution product of pseudofunctors $[G, \bullet, \bullet] \to$ 2Vec endows 2Fun($[G, \bullet, \bullet]$, 2Vec) with the structure of a fusion 2-category according to

$$(\mathsf{V} \odot \mathsf{W})_g := \boxplus_{x \in G} \mathsf{V}_x \boxtimes V_{x^{-1}g}, \tag{35}$$

---

[6]In the vein of sec. 3.6, we shall think of $2\mathsf{Vec}_G$ as a categorification of the fusion (1-)category $\mathsf{Vec}_G$ of $G$-graded vector spaces, the same way we can think of $\mathsf{Vec}_G$ as a categorification of $\mathbb{C}[G]$, whereby the ring $\mathbb{C}$ is promoted to the fusion category $\mathsf{Vec}$.

[7]A 2-groupoid is a 2-category in which every morphism is an equivalence, in the same spirit of a 1-groupoid being a (small) category in which every morhism is invertible.

with unit $\mathbb{1}$ satisfying $\mathbb{1}_g = \delta_{g,\mathbb{1}_G}\mathsf{Vec}$. Henceforth, we denote by $2\mathsf{Vec}_G$ this fusion 2-category. There are $|G|$-many simple objects in $2\mathsf{Vec}_G$ provided by the 'one-dimensional' 2-vector spaces $\mathsf{Vec}_g$, for every $g \in G$, such that $\mathsf{Vec}_{g_1} \odot \mathsf{Vec}_{g_2} \cong \mathsf{Vec}_{g_1 g_2}$ and $\mathrm{Hom}_{2\mathsf{Vec}_G}(\mathsf{Vec}_{g_1}, \mathsf{Vec}_{g_2}) \cong \delta_{g_1, g_2}\mathsf{Vec}$. At the end of the day, it follows that simple objects can be safely identified with the corresponding group elements in $G$, but the higher categorical perspective will be crucial in the following.

Let us now construct local operators that explicitly commute with symmetry operators labelled by simple objects in $2\mathsf{Vec}_G$ in the spirit of ref. [89] using the tools introduced in ref. [44]. Let $\Sigma$ be a closed oriented two-dimensional surface endowed with a triangulation $\Sigma_\triangle$. Although our construction applies to arbitrary triangulations $\Sigma_\triangle$, let us assume for concreteness that $\Sigma_\triangle$ is isotopic to the Poincaré dual of a honeycomb lattice. We further assume that $\Sigma_\triangle$ has a total ordering of its 0-simplices (vertices), referred to as a choice of *branching structure*, such that the branching structure in the neighbourhood of every vertex $\mathsf{v} \equiv (3) \subset \Sigma_\triangle$ is of the form

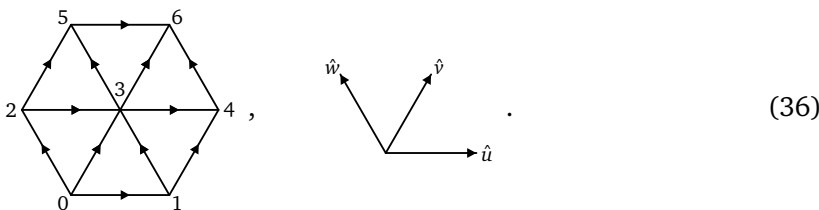

$$ (36) $$

Notice that a choice of branching structure induces an orientation of each 1-simplex (edge), which is always chosen to be from the lowest ordered vertex to the higher ordered one. Let $\mathfrak{m}$ denote an assignment of group elements in $G$ to vertices of $\Sigma_\triangle$. By a slight abuse of notation, we notate via $C^0(\Sigma_\triangle, G)$ the collection of such assignments, which corresponds to a $G$-valued 0-cochain when $G$ is Abelian. We define the microscopic Hilbert space of the system to be $\bigotimes_\mathsf{v} \mathbb{C}[G]$ and denote by $|\mathfrak{m}\rangle$ the assignment $\mathfrak{m}$ regarded as an element of the microscopic Hilbert space. The restriction of $|\mathfrak{m}\rangle$ to a given vertex $\mathsf{v} \subset \Sigma_\triangle$ is denoted by $|\mathfrak{m}[\mathsf{v}]\rangle \in \mathbb{C}[G]$, and more generally, we notate via $|\mathfrak{m}[\Xi_\triangle]\rangle := \bigotimes_{\mathsf{v} \subset \Xi_\triangle} |\mathfrak{m}[\mathsf{v}]\rangle$ the state associated with the restriction of $\mathfrak{m}$ to a sub-complex $\Xi_\triangle \subseteq \Sigma_\triangle$.

We are interested in $G$-symmetric local operators acting on the Hilbert space $\bigotimes_\mathsf{v} \mathbb{C}[G]$. Given a vertex $\mathsf{v} \subset \Sigma_\triangle$, we notate via $\varhexagon_\mathsf{v} \subseteq \Sigma_\triangle$ the hexagonal sub-complex centred around $\mathsf{v}$. Let us now consider the pinched interval cobordism $\varhexagon_\mathsf{v} \times_\mathrm{p.} \mathbb{I} \equiv \varhexagon_\mathsf{v} \times_\mathrm{p.} [0,1]$ defined as $\varhexagon_\mathsf{v} \times \mathbb{I} / \sim$, where the equivalence relation $\sim$ is such that $(x, i) \sim (x, i')$ for all $(x, i), (x, i') \in \partial \varhexagon_\mathsf{v} \times \mathbb{I}$. Graphically,

$$\varhexagon_\mathsf{v}^\mathbb{I} := \varhexagon_\mathsf{v} \times_\mathrm{p.} \mathbb{I} \equiv \qquad\qquad , \qquad\qquad (37)$$

where the branching structure induced from that of eq. (36) is such that $2 < 3' < 3$. Notice that, by definition, we have $\partial \varhexagon_\mathsf{v}^\mathbb{I} = \overline{\varhexagon_\mathsf{v} \times \{0\}} \cup_{\partial \varhexagon_\mathsf{v}} \varhexagon_\mathsf{v} \times \{1\}$. Henceforth, we employ the shorthand notations $\varhexagon_\mathsf{v}^0 \equiv \varhexagon_\mathsf{v} \times \{0\}$ and $\varhexagon_\mathsf{v}^1 \equiv \varhexagon_\mathsf{v} \times \{1\}$. Given an assignment $\mathfrak{m} \in C^0(\varhexagon_\mathsf{v}^\mathbb{I}, G)$, we can define an operator acting on a local neighbourhood of the vertex $\mathsf{v}$ as $|\mathfrak{m}[\varhexagon_\mathsf{v}^1]\rangle\langle\mathfrak{m}[\varhexagon_\mathsf{v}^0]|$. Notice that this operator acts as the identity operator at every vertex but $\mathsf{v}$ where it acts as $|\mathfrak{m}[\mathsf{v}']\rangle\langle\mathfrak{m}[\mathsf{v}]|$.

More general operators can then be constructed by considering linear combinations of the form

$$\mathbb{h}_{v,n} := \sum_{\mathfrak{m} \in C^0(\otimes_v^{\mathbb{I}}, G)} h_{v,n}\big(\{\mathfrak{m}[v_1]\mathfrak{m}[v_2]^{-1}\}_{(v_1 v_2) \subset \otimes_v^{\mathbb{I}}}\big) \big|\mathfrak{m}[\otimes_v^1]\big\rangle\big\langle\mathfrak{m}[\otimes_v^0]\big|, \tag{38}$$

where the coefficients $h_{v,n}$ are valued in $\mathbb{C}$ and the oriented edges $(v_1 v_2)$ are always such that $v_1 < v_2$. Note that, in practice, we typically consider models for which the complex coefficients $h_{v,n}$ are only non-vanishing for specific choices of assignments $\mathfrak{m} \in C^0(\otimes_v^{\mathbb{I}}, G)$. Any combinations of such operators can finally be used to define a local Hamiltonian $\mathbb{H} \equiv \sum_v \mathbb{h}_v := \sum_v \sum_n \mathbb{h}_{v,n}$.

By construction, any Hamiltonian thus defined is $G$-symmetric, whereby the symmetry is generated by operators $\prod_{v \subset \Sigma_\triangle} R_v^x$ with $R_v^x : |\mathfrak{m}[v]\rangle \mapsto |\mathfrak{m}[v]x^{-1}\rangle$ for any $x \in G$. This simply follows from a redefinition of the variable $\mathfrak{m} \in C^0(\otimes_v^{\mathbb{I}}, G)$ in the summation, together with the fact that the coefficients $h_{v,n}$ only depend on $\{\mathfrak{m}[v_1]\mathfrak{m}[v_2]^{-1}\}_{(v_1 v_2) \subset \otimes_v^{\mathbb{I}}}$ and are therefore manifestly symmetric. Identifying every $\mathfrak{m}[v] \in G$ with the corresponding simple object $\mathsf{Vec}_{\mathfrak{m}[v]}$ in $2\mathsf{Vec}_G$, one can equivalently state that any Hamiltonian thus defined is $2\mathsf{Vec}_G$-symmetric. It is a straightforward exercise—which we carry out below—to show that the transverse-field Ising model is of this form, and more generally, we can argue that every local $G$-symmetric Hamiltonian can be written in terms of combinations of local operators of the form (38). Note that Hamiltonians defined in this section only account for nearest or next-nearest neighbours interactions. However, we can readily combine such local operators—which geometrically amounts to concatenating complexes of the form (37)—so as to define local operators simultaneously acting on a larger number of sites, thereby generating the whole algebra of $G$-symmetric Hamiltonians on $\bigotimes_v \mathbb{C}[G]$. That being said, a lot of familiar and physically relevant quantum systems, e.g. Ising-like models, are already included within the present formalism.

In prevision for the following section, let us slightly reformulate the previous construction. We noticed above that the $G$ symmetry of local operators (38) is guaranteed in particular by the fact that the unitary coefficients $h_{v,n}$ only depend on the assignment $\mathfrak{m}$ through group elements $\mathfrak{m}[v_1]\mathfrak{m}[v_2]^{-1}$ for every edge $(v_1 v_2) \subset \otimes_v^{\mathbb{I}}$. This leads us to contemplate the following alternative description: Consider an assignment $\mathfrak{g}$ of group elements in $G$ to every edge of $\otimes_v^{\mathbb{I}}$ such that $\mathfrak{g}[v_1 v_2]\mathfrak{g}[v_2 v_3] = \mathfrak{g}[v_1 v_3]$ for every 2-simplex $(v_1 v_2 v_3) \subset \otimes_v^{\mathbb{I}}$, which we shall think about as a flat gauge field on $\otimes_v^{\mathbb{I}}$. By slight abuse of notation, we notate via $Z^1(\otimes_v^{\mathbb{I}}, G)$ the collection of such assignments. Let $\mathfrak{m} \in C^0(\otimes_v^{\mathbb{I}}, G)$ be an assignment as before but with the additional constraint that $\mathfrak{m}[v_1] = \mathfrak{g}[v_1 v_2]\mathfrak{m}[v_2]$ for every edge $(v_1 v_2) \subset \otimes_v^{\mathbb{I}}$. In other words, we require $d\mathfrak{m} = \mathfrak{g}$, where $d\mathfrak{m}[v_1 v_2] = \mathfrak{m}[v_1]\mathfrak{m}[v_2]^{-1}$. We can now rewrite the previous operators as follows:

$$\mathbb{h}_{v,n} = \sum_{\mathfrak{g} \in Z^1(\otimes_v^{\mathbb{I}}, G)} h_{v,n}(\mathfrak{g}) \sum_{\substack{\mathfrak{m} \in C^0(\otimes_v^{\mathbb{I}}, G) \\ d\mathfrak{m} = \mathfrak{g}}} \big|\mathfrak{m}[\otimes_v^1]\big\rangle\big\langle\mathfrak{m}[\otimes_v^0]\big|. \tag{39}$$

Notice that given $\mathfrak{g}$, the condition $d\mathfrak{m} = \mathfrak{g}$ does not fully constrain $\mathfrak{m}$. In the following section, we consider generalizations of these operators yielding dual Hamiltonian models.

**Back to the transverse-field Ising model:** Let us illustrate our construction by recasting the transverse-field Ising model in terms of local operators (39). We also discuss a finite group generalisation of this model in sec. 3.8. Consider the Hamiltonian $\mathbb{H} = \sum_{v \subset \Sigma_\triangle} \sum_{n=1}^{4} \mathbb{h}_{v,n}$. For any vertex $v \subset \Sigma_\triangle$ and gauge field $\mathfrak{g} \in Z^1(\otimes_v^{\mathbb{I}}, G)$, the defining complex coefficients $h_{v,n}(\mathfrak{g})$ are chosen to be

$$h_{v,1}(\mathfrak{g}) := -J\delta_{\mathfrak{g}[v'v],0}(-1)^{\mathfrak{g}[vv+\hat{u}]}, \quad h_{v,2}(\mathfrak{g}) := -J\delta_{\mathfrak{g}[v'v],0}(-1)^{\mathfrak{g}[vv+\hat{v}]},$$
$$h_{v,3}(\mathfrak{g}) := -J\delta_{\mathfrak{g}[v'v],0}(-1)^{\mathfrak{g}[vv+\hat{w}]}, \quad h_{v,4}(\mathfrak{g}) := -J\kappa\,\delta_{\mathfrak{g}[v'v],1}, \tag{40}$$

where the branching structure of $\bigotimes_v^{\mathbb{I}}$ is that given in eq. (37). We can readily confirm that $\mathbb{h}_{v,4}$ acts as $-J\kappa\sigma_v^x$, whereas local operators $\mathbb{h}_{v,n=1,2,3}$ act as $-J\sigma_v^z\sigma_{v+\hat{u}}^z$, $-J\sigma_v^z\sigma_{v+\hat{v}}^z$ and $-J\sigma_v^z\sigma_{v+\hat{w}}^z$, respectively. Putting everything together, we recover Hamiltonian (1) for $\tilde{\kappa} = 0$. By construction, this model has a $2\mathsf{Vec}_{\mathbb{Z}_2}$ symmetry such that the two simple objects $\mathsf{Vec}_0$ and $\mathsf{Vec}_1$ in $2\mathsf{Vec}_{\mathbb{Z}_2}$, where $0, 1 \in \mathbb{Z}_2$, are identified with the surface operators $\mathcal{O}^0$ and $\mathcal{O}^1$ defined in sec. 2, respectively. The fact that the deformation class of models generated by (40) does not host any non-trivial line operators then follows from $\mathsf{Hom}_{2\mathsf{Vec}_{\mathbb{Z}_2}}(\mathsf{Vec}_{g_1}, \mathsf{Vec}_{g_2}) \cong \delta_{g_1,g_2}\mathsf{Vec}$. Note that more complicated models hosting the same $2\mathsf{Vec}_{\mathbb{Z}_2}$ (sub-)symmetry can be readily defined in a similar fashion. For instance, we can write the spin-1/2 XY model as $\mathbb{H} = \sum_{(v_1v_2)\subset\Sigma_\triangle}\sum_{n=1}^2 \mathbb{h}_{v_1,n}\mathbb{h}_{v_2,n}$ such that

$$h_{v,1}(\mathfrak{g}) := -J\delta_{\mathfrak{g}[v'v],1}, \quad h_{v,2}(\mathfrak{g}) := -J\delta_{\mathfrak{g}[v'v],1}i(-1)^{\mathfrak{g}[v]}. \tag{41}$$

## 3.2 Dual Hamiltonians

Given a Hamiltonian $\mathbb{H} = \sum_v\sum_n \mathbb{h}_{v,n}$ with local operators $\mathbb{h}_{v,n}$ as defined in eq. (39), we shall now construct dual models. In sec. 3.4, we shall relate these various dual models to twisted gauging of the $G$ symmetry or sub-symmetries thereof. Our strategy goes as follows: Any finite group $G$ gives rise to an (abstract) algebra of local operators, in such a way that products of local operators only make use of the multiplication in $G$. A duality class of models is then determined by choosing certain linear combinations of local operators in the algebra. This choice is made through the set of coefficients $\{h_{v,n}(\mathfrak{g})\}_\mathfrak{g}$ over $\mathfrak{g} \in Z^1(\bigotimes_v^{\mathbb{I}}, G)$ in our context. This means that the group $G$ together with the collection $h_{v,n}$ of coefficients fully determine the physical characteristics of the duality class of models as encoded into their common spectrum. Notice that we have not yet specified explicit matrix/lattice representations of these local operators on a chosen Hilbert space. As a matter of fact, picking a representative of a duality class of models precisely corresponds to choosing such a matrix representation. Loosely speaking, this boils down to identifying a collection of degrees of freedom providing a particular physical realisation of the properties encoded into the spectrum. In other words, maintaining the same linear combination of symmetric operators, while choosing another matrix realisation, yields a dual model. We explain below how such choices are made, thereby defining duality classes of (2+1)d Hamiltonian models.

We begin our construction by noticing that picking a gauge field $\mathfrak{g} \in Z^1(\bigotimes_v^{\mathbb{I}}, G)$ amounts to assigning a simple object $\mathfrak{g}[v_1v_2] \equiv \mathsf{Vec}_{\mathfrak{g}[v_1v_2]}$ in $2\mathsf{Vec}_G$ to every edge $(v_1v_2) \subset \bigotimes_v^{\mathbb{I}}$ such that $\mathsf{Vec}_{\mathfrak{g}[v_1v_2]} \odot \mathsf{Vec}_{\mathfrak{g}[v_2v_3]} \cong \mathsf{Vec}_{\mathfrak{g}[v_1v_3]}$ for every 2-simplex $(v_1v_2v_3) \subset \bigotimes_v^{\mathbb{I}}$. In this context, picking an assignment $\mathfrak{m} \in C^0(\bigotimes_v^{\mathbb{I}}, G)$ such that $\mathrm{d}\mathfrak{m} = \mathfrak{g}$ amounts to assigning simple objects $\mathfrak{m}[v_1] \equiv \mathsf{Vec}_{\mathfrak{m}[v_1]}$ in $2\mathsf{Vec}_G$ such that $\mathsf{Vec}_{\mathfrak{g}[v_1v_2]} \odot \mathsf{Vec}_{\mathfrak{m}[v_2]} \cong \mathsf{Vec}_{\mathfrak{m}[v_1]}$ for every edge $(v_1v_2) \subset \bigotimes_v^{\mathbb{I}}$. We think of this latter assignment as making a choice of degrees of freedom, and thus a choice of microscopic Hilbert space. As will become clear in the following, this choice amounts to considering $2\mathsf{Vec}_G$ as a *module 2-category* over itself, inviting us to replace $2\mathsf{Vec}_G$ by another module 2-category. In that spirit, let us first review the notion of module 2-category over $2\mathsf{Vec}_G$ as considered in ref. [44, 53].

Succinctly, a module 2-category over $2\mathsf{Vec}_G$ is a 2-category with a $G$-action. More precisely, we define a (left) $2\mathsf{Vec}_G$-module 2-category as a quadruple $(\mathcal{M}, \triangleright, \alpha^\triangleright, \pi^\triangleright)$ consisting of a ($\mathbb{C}$-linear finite semisimple) 2-category $\mathcal{M}$, a binary action 2-functor $\triangleright : 2\mathsf{Vec}_G \times \mathcal{M} \to \mathcal{M}$ and an adjoint natural 2-equivalence $\alpha^\triangleright : (- \odot -) \triangleright - \xrightarrow{\sim} - \triangleright (- \triangleright -)$ satisfying a 'pentagon axiom' up

to an invertible modification $\pi^{\triangleright}$ whose components $\pi^{\triangleright}_{\mathrm{Vec}_{g_1},\mathrm{Vec}_{g_2},\mathrm{Vec}_{g_3},M}$ are defined via

$$
\begin{array}{c}
(\mathrm{Vec}_{g_1} \odot (\mathrm{Vec}_{g_2} \odot \mathrm{Vec}_{g_3})) \triangleright M \xrightarrow{\alpha^{\triangleright}_{\mathrm{Vec}_{g_1},\mathrm{Vec}_{g_2 g_3},M}} \mathrm{Vec}_{g_1} \triangleright ((\mathrm{Vec}_{g_2} \odot \mathrm{Vec}_{g_3}) \triangleright M)
\end{array}
$$

(42)

for every $g_1, g_2, g_3 \in G$ and $M \in \mathcal{M}$. The invertible modification $\pi^{\triangleright}$, which shall be referred to as the *left module pentagonator*, is required to satisfy an 'associahedron axiom'. For convenience, we shall spell out this axiom employing an alternative to commutative diagrams in terms of *string diagrams*, whereby regions represent objects, strings 1-morphisms, and blobs 2-morphisms. In practice, we shall omit labelling regions but the corresponding objects can be recovered from the string labels. On these diagrams, compositions of 1-morphisms is read from left to right, whereas the (vertical) composition of 2-morphisms is read from top to bottom. For instance, the left module pentagonator $\pi^{\triangleright}$ can be equivalently defined via the string diagram:

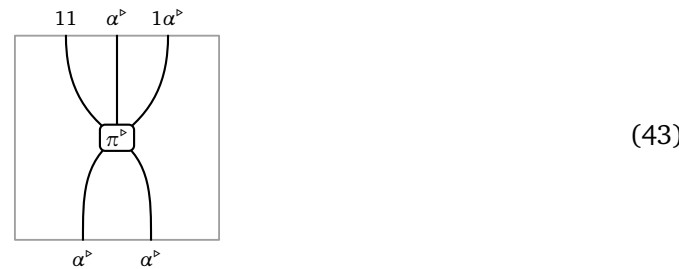

(43)

where we omitted the $\odot$ and $\triangleright$ symbols. The associahedron axiom satisfied by $\pi^{\triangleright}$ can then be conveniently expressed as the following equality of string diagrams:

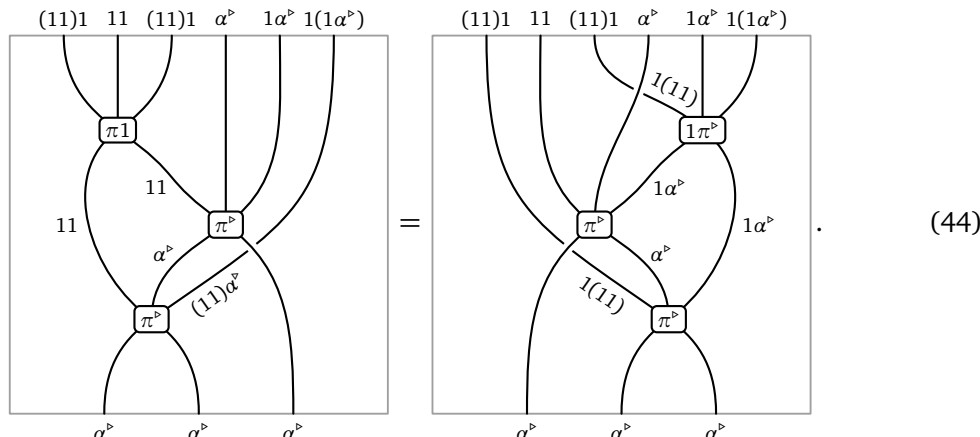

(44)

We are particularly interested in *indecomposable* $2\mathrm{Vec}_G$-module 2-categories constructed as follows [44]:[8] Given a subgroup $A \subseteq G$ and a normalised 3-cocycle $\lambda$ in $H^3(A, \mathrm{U}(1))$, let

---

[8]Note that this construction does not give all $2\mathrm{Vec}_G$-module 2-categories. Physically, it only gives those module 2-categories that correspond to either spontaneously breaking the global symmetry down to a subgroup and/or

$\mathcal{M}(A, \lambda)$ be a 2-category with object-set the set $G/A$ of left cosets. A left $2\mathrm{Vec}_G$-module structure can be defined on $\mathcal{M}(A, \lambda)$ via $\mathrm{Vec}_g \triangleright M := (g\,\mathrm{r}(M))A$ for any $g \in G$ and $M \in G/A$, where $\mathrm{r} : G/A \to G$ assigns to every left coset its representative in $G$. Notice that in general we have $g\,\mathrm{r}(M) \neq \mathrm{r}(\mathrm{Vec}_g \triangleright M)$ and we denote by $a_{g,M}$ the group element in $A$ satisfying

$$g\,\mathrm{r}(M) = \mathrm{r}(\mathrm{Vec}_g \triangleright M) a_{g,M} \,. \tag{45}$$

Associativity of the multiplication in $G$ imposes

$$a_{g_1 g_2, M} = a_{g_1, \mathrm{Vec}_{g_2} \triangleright M} \, a_{g_2, M} \,, \quad \forall\, g_1, g_2 \in G, \text{ and } M \in G/A. \tag{46}$$

Endowing the Abelian group $\mathrm{Hom}(G/A, \mathrm{U}(1))$ with a left $G$-module structure, we consider the 3-cochain $\pi^\triangleright \in C^3(G, \mathrm{Hom}(G/A, \mathrm{U}(1)))$ defined as

$$\pi^\triangleright(g_1, g_2, g_3)(M) := \lambda(a_{g_1, \mathrm{Vec}_{g_2 g_3} \triangleright M}, a_{g_2, \mathrm{Vec}_{g_2} \triangleright M}, a_{g_3, M}), \quad \forall\, g_1, g_2, g_3 \in G, \text{ and } M \in G/A. \tag{47}$$

In virtue of the 3-cocycle condition $\mathrm{d}\lambda = 1$ and eq. (46), we have

$$\begin{aligned}
\pi^\triangleright&(g_2, g_3, g_4)(M)\, \pi^\triangleright(g_1, g_2 g_3, g_4)(M)\, \pi^\triangleright(g_1, g_2, g_3)(\mathrm{Vec}_{g_4} \triangleright M) \\
&= \pi^\triangleright(g_1 g_2, g_3, g_4)(M)\, \pi^\triangleright(g_1, g_2, g_3 g_4)(M), \quad \forall\, g_1, g_2, g_3, g_4 \in G, \text{ and } M \in G/A,
\end{aligned} \tag{48}$$

so that $\pi^\triangleright$ is a $\mathrm{Hom}(G/A, \mathrm{U}(1))$-valued 3-cocycle of $G$. Defining the invertible modification $\pi^\triangleright$ with components

$$\pi^\triangleright_{\mathrm{Vec}_{g_1}, \mathrm{Vec}_{g_2}, \mathrm{Vec}_{g_3}, M} := \pi^\triangleright(g_1, g_2, g_3)(M) \cdot 1_{(g_1 g_2 g_3 \mathrm{r}(M))A}, \quad \forall\, g_1, g_2, g_3 \in G, \text{ and } M \in G/A, \tag{49}$$

we finally obtain that the quadruple $\mathcal{M}(A, \lambda) \equiv (\mathcal{M}(A, \lambda), \triangleright, 1, \pi^\triangleright)$ does define a left $2\mathrm{Vec}_G$-module 2-category. For any group $G$, we can always choose the subgroup $A$ to be either the trivial subgroup $\{\mathbb{1}_G\}$ or the whole group $G$, and the 3-cocycle to be trivial. The corresponding module 2-categories are $\mathcal{M}(\{\mathbb{1}_g\}, 1) \cong 2\mathrm{Vec}_G$ and $\mathcal{M}(G, 1) \cong 2\mathrm{Vec}$ with action 2-functors given by the monoidal product in $2\mathrm{Vec}_G$ and the forgetful functor $2\mathrm{Vec}_G \to 2\mathrm{Vec}$, respectively. Let us now put these module 2-categories to use in order to construct dual Hamiltonians. Given a vertex $\mathsf{v} \subset \Sigma_\triangle$, a gauge field $\mathfrak{g} \in Z^1(\otimes^{\mathbb{0}}_{\mathsf{v}}, G)$ and a $2\mathrm{Vec}_G$-module 2-category $\mathcal{M} \equiv \mathcal{M}(A, \lambda)$ as defined above, let us consider an assignment $\mathfrak{m}$ of simple objects $\mathfrak{m}[\mathsf{v}_1] \in \mathcal{M}$ to every vertex $\mathsf{v}_1 \subset \otimes^{\mathbb{0}}_{\mathsf{v}}$ such that

$$\mathfrak{m}[\mathsf{v}_1] \overset{!}{=} \mathrm{Vec}_{\mathfrak{g}[\mathsf{v}_1 \mathsf{v}_2]} \triangleright \mathfrak{m}[\mathsf{v}_2], \quad \forall\, (\mathsf{v}_1 \mathsf{v}_2) \subset \otimes^{\mathbb{0}}_{\mathsf{v}}. \tag{50}$$

We notate via $C^0_{\mathfrak{g}}(\otimes^{\mathbb{0}}_{\mathsf{v}}, \mathcal{M})$ the set of assignments $\mathfrak{m}$ fulfilling conditions (50). Given such a pair $(\mathfrak{g}, \mathfrak{m}) \in Z^1(\otimes^{\mathbb{0}}_{\mathsf{v}}, G) \times C^0_{\mathfrak{g}}(\otimes^{\mathbb{0}}_{\mathsf{v}}, \mathcal{M})$, let us introduce the following phase factor:

$$\pi^\triangleright_{\mathsf{v}}(\mathfrak{g}, \mathfrak{m}) := \prod_{(\mathsf{v}_1 \mathsf{v}_2 \mathsf{v}_3 \mathsf{v}_4) \subset \otimes^{\mathbb{0}}_{\mathsf{v}}} \pi^\triangleright(\mathfrak{g}[\mathsf{v}_1 \mathsf{v}_2], \mathfrak{g}[\mathsf{v}_2 \mathsf{v}_3], \mathfrak{g}[\mathsf{v}_3 \mathsf{v}_4])(\mathfrak{m}[\mathsf{v}_4])^{\epsilon(\mathsf{v}_1 \mathsf{v}_2 \mathsf{v}_3 \mathsf{v}_4)}, \tag{51}$$

where $\pi^\triangleright$ is the $\mathrm{Hom}(G/A, \mathrm{U}(1))$-valued 3-cocycle of $G$ defined in eq. (47), and $\epsilon(\mathsf{v}_1 \mathsf{v}_2 \mathsf{v}_3 \mathsf{v}_4) = \pm 1$ depends on the orientation of the 3-simplex $(\mathsf{v}_1 \mathsf{v}_2 \mathsf{v}_3 \mathsf{v}_4)$. Borrowing the notations of sec. 3.1, we finally define new local operators as follows:

$$\mathbb{h}^{\mathcal{M}}_{\mathsf{v}, n} = \sum_{\mathfrak{g} \in Z^1(\otimes^{\mathbb{0}}_{\mathsf{v}}, G)} h_{\mathsf{v}, n}(\mathfrak{g}) \sum_{\mathfrak{m} \in C^0_{\mathfrak{g}}(\otimes^{\mathbb{0}}_{\mathsf{v}}, \mathcal{M})} \pi^\triangleright_{\mathsf{v}}(\mathfrak{g}, \mathfrak{m}) \left| (\mathfrak{g}, \mathfrak{m})[\otimes^1_{\mathsf{v}}] \right\rangle \left\langle (\mathfrak{g}, \mathfrak{m})[\otimes^0_{\mathsf{v}}] \right|. \tag{52}$$

---

pasting a symmetry-protected topological phase labelled by a 3-cocycle of the preserved subgroup. Notably, we do not discuss those module 2-categories that correspond to coupling to a inherently two-dimensional non-anomalous topological order. It is expected that these topological orders would be contained in the completion of the 3-category of $2\mathrm{Vec}_G$-module 2-categories [23].

Notice immediately that choosing $\mathcal{M}$ to be $2\mathsf{Vec}_G$ itself, we recover local operators $\mathbb{h}_{\mathsf{v},n}$ defined in eq. (39). We are now ready to state one of the main results of this manuscript: *Hamiltonian models that only differ in a choice of* $2\mathsf{Vec}_G$-*module 2-category are dual to one another via definition* (52) *of the local operators.* In other words, Hamiltonians $\mathbb{H}^{\mathcal{M}} = \sum_{\mathsf{v}} \sum_n \mathbb{h}_{\mathsf{v},n}^{\mathcal{M}}$ and $\mathbb{H}^{\mathcal{M}'} = \sum_{\mathsf{v}} \sum_n \mathbb{h}_{\mathsf{v},n}^{\mathcal{M}'}$ for any two indecomposable $2\mathsf{Vec}_G$-module 2-categories $\mathcal{M} \equiv \mathcal{M}(A, \lambda)$ and $\mathcal{M}' \equiv \mathcal{M}(A', \lambda')$ are dual to one another.

As motivated above, duality between $\mathbb{H}^{\mathcal{M}}$ and $\mathbb{H}^{\mathcal{M}'}$ follows from the fact $2\mathsf{Vec}_G$-module 2-categories $\mathcal{M}$ and $\mathcal{M}'$ merely encode distinct matrix realisations of the same local operators. More concretely, regardless of the choice of $2\mathsf{Vec}_G$-module 2-category $\mathcal{M}$, the set of local operators $\mathbb{h}_{\mathsf{v},n}^{\mathcal{M}}$ generate the same algebra of operators characteristic of the duality class of models. In the same vein as the lower-dimensional study carried out in ref. [89], we can readily confirm that products of local operators indeed only involve the group $G$ and coefficients $h_{\mathsf{v},n}$. Concretely, computing products of local operators (52) involve algebraic manipulations that geometrically translate into three-dimensional Pachner moves, which are encoded into the associahedron axiom given in eq. (44). The associahedron axiom dictates that products of operators only depend on the monoidal structure of $2\mathsf{Vec}_G$ via the monoidal pentagonator $\pi$, which happens to be trivial, and is a fortiori independent of $\mathcal{M}$. An alternative justification consists in showing that the Hamiltonian $\mathbb{H}^{\mathcal{M}}$ with $\mathcal{M} \equiv \mathcal{M}(A, \lambda)$ is the result of the $\lambda$-twisted gauging of the $A$ sub-symmetry of $\mathbb{H}$. This will be the purpose of sec. 3.4.

Importantly, dualities as considered in this manuscript systematically map symmetric local operators to dual symmetric local operators—this almost tautologically follows from our definition of a duality as a change of matrix realisation of the local operator encoded into a choice of $2\mathsf{Vec}_G$-module 2-category—whereas *non-symmetric* local operators are mapped to dual *non-local* non-symmetric operators. These various mappings are realised via (typically non-local) lattice *duality* operators that transmute in particular local operators into one another. Below, we explicitly construct these lattice operators.

**Back to the transverse-field Ising model:** We explained in the previous section how to recast the transverse-field Ising model within our framework. The input fusion 2-category being $2\mathsf{Vec}_{\mathbb{Z}_2}$, we distinguish three choices of module 2-categories, namely $2\mathsf{Vec}_{\mathbb{Z}_2}$ itself, $2\mathsf{Vec}$ and $2\mathsf{Vec}^{\lambda}$, respectively, where $\lambda$ corresponds to the non-identity element in $H^3(\mathbb{Z}_2, \mathsf{U}(1)) \simeq \mathbb{Z}_2$. Given the coefficients (40), it readily follows from definition (52) of the local operators that $\mathbb{h}_{\mathsf{v},4}^{2\mathsf{Vec}}$ acts as $-J\kappa \prod_{\mathsf{e} \supset \mathsf{v}} \sigma_{\mathsf{e}}^x$, whereas local operators $\mathbb{h}_{\mathsf{v},n=1,2,3}^{2\mathsf{Vec}}$ acts as $-J\sigma_{(\mathsf{v}\mathsf{v}+\hat{u})}^z$, $-J\sigma_{(\mathsf{v}\mathsf{v}+\hat{v})}^z$ and $-J\sigma_{(\mathsf{v}\mathsf{v}+\hat{w})}^z$, respectively. Putting everything together, we recover the Hamiltonian (10) with $\tilde{\kappa} = 0$ resulting from gauging the $\mathbb{Z}_2$ symmetry of (1). Choosing instead the $2\mathsf{Vec}_{\mathbb{Z}_2}$-module 2-category $2\mathsf{Vec}^{\lambda}$ amounts to the $\lambda$-twisted gauging of the $\mathbb{Z}_2$ symmetry and results in Hamiltonian (10) with $\kappa = 0$.

## 3.3 Duality operators

We are interested in dualities between Hamiltonians $\mathbb{H}^{\mathcal{M}}$ and $\mathbb{H}^{\mathcal{M}'}$ whose local operators $\mathbb{h}_{\mathsf{v},n}^{\mathcal{M}}$ and $\mathbb{h}_{\mathsf{v},n}^{\mathcal{M}'}$ only differ in the choice of $2\mathsf{Vec}_G$-module 2-category. Therefore, a duality operator should have the interpretation of a map between the module 2-categories that is compatible with the action of $2\mathsf{Vec}_G$. More concretely, given a pair of (left) $2\mathsf{Vec}_G$-module 2-categories $(\mathcal{M}, \rhd, \alpha^{\rhd}, \pi^{\rhd})$ and $(\mathcal{M}', \rhd, \alpha^{\rhd}, \pi^{\rhd})$, we define a $2\mathsf{Vec}_G$-*module 2-functor* between them as a triple $(F, \omega, \Omega)$ consisting of a 2-functor $F : \mathcal{M} \to \mathcal{M}'$ and an adjoint natural 2-equivalence $\omega : F(- \rhd -) \xrightarrow{\sim} - \rhd F(-)$, with components $\omega_{\mathsf{Vec}_g, M}$ for $g \in G$ and $M \in \mathcal{M}$, satisfying a 'pentagon axiom' up to an invertible modification $\Omega$ whose components $\Omega_{\mathsf{Vec}_{g_1}, \mathsf{Vec}_{g_2}, M}$ are defined

via

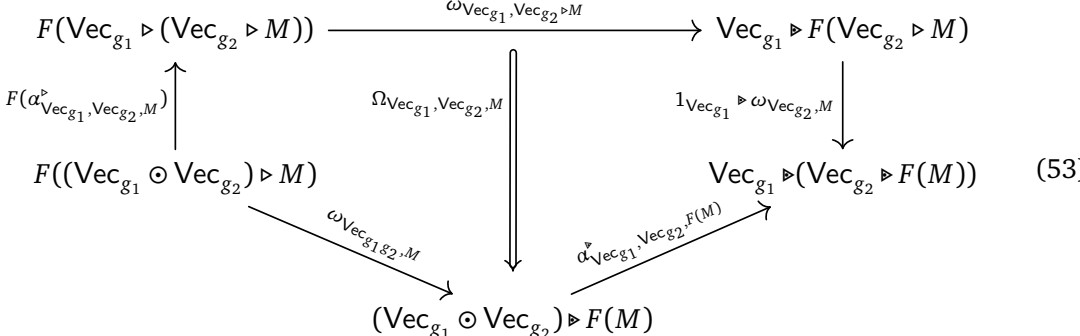

$$(53)$$

for every $g_1, g_2 \in G$ and $M \in \mathcal{M}$. As before, we shall prefer the equivalent definition in terms of the string diagram

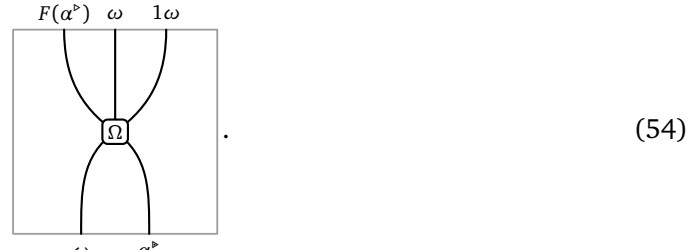

$$(54)$$

This invertible modification $\Omega$ is required to satisfy an 'associahedron axiom' encoded into the following equality of string diagrams:

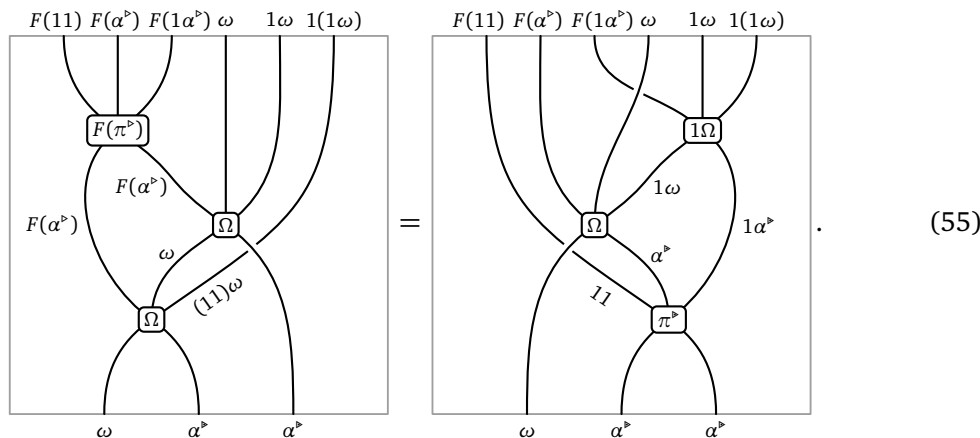

$$(55)$$

Let us now use the data of a module 2-functor $\mathcal{M} \to \mathcal{M}'$ to construct lattice operators that transmute local operators $\mathbb{h}_{\mathsf{v},n}^{\mathcal{M}}$ into $\mathbb{h}_{\mathsf{v},n}^{\mathcal{M}'}$. Since module 2-functors can be composed—and by extension so do the corresponding dualitiy operators—we can focus without loss of generality on $2\mathsf{Vec}_G$-module 2-functors between $2\mathsf{Vec}_G$ itself and $\mathcal{M}' \equiv \mathcal{M}(A', \lambda')$. Every such module 2-functor is of the form $(- \triangleright M', 1, \pi_{-,-,-,M'}^{\triangleright})$, with $M' \in \mathcal{M}'$, in which case the associahedron axiom (55) boils down to (44). In the spirit of our definition of the local operators, let us

consider the complex[9]

$$\bigotimes_{\mathsf{v}} \times \mathbb{I} \equiv \qquad\qquad\qquad , \qquad\qquad (56)$$

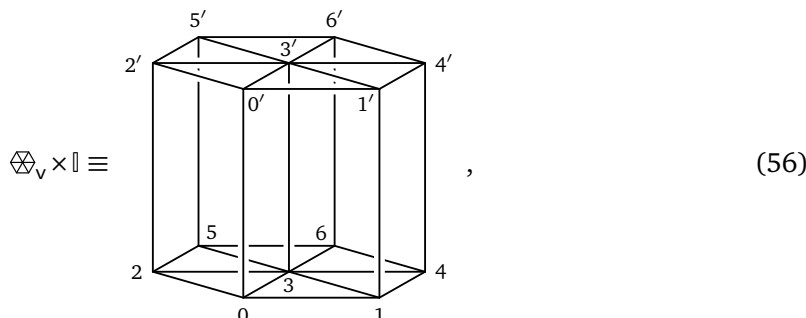

centred around $\mathsf{v} \equiv (3)$. The branching structure agrees with that of eq. (37), and in particular we have $\mathsf{v}'_1 < \mathsf{v}_1$ for every $\mathsf{v}_1 \subset \bigotimes_{\mathsf{v}}$. Given a simple object $M' \in \mathcal{M}'$ interpreted as a $2\mathrm{Vec}_G$-module 2-functor $2\mathrm{Vec}_G \to \mathcal{M}'$, we consider the following assignment of degrees of freedom: First, let $\mathfrak{g} \in Z^1(\bigotimes_{\mathsf{v}}^0, G)$ and $\mathfrak{g}' \in Z^1(\bigotimes_{\mathsf{v}}^1, G)$ such that $\mathfrak{g}[\mathsf{v}_1\mathsf{v}_2] = \mathfrak{g}'[\mathsf{v}'_1\mathsf{v}'_2]$ for every $\mathsf{v}_1, \mathsf{v}_2 \subset \bigotimes_{\mathsf{v}}$. We then consider an assignment $\mathfrak{m}$ of simple objects $\mathfrak{m}[\mathsf{v}_1] \in 2\mathrm{Vec}_G$ to every vertex $\mathsf{v}_1 \subset \bigotimes_{\mathsf{v}}^0$ and an assignment $\mathfrak{m}'$ of simple objects $\mathfrak{m}'[\mathsf{v}'_1] \in \mathcal{M}'$ to every vertex $\mathsf{v}'_1 \subset \bigotimes_{\mathsf{v}}^1$ such that $\mathfrak{m}[\mathsf{v}_1] = \mathrm{Vec}_{\mathfrak{g}[\mathsf{v}_1\mathsf{v}_2]} \odot \mathfrak{m}[\mathsf{v}_2]$ for every $(\mathsf{v}_1\mathsf{v}_2) \in \bigotimes_{\mathsf{v}}^0$, $\mathfrak{m}'[\mathsf{v}'_1] = \mathrm{Vec}_{\mathfrak{g}'[\mathsf{v}'_1\mathsf{v}'_2]} \triangleright \mathfrak{m}'[\mathsf{v}'_2]$ for every $(\mathsf{v}'_1\mathsf{v}'_2) \in \bigotimes_{\mathsf{v}}^1$, and $\mathfrak{m}[\mathsf{v}_1] \triangleright M' = \mathfrak{m}'[\mathsf{v}'_1]$ for every $(\mathsf{v}'_1\mathsf{v}_1) \subset \bigotimes_{\mathsf{v}} \times \mathbb{I}$. As before, we notate via $C_{\mathfrak{g}}^0(\bigotimes_{\mathsf{v}}^0, G)$ the collection of assignments $\mathfrak{m}$ fulfilling the conditions spelt out above. Notice that assignment $\mathfrak{m} \in C_{\mathfrak{g}}^0(\bigotimes_{\mathsf{v}}^0, G)$ together with $M' \in \mathcal{M}'$ uniquely specifies an assignment $\mathfrak{m}'$ via the constraints $\mathfrak{m}[\mathsf{v}_1] \triangleright M' = \mathfrak{m}'[\mathsf{v}'_1]$ for every $(\mathsf{v}'_1\mathsf{v}_1) \subset \bigotimes_{\mathsf{v}} \times \mathbb{I}$, and we denote by $\mathfrak{m} \triangleright M'$ this assignemnt.

Given assignments $(\mathfrak{g}, \mathfrak{m}) \in Z^1(\bigotimes_{\mathsf{v}}^0, G) \times C_{\mathfrak{g}}^0(\bigotimes_{\mathsf{v}}^0, G)$, let us introduce the following phase factor:

$$\pi_{\mathsf{v}}^{\triangleright}(\mathfrak{g}, \mathfrak{m}, M') := \prod_{(\mathsf{v}_1\mathsf{v}_2\mathsf{v}_3) \subset \bigotimes_{\mathsf{v}}^0} \pi^{\triangleright}(\mathfrak{g}[\mathsf{v}_1\mathsf{v}_2], \mathfrak{g}[\mathsf{v}_2\mathsf{v}_3], \mathfrak{m}[\mathsf{v}_3])(M')^{\epsilon(\mathsf{v}_1\mathsf{v}_2\mathsf{v}_3)}, \qquad (57)$$

where $\pi^{\triangleright}$ is the $\mathrm{Hom}(G/A', \mathrm{U}(1))$-valued 3-cocycle of $G$ defined previously, and $\epsilon(\mathsf{v}_1\mathsf{v}_2\mathsf{v}_3) = \pm 1$ depends on the orientation of the 2-simplex $(\mathsf{v}_1\mathsf{v}_2\mathsf{v}_3)$. We finally define the *duality operator* labelled by $M' \in \mathcal{M}'$ acting at vertex $\mathsf{v} \subset \Sigma_{\triangle}$ as follows:

$$\mathbb{d}_{\mathsf{v}}^{M'} = \sum_{\substack{\mathfrak{g} \in Z^1(\bigotimes_{\mathsf{v}}^0, G) \\ \mathfrak{m} \in C_{\mathfrak{g}}^0(\bigotimes_{\mathsf{v}}^0, G)}} \pi_{\mathsf{v}}^{\triangleright}(\mathfrak{g}, \mathfrak{m}, M') |\mathfrak{g}, \mathfrak{m} \triangleright M'\rangle\langle\mathfrak{g}, \mathfrak{m}| . \qquad (58)$$

What is the action of these operators? On the one hand, the operator turns degrees of freedom provided by simple objects in $2\mathrm{Vec}_G$—thought as a module 2-category over itself—into simple objects in $\mathcal{M}'$ via the module 2-functor $- \triangleright M'$. On the other hand, it acts by scalar multiplication by the phase factors $\pi_{\mathsf{v}}^{\triangleright}(\mathfrak{g}, \mathfrak{m}, M')$. It follows that acting with $\mathbb{d}_{\mathsf{v}}^{M'}$ on $\mathbb{h}_{\mathsf{v},n}$ yields $\mathbb{h}_{\mathsf{v},n}^{\mathcal{M}'}$, i.e.,

$$\mathbb{d}_{\mathsf{v}}^{M'} \circ \mathbb{h}_{\mathsf{v},n} = \mathbb{h}_{\mathsf{v},n}^{\mathcal{M}'} \circ \mathbb{d}_{\mathsf{v}}^{M'} . \qquad (59)$$

The only non-trivial aspect to confirm is the compatibility of the various phase factors. It is convenient to do so using the geometrical interpretations of the operators. Geometrically, the

---

[9]Alternative operators more suited to the more general case of an arbitrary fusion 2-category can be found in ref. [44].

commutation relation (59) can be represented as follows:

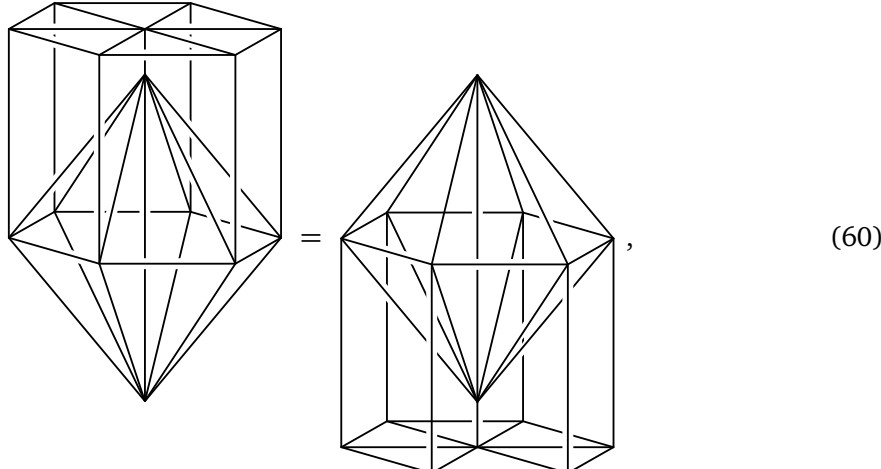

$$\tag{60}$$

where we think of the complex $\bigotimes_{\mathsf{v}}^{\mathbb{l}}$ on the l.h.s. as supporting the local operator $\mathbb{h}_{\mathsf{v},n}$ and that on the r.h.s. as supporting $\mathbb{h}_{\mathsf{v},n}^{\mathcal{M}'}$. Recall that the phase factors entering the definition of $\mathbb{h}_{\mathsf{v},n}$ are trivial, whereas those entering the definition of $\mathbb{h}_{\mathsf{v},n}^{\mathcal{M}'}$ are given by $\pi^{\triangleright}$. Similarly, the phase factors entering the definition of $\mathbb{d}_{\mathsf{v}}^{M'}$ evaluates to $\pi^{\triangleright}$. It follows from the associahedron axiom satisfied by the module pentagonator $\pi^{\triangleright}$—or rather the cocycle condition of the 3-cocycle it evaluates to—as well as the fact that every pair of neighbouring 3-simplices share a 2-simplex, that the commutation relation is satisfied. Indeed, eq. (55) graphically translates as

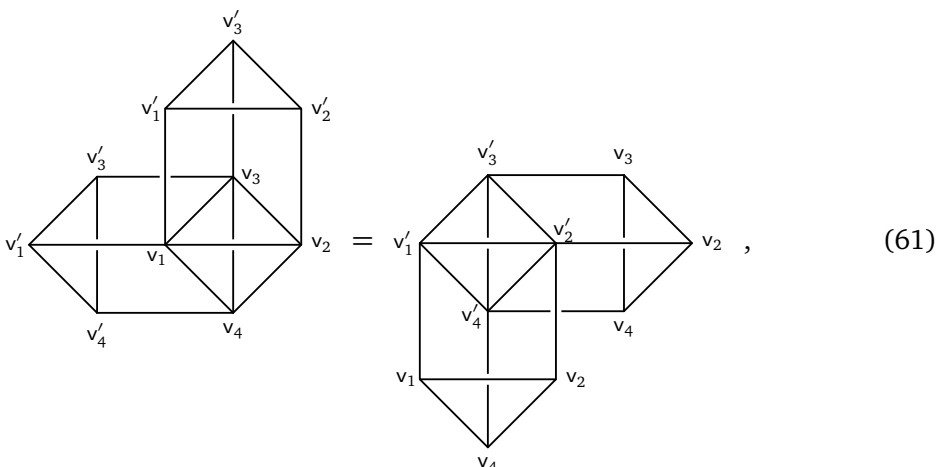

$$\tag{61}$$

where vertices carrying the same label are identified. Applying the assignment rules of degrees of freedom presented above, we find that the phase factors associated with the 3-simplices on the l.h.s. and r.h.s. are 1 and $\pi^{\triangleright}(\mathfrak{g}[v_1 v_2], \mathfrak{g}[v_2 v_3], \mathfrak{g}[v_3 v_4])(\mathfrak{m}'[v'_4])$, respectively. Similarly, we associate to the prisms on the l.h.s. the phase factors $\pi^{\triangleright}(\mathfrak{g}[v_1 v_2], \mathfrak{g}[v_2 v_3], \mathfrak{m}[v_3])(M')$ and $\pi^{\triangleright}(\mathfrak{g}[v_1 v_3], \mathfrak{g}[v_3 v_4], \mathfrak{m}[v_4])(M')$, whereas we associate to the prisms on the r.h.s. the phase factors $\pi^{\triangleright}(\mathfrak{g}[v_2 v_3], \mathfrak{g}[v_3 v_4], \mathfrak{m}[v_4])(M')$ and $\pi^{\triangleright}(\mathfrak{g}[v_1 v_2], \mathfrak{g}[v_2 v_4], \mathfrak{m}[v_4])(M')$. Choosing $\mathfrak{g}[v_1 v_2] \equiv g_1$, $\mathfrak{g}[v_2 v_3] \equiv g_2$, $\mathfrak{g}[v_3 v_4] \equiv g_3$ and $\mathfrak{m}[v_4] \equiv g_4 \in G$, it follows from the various assignment rules—e.g. $\mathfrak{m}'[v'_4] = \mathfrak{m}[v_4] \triangleright M' = \mathrm{Vec}_{g_4} \triangleright M'$—that eq. (61) exactly encodes eq. (48). Applying eq. (61) for every 3-simplex in $\bigotimes_{\mathsf{v}}^{\mathbb{l}}$, we find that all the phase factors of the form $\pi^{\triangleright}(\mathfrak{g}[v_1 v_3], \mathfrak{g}[v_3 v_4], \mathfrak{m}[v_4])(M')$ and $\pi^{\triangleright}(\mathfrak{g}[v_2 v_3], \mathfrak{g}[v_3 v_4], \mathfrak{m}[v_4])(M')$ cancel two-by-two resulting in eq. (60). More generally, given local operators $\mathbb{h}_{\mathsf{v},n}^{\mathcal{M}}$ and $\mathbb{h}_{\mathsf{v},n}^{\mathcal{M}'}$, the analogue of eq. (61) will be guaranteed by the associahedron axiom (55) fulfilled by the $2\mathrm{Vec}_G$-module structure of the 2-functor $\mathcal{M} \to \mathcal{M}'$ (see sec. 3.5 for the case of module 2-endofunctors).

So we have found duality operators performing the transmutation of local operators $\mathbb{h}_{\mathsf{v},n}$ into $\mathbb{h}_{\mathsf{v},n}^{\mathcal{M}'}$. In order to perform this operation to the whole Hamiltonian, it suffices to extend the definition of our duality operator to the whole $\Sigma_{\triangle}$ following exactly the same construction:

$$
\mathbb{d}^{M'} = \sum_{\substack{\mathfrak{g} \in Z^1(\Sigma_{\triangle}^0, G) \\ \mathfrak{m} \in C_{\mathfrak{g}}^0(\Sigma_{\triangle}^0, G)}} \left( \prod_{(\mathsf{v}_1\mathsf{v}_2\mathsf{v}_3) \subset \Sigma_{\triangle}^0} \pi^{\triangleright}(\mathfrak{g}[\mathsf{v}_1\mathsf{v}_2], \mathfrak{g}[\mathsf{v}_2\mathsf{v}_3], \mathfrak{m}[\mathsf{v}_3])(M')^{\epsilon(\mathsf{v}_1\mathsf{v}_2\mathsf{v}_3)} \right) |\mathfrak{g}, \mathfrak{m} \triangleright M'\rangle\langle\mathfrak{g}, \mathfrak{m}| . \quad (62)
$$

Let us conclude with a couple of important remarks: Firstly, duality operators are oblivious to the details of the definition of the local operators. In particular they act in the same way regardless of the coefficients $h_{\mathsf{v},n}$, and as such are valid for infinitely many lattice models. This is because duality operators only care about matrix/lattice realisations of a given symmetry and not specific choices of algebra of local operators. In other words, a given duality operator will systematically transmute every symmetric operator with respect to a given lattice realisation of a symmetry into a symmetric operator with respect to another realisation, regardless of the precise definition of these operators. These symmetries will be analysed in detail in sec. 3.5. Secondly, the knowledge of such a duality operator is not sufficient to rigorously write down an *isometry* mapping the corresponding Hamiltonians to one another. As detailed in ref. [90] for the lower-dimensional setting, defining such an isometry would require analysing all the *topological sectors* of the models. We comment on this aspect in sec. 6 but a detailed analysis will be carried out elsewhere.

**Back to the transverse-field Ising model:** We established in the previous section how, given the coefficients (40), choosing the $2\mathsf{Vec}_{\mathbb{Z}_2}$-module 2-categories $2\mathsf{Vec}_{\mathbb{Z}_2}$, $2\mathsf{Vec}$ or $2\mathsf{Vec}^{\lambda}$ yields the transverse-field Ising model, its $\mathbb{Z}_2$-gauged dual or its $\lambda$-twisted $\mathbb{Z}_2$-gauged dual. The results obtained above now allow us to construct the lattice operators performing the transmutations of the corresponding local symmetric operators. Succinctly, there is a unique $2\mathsf{Vec}_{\mathbb{Z}_2}$-module functor from $2\mathsf{Vec}_{\mathbb{Z}_2}$ to $2\mathsf{Vec}$, namely the forgetful functor, identified with the unique simple object $\mathsf{Vec} \in 2\mathsf{Vec}$. The corresponding duality operator acts as $\mathbb{d}^{\mathsf{Vec}} : |\mathfrak{g}, \mathfrak{m}\rangle \mapsto |\mathfrak{g}, \mathfrak{m} \triangleright \mathsf{Vec}\rangle$ for any $(\mathfrak{g}, \mathfrak{m}) \in Z^1(\Sigma_{\triangle}, \mathbb{Z}_2) \times C_{\mathfrak{g}}^0(\Sigma_{\triangle}, \mathsf{Vec}_{\mathbb{Z}_2})$. But $\mathfrak{m} \triangleright \mathsf{Vec} \cong \mathsf{Vec}$ and $\mathfrak{g}$ is fully constrained by $\mathfrak{m}$ according to $\mathfrak{m}[\mathsf{v}_1] = \mathsf{Vec}_{\mathfrak{g}[\mathsf{v}_1\mathsf{v}_2]} \odot \mathfrak{m}[\mathsf{v}_2]$ for any $(\mathsf{v}_1\mathsf{v}_2) \subset \Sigma_{\triangle}$. It follows that the duality operator effectively acts as $\mathbb{d}^{\mathsf{Vec}} : |\mathfrak{m}\rangle \mapsto |\mathbb{d}\mathfrak{m}\rangle$ in the notation of (39). It readily follows that $\mathbb{d}^{\mathsf{Vec}} : \sigma_{s(e)}^z \sigma_{t(e)}^z \mapsto \sigma_e^z$ and $\mathbb{d}^{\mathsf{Vec}} : \sigma_{\mathsf{v}}^x \mapsto \prod_{e \supset \mathsf{v}} \sigma_e^x$, as expected. The treatment of the duality $2\mathsf{Vec}_{\mathbb{Z}_2} \to 2\mathsf{Vec}^{\lambda}$ follows the same steps. Note that these duality operators were already obtained in [44] exploiting the graphical calculus of monoidal 2-categories.

## 3.4  Duality as twisted gauging

Let us now clarify in which sense the dual Hamiltonians described above are the results of applying some (twisted) gauging to the original $G$-symmetric Hamiltonian. We begin by providing an alternative expression for local operators (52). By definition of our notations, local operators $\mathbb{h}_{\mathsf{v},n}^{\mathcal{M}}$ act on degrees of freedom located at vertices and edges labelled by simple objects in $\mathcal{M}(A, \lambda)$ and $2\mathsf{Vec}_G$, respectively. However, these degrees of freedom must satisfy (50), which we shall think of as *kinematical* constraints. Resolving these kinematical constraints allow us to consider a smaller effective microscopic Hilbert space. Consider for instance the $2\mathsf{Vec}_G$-module 2-category $\mathcal{M}(\{\mathbb{1}_G\}, 1) \cong 2\mathsf{Vec}_G$. A choice $\mathfrak{m}$ of assignments of objects in $\mathcal{M}$ to every vertex $\mathsf{v}_1 \subset \bigotimes_{\mathsf{v}}^{\mathbb{I}}$ fully constraints $\mathfrak{g} \in Z^1(\bigotimes_{\mathsf{v}}^{\mathbb{I}}, G)$ in virtue of eq. (50), so that we should consider the effective Hilbert space $\bigotimes_{\mathsf{v}} \mathbb{C}[G]$, at which point the operators (52) boil down to (39) as expected. More generally, given $\mathcal{M}(A, \lambda)$ and a pair $(\mathfrak{m}[\mathsf{v}_1], \mathfrak{m}[\mathsf{v}_2])$ of simple objects in $\mathcal{M}(A, \lambda)$, there are exactly $|A|$-many distinct group elements $g \in G$ such

that $\mathfrak{m}[v_1] \cong \mathsf{Vec}_g \rhd \mathfrak{m}[v_2]$.[10] Consequently, local operators (52) effectively act on a microscopic Hilbert space constituted of degrees of freedom at vertices labelled by simple objects in $\mathcal{M}(A, \lambda)$ and degrees of freedom at edges labelled by group elements in $A$—or rather simple objects in $2\mathsf{Vec}_A$. Given a pair $(\mathfrak{g}, \mathfrak{m}) \in Z^1(\otimes_v^{\mathbb{I}}, G) \times C^0_{\mathfrak{g}}(\otimes_v^{\mathbb{I}}, \mathcal{M})$, we denote by $\mathfrak{a}_{\mathfrak{g},\mathfrak{m}}$ the assignment of group elements $\mathfrak{a}_{\mathfrak{g},\mathfrak{m}}[v_1 v_2]$ to every edge $(v_1 v_2) \subset \otimes^{\mathbb{I}}$, where

$$\mathfrak{a}_{\mathfrak{g},\mathfrak{m}}[v_1 v_2] := r(\mathfrak{m}[v_1])^{-1} \mathfrak{g}[v_1 v_2] \, r(\mathfrak{m}[v_2]) \equiv a_{\mathfrak{g}[v_1 v_2], \mathfrak{m}[v_2]}, \tag{63}$$

where we used in the last identification the notation introduced in eq. (45) when defining $\mathcal{M}(A, \lambda)$. Note that in virtue of eq. (50), we have $\mathfrak{a}_{\mathfrak{g},\mathfrak{m}}[v_1 v_2]\mathfrak{a}_{\mathfrak{g},\mathfrak{m}}[v_2 v_3] = \mathfrak{a}_{\mathfrak{g},\mathfrak{m}}[v_1 v_3]$ for every $(v_1 v_2 v_3) \subset \otimes^{\mathbb{I}}$. Recalling the definition of the module pentagonator $\pi^{\rhd}$, we introduce

$$\pi_v^{\rhd}(\mathfrak{a}_{\mathfrak{g},\mathfrak{m}}) := \prod_{(v_1 v_2 v_3 v_4) \subset \otimes_v^{\mathbb{I}}} \lambda(\mathfrak{a}_{\mathfrak{g},\mathfrak{m}}[v_1 v_2], \mathfrak{a}_{\mathfrak{g},\mathfrak{m}}[v_2 v_3], \mathfrak{a}_{\mathfrak{g},\mathfrak{m}}[v_3 v_4])^{\epsilon(v_1 v_2 v_3 v_4)}. \tag{64}$$

Putting everything together, we find that local operators (52) act on the effective Hilbert space as

$$\mathbb{h}_{v,n}^{\mathcal{M}(A,\lambda)} \stackrel{\text{eff.}}{=} \sum_{\mathfrak{g} \in Z^1(\otimes_v^{\mathbb{I}}, G)} h_{v,n}(\mathfrak{g}) \sum_{\mathfrak{m} \in C^0_{\mathfrak{g}}(\otimes_v^{\mathbb{I}}, \mathcal{M})} \pi_v^{\rhd}(\mathfrak{a}_{\mathfrak{g},\mathfrak{m}}) \big| (\mathfrak{a}_{\mathfrak{g},\mathfrak{m}}, \mathfrak{m})\big[\otimes_v^1\big]\big\rangle \big\langle (\mathfrak{a}_{\mathfrak{g},\mathfrak{m}}, \mathfrak{m})\big[\otimes_v^0\big]\big|. \tag{65}$$

In practice, this is the expression we shall employ when discussing explicit models.

Let now employ this expression to clarify why $\mathbb{h}_{v,n}^{\mathcal{M}(A,\lambda)}$ is the result of a $\lambda$-twisted gauging of the $A$ sub-symmetry of the $G$-symmetric Hamiltonian defined in terms of local operators (39). Consider the (untwisted) gauging of the whole $G$ symmetry. Typically, this operation goes as follows: The macroscopic Hilbert space is enlarged by the introduction of a $G$-gauge field and a (local) Gauß constraint is imposed at every vertex in such a way that they commute with one another. We then require the Hamiltonian to commute with such Gauß constraints, which is accomplished by minimally coupling the Hamiltonian with the gauge field. Finally, the Gauß constraints are imposed kinematically allowing for the initial (matter) degrees of freedom to be gauged away. Within our framework, these operations are simply accomplished by considering the $2\mathsf{Vec}_G$-module 2-category $\mathcal{M} \equiv \mathcal{M}(G, 1) \cong 2\mathsf{Vec}$. In particular, it follows form the definition that we have $\mathfrak{a}_{\mathfrak{g},\mathfrak{m}} = \mathfrak{g}$.

More generally, let us consider the gauging of the $A$ sub-symmetry of a $G$-symmetric Hamiltonian. As above, we begin by introducing an $A$-gauge field $\mathfrak{a} \in Z^1(\Sigma_\triangle, A)$ and impose the following Gauß constraints at every vertex:

$$\mathbb{G}_v := \frac{1}{|A|} \sum_{x \in A} \Big( \prod_{e \to v} R_e^x \Big) R_v^x \Big( \prod_{e \leftarrow v} L_e^x \Big) \stackrel{!}{=} \mathrm{id}, \tag{66}$$

where

$$R_e^x : |\mathfrak{a}[e]\rangle \mapsto |\mathfrak{a}[e] x^{-1}\rangle, \quad L_e^x : |\mathfrak{a}[e]\rangle \mapsto |x\mathfrak{a}[e]\rangle, \tag{67}$$

so that the physical Hilbert space does not have a tensor product structure anymore. In order to kinematically enforce these Gauß constraints, it is convenient to disentangle degrees of freedom by applying the following unitary:

$$\mathbb{U} := \prod_v \Big( \prod_{e \to v} cR_{v,e} \prod_{e \leftarrow v} cL_{v,e} \Big), \tag{68}$$

---

[10]Given a pair $(\mathfrak{m}[v_1], \mathfrak{m}[v_2])$ of simple objects in $\mathcal{M}(A, \lambda)$, there must exist $g \in G$ such that $\mathsf{Vec}_g \rhd \mathfrak{m}[v_2] \cong \mathfrak{m}[v_1]$. Let $g' \in G$ such that $\mathfrak{m}[v_1] \cong \mathsf{Vec}_{g'g} \rhd \mathfrak{m}[v_2] \cong \mathsf{Vec}_{g'} \rhd \mathfrak{m}[v_1]$. This requires $\mathsf{Vec}_{g'}$ to be in the stabiliser of $\mathfrak{m}[v_1]$, which in turn requires $g' \in r(\mathfrak{m}[v_1]) A r(\mathfrak{m}[v_1])^{-1}$. Our statement finally follows from $|r(\mathfrak{m}[v_1]) A r(\mathfrak{m}[v_1])^{-1}| = |A|$.

where we introduced the controlled group actions

$$
\begin{aligned}
cR_{\mathsf{v,e}} &: |g_1\rangle_{\mathsf{v}} \otimes |g_2\rangle_{\mathsf{e}} \mapsto |g_1\rangle_{\mathsf{v}} \otimes |g_2 g_1^{-1}\rangle_{\mathsf{e}}, \\
cL_{\mathsf{v,e}} &: |g_1\rangle_{\mathsf{v}} \otimes |g_2\rangle_{\mathsf{e}} \mapsto |g_1\rangle_{\mathsf{v}} \otimes |g_1 g_2\rangle_{\mathsf{e}}.
\end{aligned}
\tag{69}
$$

In particular, we have

$$
\mathbb{U}\mathbb{G}_{\mathsf{v}}\mathbb{U}^{\dagger} = \frac{1}{|A|} \sum_{x \in A} R_{\mathsf{v}}^{x} \overset{!}{=} \mathrm{id},
\tag{70}
$$

so that imposing the Gauß constraints amounts to considering an effective microscopic Hilbert space whereby degrees of freedom at vertices are labelled by elements in $G/A$, or rather simple objects $\mathfrak{m}[\mathsf{v}]$ in $\mathcal{M}(A,1)$. Notice finally that

$$
\begin{aligned}
cL_{\mathsf{v},(\mathsf{vv_1})}^{-1} (L_{\mathsf{v}}^{x} \otimes L_{(\mathsf{vv_1})}^{x}) \, cL_{\mathsf{v},(\mathsf{vv_1})} : |\mathfrak{m}[\mathsf{v}], \mathfrak{a}[\mathsf{vv_1}]\rangle &\mapsto |\mathbb{C}_x \triangleright \mathfrak{m}[\mathsf{v}], a_{x,\mathfrak{m}[\mathsf{v}]}\mathfrak{a}[\mathsf{vv_1}]\rangle \\
&\equiv |\mathbb{C}_x \triangleright \mathfrak{m}[\mathsf{v}], \mathfrak{a}[\mathsf{v'v_1}]\rangle, \\
cR_{\mathsf{v},(\mathsf{v_1 v})}^{-1} (L_{\mathsf{v}}^{x} \otimes R_{(\mathsf{v_1 v})}^{x}) \, cR_{\mathsf{v},(\mathsf{v_1 v})} : |\mathfrak{m}[\mathsf{v}], \mathfrak{a}[\mathsf{v_1 v}]\rangle &\mapsto |\mathbb{C}_x \triangleright \mathfrak{m}[\mathsf{v}], \mathfrak{a}[\mathsf{v_1 v}]a_{x^{-1}, \mathsf{Vec}_x \triangleright \mathfrak{m}[\mathsf{v}]}\rangle \\
&\equiv |\mathbb{C}_x \triangleright \mathfrak{m}[\mathsf{v}], \mathfrak{a}[\mathsf{v_1 v'}]\rangle,
\end{aligned}
$$

where we identified $x \equiv \mathfrak{g}[\mathsf{v'v}]$, at which point we recover the image of the operator $L_{\mathsf{v}}^{x}$ under the duality map, as encoded into local operators of the form (65) with $\mathcal{M} \equiv \mathcal{M}(A,1)$. Given this understanding of choosing the $2\mathsf{Vec}_G$-module 2-category $\mathcal{M}(A,1)$ as gauging the $A$ sub-symmetry, we interpret choosing $\mathcal{M}(A,\lambda)$ with $\lambda$ a non-trivial 3-cocycle in $H^3(A, \mathrm{U}(1))$ as performing a $\lambda$-twisted gauging of the $A$ sub-symmetry. More details on this gauging perspective are provided in sec. 3.8 for the case of the finite group generalisation of the transverse-field Ising model.

## 3.5 Dual symmetries

We commented earlier that dualities considered in this manuscript map local symmetric operators to dual local symmetric operators. However, we have not yet revealed what the symmetry of a given Hamiltonian $\mathbb{H}^{\mathcal{M}}$ is. Notice that we are still not choosing any specific Hamiltonian, it is enough to know that it is defined in terms of local operators of the form (52).

Recall that we introduced in sec. 3.3 the notion of $2\mathsf{Vec}_G$-module 2-functors and explained how these provide duality operators between local operators that only differ in a choice of $2\mathsf{Vec}_G$-module category. Given a Hamiltonian $\mathbb{H}^{\mathcal{M}} = \sum_{\mathsf{v}} \sum_n \mathbb{h}_{\mathsf{v},n}^{\mathcal{M}}$, a module 2-functor from $\mathcal{M}$ to itself should thus correspond to a symmetry operator of the model. Indeed, we shall demonstrate that $2\mathsf{Vec}_G$-module 2-endofunctors of $\mathcal{M}$ label surface symmetry operators. Furthermore, these surface operators can host topological line operators. More generally, surface operators are not necessarily closed, in which case topological lines living at the junctions of distinct topological surfaces are required so the Hamiltonian is left invariant. Mathematically, these topological lines are captured by the notion of module natural 2-transformation between module 2-functors: Given a pair of left $2\mathsf{Vec}_G$-module 2-functors $(F, \omega, \Omega)$ and $(\tilde{F}, \tilde{\omega}, \tilde{\Omega})$, we define a $2\mathsf{Vec}_G$-module natural 2-transformation between them as a tuple $(\theta, \Theta)$ consisting of a natural 2-transformation $\theta : F \Rightarrow \tilde{F}$ satisfying a coherence axiom up to an invertible modification $\Theta$ with components $\Theta_{\mathsf{Vec}_g, M}$ defined according to the string diagram

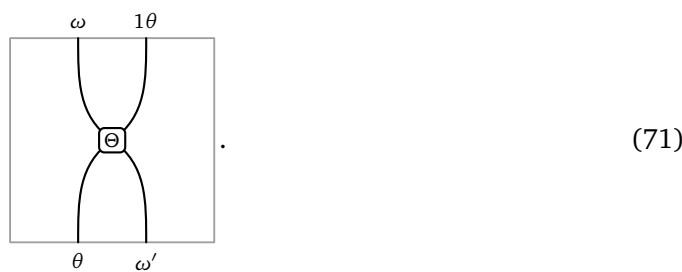

$$
\tag{71}
$$

This invertible modification is required to satisfy a coherence axiom encoded into the following equality of string diagrams:

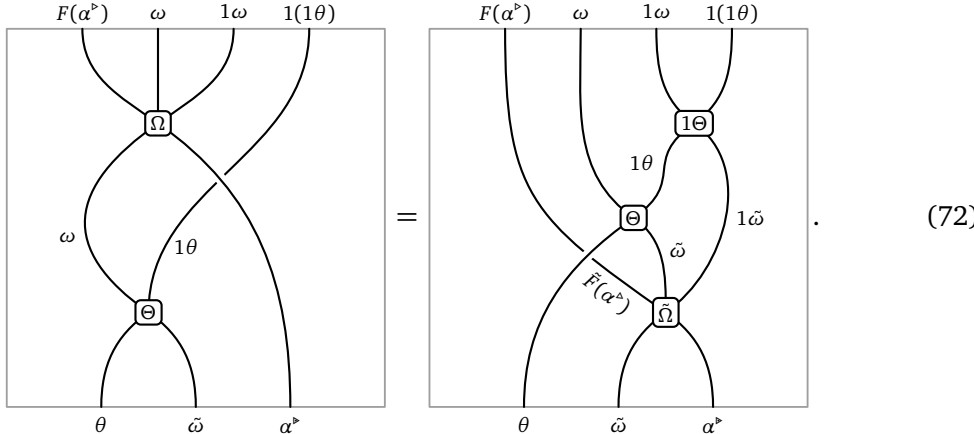

$$\tag{72}$$

Furthermore, given a pair of $2\mathsf{Vec}_G$-module natural 2-transformations $(\theta, \Theta)$ and $(\tilde{\theta}, \tilde{\Theta})$, we can define a $2\mathsf{Vec}_G$-module modification from $(\theta, \Theta)$ to $(\tilde{\theta}, \tilde{\Theta})$ as a modification $\vartheta : \theta \Rightarrow \tilde{\theta}$ such that

$$\tilde{\Theta}_{\mathsf{Vec}_g, M} \circ (1_{1_{\mathsf{Vec}_g}} \triangleright \vartheta_M) = \vartheta_{\mathsf{Vec}_g \triangleright M} \circ \Theta_{\mathsf{Vec}_g, M}, \tag{73}$$

for all $g \in G$ and $M \in \mathcal{M}$.

Given a pair $(\mathcal{M}, \mathcal{M}')$ of $2\mathsf{Vec}_G$-module 2-categories, we shall refer to $2\mathsf{Fun}_{2\mathsf{Vec}_G}(\mathcal{M}, \mathcal{M}')$ as the 2-category whose objects are $2\mathsf{Vec}_G$-module 2-functors $\mathcal{M} \to \mathcal{M}'$, 1-morphisms are $2\mathsf{Vec}_G$-module 2-natural transformations, and 2-morphisms are $2\mathsf{Vec}_G$-module modifications. In the present context, we are specifically interested in 2-categories of the form $(2\mathsf{Vec}_G)^\star_{\mathcal{M}(A,\lambda)} := 2\mathsf{Fun}_{2\mathsf{Vec}_G}(\mathcal{M}(A,\lambda), \mathcal{M}(A,\lambda))$ that shall be referred to as 'Morita duals' of $2\mathsf{Vec}_G$ with respect to $\mathcal{M}(A,\lambda)$. Crucially, these inherit a fusion structure from the composition of module 2-functors. We shall now demonstrate that for any $2\mathsf{Vec}_G$-module category $\mathcal{M} \equiv \mathcal{M}(A,\lambda)$, the Hamiltonian $\mathbb{H}^{\mathcal{M}}$ is left invariant by topological operators organised into the Morita dual 2-category $(2\mathsf{Vec}_G)^\star_{\mathcal{M}}$.

Let us begin by constructing topological surface operators labelled by simple objects in the fusion 2-category $(2\mathsf{Vec}_G)^\star_{\mathcal{M}}$. Given the complex $\bigotimes_v \times \mathbb{I}$ depicted in eq. (56) and a simple object $(F, \omega, \Omega)$ in $(2\mathsf{Vec}_G)^\star_{\mathcal{M}}$, we consider the following assignment of degrees of freedom: First, we assign as before the same gauge field $\mathfrak{g} \in Z^1(\bigotimes_v, G)$ to $\bigotimes_v^0$ and $\bigotimes_v^1$. We then consider an assignment $\mathfrak{m}$ of simple objects $\mathfrak{m}[v_1], \mathfrak{m}[v_1'] \in \mathcal{M}$ to every vertex $v_1 \subset \bigotimes_v^0$ and $v_1' \subset \bigotimes_v^1$ such that $\mathfrak{m}[v_1] = \mathsf{Vec}_{\mathfrak{g}[v_1 v_2]} \triangleright \mathfrak{m}[v_2]$ for every $(v_1 v_2) \in \bigotimes_v^0$, $\mathfrak{m}[v_1'] = \mathsf{Vec}_{\mathfrak{g}[v_1' v_2']} \triangleright \mathfrak{m}[v_2']$ for every $(v_1' v_2') \in \bigotimes_v^1$. We notate via $C_{\mathfrak{g}}^0(\bigotimes_v \times \mathbb{I}, \mathcal{M})$ the collection of assignments $\mathfrak{m}$ fulfilling these conditions. Every edge of the form $(v_1' v_1) \subset \bigotimes_v \times \mathbb{I}$ is further allocated a simple 1-morphism $\mathfrak{f}[v_1' v_1]$ in the (possibly terminal) hom-category $\mathsf{Hom}_{\mathcal{M}}(F(\mathfrak{m}[v_1]), \mathfrak{m}[v_1'])$. Given any prism $(v_1 v_2 v_3) \times \mathbb{I} \subset \bigotimes_v \times \mathbb{I}$, every plaquette $(v_1 v_2) \times \mathbb{I} \equiv (v_1' v_2' v_1 v_2)$ is labelled by a basis vector $\mathfrak{f}[v_1' v_2' v_1 v_2]$ in the vector space $V^{\epsilon(v_1' v_2' v_1 v_2)}\big((\mathfrak{g}, \mathfrak{m}, \mathfrak{f})[v_1' v_2' v_1 v_2]\big)$ given by

$$
\begin{aligned}
&V^+\big((\mathfrak{g}, \mathfrak{m}, \mathfrak{f})[v_1' v_2' v_1 v_2]\big) \\
&\quad := \mathsf{Hom}_{\mathcal{M}}\big(1_{\mathfrak{m}[v_1']} \circ (\mathsf{Vec}_{\mathfrak{g}[v_1 v_2]} \triangleright \mathfrak{f}[v_2' v_2]) \circ \omega_{\mathsf{Vec}_{\mathfrak{g}[v_1 v_2]}, \mathfrak{m}[v_2]}, \mathfrak{f}[v_1' v_1] \circ F(1_{\mathfrak{m}[v_1]})\big), \\
&V^-\big((\mathfrak{g}, \mathfrak{m}, \mathfrak{f})[v_1' v_2' v_1 v_2]\big) \\
&\quad := \mathsf{Hom}_{\mathcal{M}}\big(\mathfrak{f}[v_1' v_1] \circ F(1_{\mathfrak{m}[v_1]}), 1_{\mathfrak{m}[v_1']} \circ (\mathsf{Vec}_{\mathfrak{g}[v_1 v_2]} \triangleright \mathfrak{f}[v_2' v_2]) \circ \omega_{\mathsf{Vec}_{\mathfrak{g}[v_1 v_2]}, \mathfrak{m}[v_2]}\big),
\end{aligned}
\tag{74}
$$

where $\epsilon(v_1' v_2' v_1 v_2) = \pm 1$ depends on the orientation of $(v_1 v_2)$ relative to that of $(v_1 v_2 v_3)$. For

convenience, we summarise these various notations below:

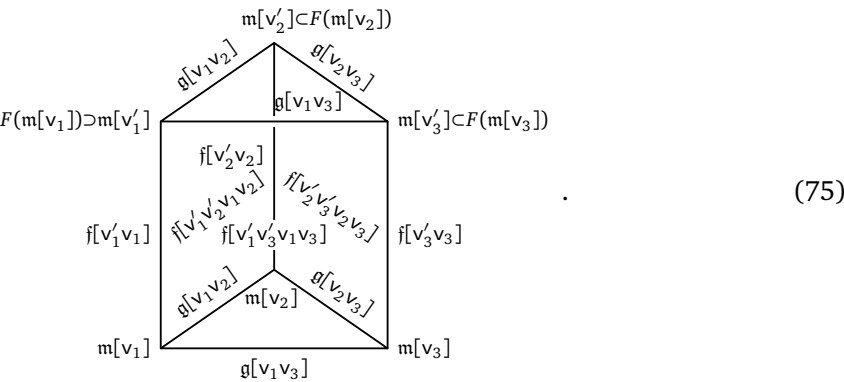

$$(75)$$

Given assignments $(\mathfrak{g}, \mathfrak{m}, \mathfrak{f})$ described above, we finally associate to the prism $(v_1v_2v_3) \times \mathbb{I}$ the symbol $\Omega^{\epsilon(v_1v_2v_3)}\big((\mathfrak{g}, \mathfrak{m}, \mathfrak{f})[(v_1v_2v_3) \times \mathbb{I}]\big)$ defined as the matrix entry corresponding to the vectors $\mathfrak{f}[v'_1v'_2v_1v_2]$, $\mathfrak{f}[v'_2v'_3v_2v_3]$ and $\mathfrak{f}[v'_1v'_3v_1v_3]$ of the map

$$V^+\big((\mathfrak{g}, \mathfrak{m}, \mathfrak{f})[v'_1v'_2v_1v_2]\big) \otimes V^+\big((\mathfrak{g}, \mathfrak{m}, \mathfrak{f})[v'_2v'_3v_2v_3]\big) \xrightarrow{\sim} V^+\big((\mathfrak{g}, \mathfrak{m}, \mathfrak{f})[v'_1v'_3v_1v_3]\big), \qquad (76)$$

that is determined by the component $\Omega_{\mathsf{Vec}_{\mathfrak{g}[v_1v_2]}, \mathsf{Vec}_{\mathfrak{g}[v_2v_3]}, \mathfrak{m}[v_3]}$ of the invertible modification $\Omega$. By convention, we set the symbols of this map to vanish whenever assignment $\mathfrak{f}$ of 1-morphisms and basis vectors in $\mathcal{M}$ is such that one of hom-categories associated with edges of the form $(v'_1v_1)$ is the terminal category or one of the vector spaces associated with plaquettes of the form $(v_1v_2) \times \mathbb{I}$ is the zero vector space.

Applying the rules described above, we find that the associahedron axiom (55) yields an equation in terms of the symbols defined above that graphically translates as eq. (61). Summing over all possible labels associated with simplices shared by several prisms $\bigotimes_v \times \mathbb{I}$ that fulfil all the rules described above, as well as tracing over basis vectors associated with plaquettes shared by two prisms via the canonical pairing

$$V^-\big((\mathfrak{g}, \mathfrak{m}, \mathfrak{f})[v'_1v'_2v_1v_2]\big) \otimes V^+\big((\mathfrak{g}, \mathfrak{m}, \mathfrak{f})[v'_1v'_2v_1v_2]\big) \to \mathbb{C}, \qquad (77)$$

yields an operator that commutes with $\mathbb{h}_{v,n}^{\mathcal{M}}$. In order to construct the surface operator commuting with the whole Hamiltonian $\mathbb{H}^{\mathcal{M}}$, it suffices to extend the previous construction to the whole $\Sigma_{\triangle}$, thereby summing over all simple objects and simple 1-morphisms, and tracing over all basis basis vectors associated with plaquettes:

$$\sum_{\substack{\mathfrak{g} \in Z^1(\bigotimes_v^0, G) \\ \mathfrak{m} \in C_{\mathfrak{g}}^0(\bigotimes_v \times \mathbb{I}, \mathcal{M}) \\ \mathfrak{f}}} \bigg(\prod_{(v_1v_2v_3) \subset \Sigma_{\triangle}} \Omega^{\epsilon(v_1v_2v_3)}\big((\mathfrak{g}, \mathfrak{m}, \mathfrak{f})[(v_1v_2v_3) \times \mathbb{I}]\big)\bigg)\big|(\mathfrak{g}, \mathfrak{m})\big[\Sigma_{\triangle}^1\big]\big\rangle\big\langle(\mathfrak{g}, \mathfrak{m})\big[\Sigma_{\triangle}^0\big]\big|. \quad (78)$$

It follows from the construction that fusion of surface operators is provided by the composition of the corresponding module 2-endofunctors.

Given the above, let us now consider line operators at the junction of two surface operators. In many ways, the derivation is merely a lower-dimensional analogue of that above. Given a pair of simple objects $F \equiv (F, \omega, \Omega)$ and $\tilde{F} \equiv (\tilde{F}, \tilde{\omega}, \tilde{\Omega})$ in $(2\mathsf{Vec}_G)_{\mathcal{M}}^{\star}$, let $(\theta, \Theta)$ be a simple 1-morphism in $(2\mathsf{Vec}_G)_{\mathcal{M}}^{\star}$ between them. As previously, we shall define the line operators by means of a labelled complex. Given a cube $(\tilde{v}_1\tilde{v}_2v_1v_2) \times \mathbb{I}$, we consider an assignment of degrees of freedom that resembles that of the prisms: We assign the same group variable $\mathfrak{g}[v_1v_2]$ to edges $(v_1v_2)$, $(v'_1v'_2)$,

$(\tilde{v}_1\tilde{v}_2)$ and $(\tilde{v}'_1\tilde{v}'_2)$, as well as simple objects $\mathfrak{m}[v_1], \mathfrak{m}[v'_1], \mathfrak{m}[\tilde{v}_1], \mathfrak{m}[\tilde{v}'_1] \in \mathcal{M}$ to vertices of the cube such that $\mathfrak{m}[\tilde{v}_1] = \mathfrak{m}[v_1]$, $\mathfrak{m}[\tilde{v}'_1] = \mathfrak{m}[v'_1]$, $\mathfrak{m}[v_1] = \mathsf{Vec}_{\mathfrak{g}[v_1v_2]} \triangleright \mathfrak{m}[v_2]$, $\mathfrak{m}[v'_1] = \mathsf{Vec}_{\mathfrak{g}[v_1v_2]} \triangleright \mathfrak{m}[v'_2]$. We further allocate to the corresponding edges simple 1-morphisms $\mathfrak{f}[v'_1v_1]$ and $\tilde{\mathfrak{f}}[\tilde{v}'_1\tilde{v}_1]$ in the (possibly terminal) hom-categories $\mathsf{Hom}_{\mathcal{M}}(F(\mathfrak{m}[v_1]), \mathfrak{m}[v'_1])$ and $\in \mathsf{Hom}_{\mathcal{M}}(\tilde{F}(\mathfrak{m}[\tilde{v}_1]), \mathfrak{m}[\tilde{v}'_1])$, respectively. Plaquettes $(v'_1v'_2v_1v_2)$ and $(\tilde{v}'_1\tilde{v}'_2\tilde{v}_1\tilde{v}_2)$ are labelled by basis vectors $\mathfrak{f}[v'_1v'_2v_1v_2]$ and $\tilde{\mathfrak{f}}[\tilde{v}'_1\tilde{v}'_2\tilde{v}_1\tilde{v}_2]$ in (possibly zero) vector spaces $V^{\epsilon(v'_1v'_2v_1v_2)}\big((\mathfrak{g},\mathfrak{m},\mathfrak{f})[v'_1v'_2v_1v_2]\big)$ and $V^{\epsilon(\tilde{v}'_1\tilde{v}'_2\tilde{v}_1\tilde{v}_2)}\big((\mathfrak{g},\mathfrak{m},\tilde{\mathfrak{f}})[\tilde{v}'_1\tilde{v}'_2\tilde{v}_1\tilde{v}_2]\big)$ as defined in eq. (74), respectively, whereas plaquettes $(\tilde{v}'_1\tilde{v}_1v'_1v_1)$ are labelled by basis vectors $\mathfrak{s}[\tilde{v}'_1\tilde{v}_1v'_1v_1]$ in vector spaces $V^{\epsilon(\tilde{v}'_1\tilde{v}_1v'_1v_1)}\big((\mathfrak{m},\mathfrak{f},\tilde{\mathfrak{f}},\mathfrak{s})[\tilde{v}'_1\tilde{v}_1v'_1v_1]\big)$ given by

$$
\begin{aligned}
V^+\big((\mathfrak{m},\mathfrak{f},\tilde{\mathfrak{f}},\mathfrak{s})[\tilde{v}'_1\tilde{v}_1v'_1v_1]\big) &:= \mathsf{Hom}_{\mathcal{M}}\big(\tilde{\mathfrak{f}}[\tilde{v}'_1\tilde{v}_1] \circ \theta_{\mathfrak{m}[v_1]}, \mathfrak{f}[v'_1v_1]\big), \\
V^-\big((\mathfrak{m},\mathfrak{f},\tilde{\mathfrak{f}},\mathfrak{s})[\tilde{v}'_1\tilde{v}_1v'_1v_1]\big) &:= \mathsf{Hom}_{\mathcal{M}}\big(\mathfrak{f}[v'_1v_1], \tilde{\mathfrak{f}}[\tilde{v}'_1\tilde{v}_1] \circ \theta_{\mathfrak{m}[v_1]}\big).
\end{aligned}
\tag{79}
$$

As before, let us summarise our notations via the following diagram:

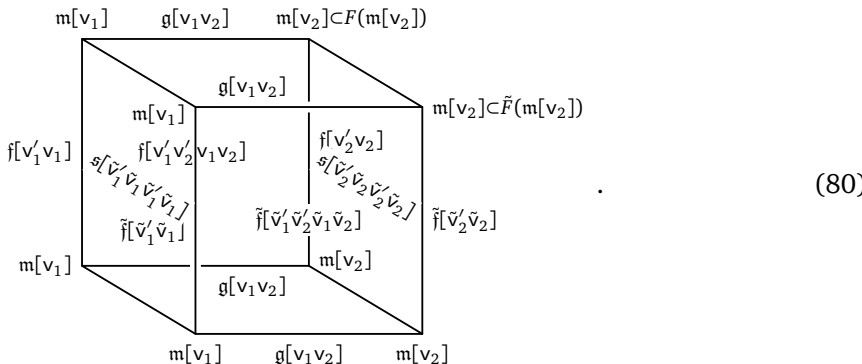

$$\tag{80}$$

We finally associate to the cube $(\tilde{v}_1\tilde{v}_2v_1v_2)\times\mathbb{I}$ the symbol $\Theta^{\epsilon(\tilde{v}_1\tilde{v}_2v_1v_2)}\big((\mathfrak{g},\mathfrak{m},\mathfrak{f},\tilde{\mathfrak{f}},\mathfrak{s})[(v_1v_2v_3)\times\mathbb{I}]\big)$ corresponding to the vectors $\mathfrak{f}[v'_1v'_2v_1v_2]$, $\tilde{\mathfrak{f}}[\tilde{v}'_1\tilde{v}'_2\tilde{v}_1\tilde{v}_2]$, $\mathfrak{s}(\tilde{v}'_1\tilde{v}_1v'_1v_1)$ and $\mathfrak{s}(\tilde{v}'_2\tilde{v}_2v'_2v_2)$ of the map

$$
\begin{aligned}
V^+\big((\mathfrak{g},\mathfrak{m},\mathfrak{f})[v'_1v'_2v_1v_2]\big) &\otimes V^+\big((\mathfrak{m},\mathfrak{f},\tilde{\mathfrak{f}},\mathfrak{s})[\tilde{v}'_1\tilde{v}_1v'_1v_1]\big) \\
&\xrightarrow{\sim} V^+\big((\mathfrak{m},\mathfrak{f},\tilde{\mathfrak{f}},\mathfrak{s})[\tilde{v}'_2\tilde{v}_2v'_2v_2]\big) \otimes V^+\big((\mathfrak{g},\mathfrak{m},\tilde{\mathfrak{f}})[\tilde{v}'_1\tilde{v}'_2\tilde{v}_1\tilde{v}_2]\big),
\end{aligned}
\tag{81}
$$

that is determined by the component $\Theta_{\mathsf{Vec}_{\mathfrak{g}[v_1v_2]},\mathfrak{m}[v_2]}$ of the invertible modification $\Theta$. By convention, we set matrix entries of this map to vanish whenever one of the hom-categories associated with edges of the form $(v'_1v_1)$ or $(\tilde{v}'_1\tilde{v}_1)$ is the terminal category, or one of the hom-spaces associated with plaquettes $(\tilde{v}'_1\tilde{v}_1v'_1v_1)$ is the zero vector space. Applying all the rules introduced above to the following complexes

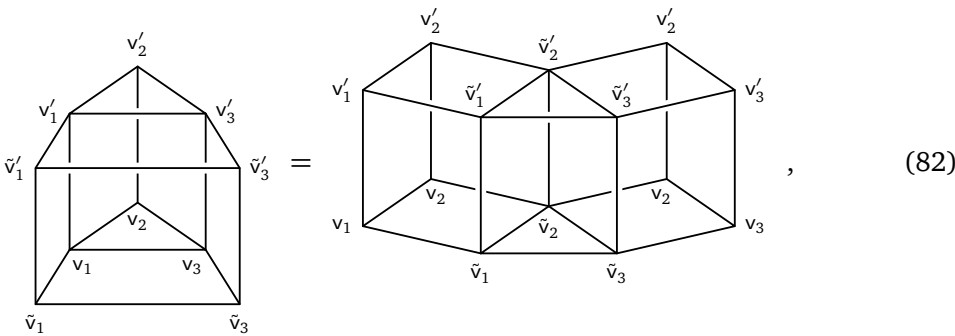

$$\tag{82}$$

yields an equation in terms of symbols of $\Omega_{\mathsf{Vec}_{\mathfrak{g}[v_1v_2]},\mathsf{Vec}_{\mathfrak{g}[v_2v_3]},\mathfrak{m}[v_3]}$, $\Theta_{\mathsf{Vec}_{\mathfrak{g}[v_1v_3]},\mathfrak{m}[v_3]}$ on the l.h.s. and $\tilde{\Omega}_{\mathsf{Vec}_{\mathfrak{g}[v_1v_2]},\mathsf{Vec}_{\mathfrak{g}[v_2v_3]},\mathfrak{m}[v_3]}$, $\Theta_{\mathsf{Vec}_{\mathfrak{g}[v_1v_2]},\mathsf{Vec}_{\mathfrak{g}[v_2v_3]}\triangleright\mathfrak{m}[v_3]}$, $\Theta_{\mathsf{Vec}_{\mathfrak{g}[v_2v_3]},\mathfrak{m}[v_3]}$ on the r.h.s., where as before we trace over basis vectors associated with plaquettes shared by complexes. This equation

is guaranteed by the coherence axiom (72). Concretely, this equation means that a simple object in $\mathsf{Hom}_{(2\mathsf{Vec}_G)^\star_{\mathcal{M}}}(F,\tilde{F})$ defines a topological line operator at the interface of two surface operators labelled by simple objects $(F,\omega,\Omega)$ and $(\tilde{F},\tilde{\omega},\tilde{\Omega})$ in $(2\mathsf{Vec}_G)^\star_{\mathcal{M}}$. Vertical composition of $2\mathsf{Vec}_G$-module natural 2-transformations in $(2\mathsf{Vec}_G)^\star_{\mathcal{M}}$ finally provides the fusion of topological lines.

## 3.6 Higher representation theory

We demonstrated above that symmetry operators of Hamiltonians $\mathbb{H}^{\mathcal{M}}$ with $\mathcal{M} \equiv \mathcal{M}(A,\lambda)$ form the fusion 2-category $(2\mathsf{Vec}_G)^\star_{\mathcal{M}}$. Concretely, this means that starting from a $G$-symmetric Hamiltonian that has been rewritten in terms of local operators (38), simply replacing the implicit choice of $2\mathsf{Vec}_G$-module 2-category $2\mathsf{Vec}_G$ by any other indecomposable $2\mathsf{Vec}_G$-module 2-category $\mathcal{M}(A,\lambda)$ yields a dual model with a fusion 2-categorical $(2\mathsf{Vec}_G)^\star_{\mathcal{M}}$ symmetry in virtue of the demonstration above. Within this context, identifying the symmetry of a dual model thus boils down to computing the Morita dual fusion 2-category $(2\mathsf{Vec}_G)^\star_{\mathcal{M}}$ of $2\mathsf{Vec}_G$ with respect to $\mathcal{M}$. Interestingly—and to some extent this is the purpose of the following sections—knowing from the general demonstration that a given model possesses a $(2\mathsf{Vec}_G)^\star_{\mathcal{M}}$ symmetry does not mean it is easily verifiable in terms of explicit lattice operators.[11] We shall explicitly compute Morita duals in sec. 3.7 but, in order to understand the resulting fusion 2-categories, we first need to discuss *higher representation theory*.

We set the stage with a review of the category theoretic viewpoint on (ordinary) representation theory. Given a finite group $G$, we denote by $[G,\bullet]$ the 1-groupoid with object-set $G$ and no non-trivial morphisms. Let us consider the category $\mathsf{Fun}([G,\bullet],\mathsf{Vec})$ of functors from $[G,\bullet]$ to the category $\mathsf{Vec}$. By definition, an object $V$ in $\mathsf{Fun}([G,\bullet],\mathsf{Vec})$ assigns to every $g \in G$ a vector space $V_g$ in $\mathsf{Vec}$, and thus amounts to a $G$-graded vector space of the form $V = \bigoplus_{g \in G} V_g$. Natural transformations in $\mathsf{Fun}([G,\bullet],\mathsf{Vec})$ then correspond to grading preserving linear maps. The convolution product of functors $[G,\bullet] \to \mathsf{Vec}$, which descends from the multiplication rule of $G$, further endows $\mathsf{Fun}([G,\bullet],\mathsf{Vec})$ with the structure of a fusion (1-)category according to

$$(V \odot W)_g := \bigoplus_{x \in G} V_x \otimes W_{x^{-1}g}\,, \tag{83}$$

with unit $\mathbb{1}$ satisfying $\mathbb{1}_g = \delta_{g,\mathbb{1}_G}\mathbb{C}$. Henceforth, we denote this fusion category by $\mathsf{Vec}_G$. There are $|G|$-many simple objects in $\mathsf{Vec}_G$ provided by the one-dimensional vector spaces $\mathbb{C}_g$, for every $g \in G$, such that $\mathbb{C}_g \odot \mathbb{C}_h \simeq \mathbb{C}_{gh}$ and $\mathsf{Hom}_{\mathsf{Vec}_G}(\mathbb{C}_g,\mathbb{C}_h) \simeq \delta_{g,h}\mathbb{C}$. Notice that $\mathsf{Vec}_G$ can be equivalently defined as the fusion category $\mathsf{Mod}(\mathbb{C}^G)$ of modules over the algebra $\mathbb{C}^G$ of functions on $G$. Fusion category $\mathsf{Vec}_G$ is merely the lower categorical analogue of the fusion 2-category $2\mathsf{Vec}_G$ we have been considering. More specifically, we shall think of $2\mathsf{Vec}_G$ as a categorification of $\mathsf{Vec}_G$.

Another way to treat a finite group $G$ as a 1-category is to consider the *delooping* of $G$ defined as the 1-groupoid $[\bullet,G]$ with a single object $\bullet$ and $\mathsf{Hom}_{[\bullet,G]}(\bullet,\bullet) = G$ such that the composition of morphisms is given by the multiplication rule of $G$. As before, we can consider the category $\mathsf{Fun}([\bullet,G],\mathsf{Vec})$ of functors $[\bullet,G] \to \mathsf{Vec}$. By definition, an object $\rho$ in $\mathsf{Fun}([\bullet,G],\mathsf{Vec})$ assigns to the unique object $\bullet$ a vector space $V := \rho(\bullet)$, and to every morphism $g : \bullet \to \bullet$ a linear map $\rho(g) : V \to V$ fulfilling $\rho(g) \circ \rho(h) = \rho(gh)$ for every $g,h \in G$. In other words, $\rho$ is a representation of $G$. It follows that natural transformations in $\mathsf{Fun}([\bullet,G],\mathsf{Vec})$ correspond to intertwiners. The symmetric monoidal structure of $\mathsf{Vec}$ endows $\mathsf{Fun}([\bullet,G],\mathsf{Vec})$

---

[11]Already in (1+1)d systems, it is easy to construct models with $\mathsf{Rep}(G)$-symmetries for instance, which are very tedious to confirm without a systematic approach analogous to the one employed in this manuscript [90].

with the structure of a fusion 1-category according to

$$
\begin{aligned}
(\rho \odot \varrho)(\bullet) &:= \rho(\bullet) \otimes \varrho(\bullet) \equiv V \otimes W, \\
(\rho \odot \varrho)(g) &:= \rho(g) \otimes \varrho(g) \in \mathrm{End}(V \otimes W),
\end{aligned}
\tag{84}
$$

where the tensor products on the r.h.s. are that in Vec. Henceforth, we denote by $\mathrm{Rep}(G)$ this fusion category. Note that the simple objects are provided by the irreducible representations of $G$. Given the equivalence between representations of $G$ and modules over the group algebra $\mathbb{C}[G]$, we also have $\mathrm{Rep}(G) \cong \mathrm{Mod}(\mathbb{C}[G])$.

In the following, we shall find that Morita duals of $2\mathrm{Vec}_G$ are often related to 'higher' notions of group representation obtained by following the ethos categorification. We remarked above that a group representation is equivalent to a module over the group algebra. Let us adopt this viewpoint and categorify it. Recall that $\mathbb{C}[G]$ is the *associative* algebra whose elements are given by formal linear combinations of group elements over $\mathbb{C}$ and multiplication rule descends from that of the group. Loosely speaking, categorifying the notion of group algebra requires in particular to loosen the associativity condition so that it only holds up to isomorphisms [11]. But this would be inconsistent with having coefficients valued in $\mathbb{C}$. One solution is to consider instead coefficients valued in Vec, where we should think of Vec as being a categorification of $\mathbb{C}$. The result is the fusion category $\mathrm{Vec}_G$, whose definition was reviewed above, thought as a group '2-algebra'. This incites us to consider a notion of '2-representation' of a group $G$ as a module category over $\mathrm{Vec}_G$. We shall refine this notion of 2-representation later, but let us accept it for the moment and proceed.

The notion of $\mathrm{Vec}_G$-module category is defined in close analogy with that of $2\mathrm{Vec}_G$-module 2-category reviewed in sec. 3.1. Concretely, a (left) $\mathrm{Vec}_G$-module category can be defined as a triple $(\mathcal{N}, \triangleright, \alpha^\triangleright)$ consisting of a ($\mathbb{C}$-linear finite semisimple) category $\mathcal{N}$, a binary action functor $\triangleright : \mathrm{Vec}_G \times \mathcal{N} \to \mathcal{N}$ and a natural isomorphism $\alpha^\triangleright : (- \odot -) \triangleright - \xrightarrow{\sim} - \triangleright (- \triangleright -)$ referred to as the left *module associator*, which is required to satisfy a 'pentagon axiom' akin to (42).[12] Indecomposable module categories over $\mathrm{Vec}_G$ are obtained following the same recipe as in the $2\mathrm{Vec}_G$ case: Given a subgroup $B \subseteq G$ and a normalised 2-cocycle $\psi$ in $H^2(B, \mathrm{U}(1))$, let $\mathcal{N}(B, \psi)$ be a category with object-set the set $G/B$ of left cosets. A left $\mathrm{Vec}_G$-module structure can be defined on $\mathcal{N}(B, \psi)$ via $\mathbb{C}_g \triangleright N := (g\,\mathrm{r}(N))B$ for any $g \in G$ and $N \in G/B$, where $\mathrm{r} : G/B \to G$ assigns to every left coset its representative element in $G$. As before, we notate via $b_{g,N}$ the group element in $B$ satisfying $g\,\mathrm{r}(N) = \mathrm{r}(\mathbb{C}_g \triangleright N)b_{g,N}$. Associativity of the multiplication in $G$ imposes $b_{g_1 g_2, N} = b_{g_1, \mathbb{C}_{g_2} \triangleright N}\, b_{g_2, N}$, for every $g_1, g_2 \in G$ and $N \in G/B$, in exact analogy with eq. (46). Thinking of the Abelian group $\mathrm{Hom}(G/B, \mathrm{U}(1))$ as a left $G$-module, let us consider the 2-cochain $\alpha^\triangleright \in C^2(G, \mathrm{Hom}(G/A, \mathrm{U}(1)))$ defined as $\alpha^\triangleright(g_1, g_2)(N) := \psi(b_{g_1, \mathbb{C}_{g_2} \triangleright N}, b_{g_2, N})$ for any $g_1, g_2 \in G$ and $N \in G/B$. In virtue of the cocycle condition $\mathrm{d}\psi = 1$ and the equation above, we have

$$
\alpha^\triangleright(g_2, g_3)(N)\,\alpha^\triangleright(g_1, g_2 g_3)(N) = \alpha^\triangleright(g_1 g_2, g_3)(N)\,\alpha^\triangleright(g_1, g_2)(\mathbb{C}_{g_3} \triangleright N),
\tag{85}
$$

for every $g_1, g_2, g_3 \in G$ and $N \in G/B$, so that $\alpha^\triangleright$ is a $\mathrm{Hom}(G/B, \mathrm{U}(1))$-valued 2-cocycle of $G$. Defining the natural isomorphism $\alpha^\triangleright$ with components

$$
\alpha^\triangleright_{\mathbb{C}_{g_1}, \mathbb{C}_{g_2}, N} := \alpha^\triangleright(g_1, g_2)(N) \cdot 1_{(g_1 g_2 \mathrm{r}(N))B} : (\mathbb{C}_{g_1} \odot \mathbb{C}_{g_2}) \triangleright N \xrightarrow{\sim} \mathbb{C}_{g_1} \triangleright (\mathbb{C}_{g_2} \triangleright N),
$$

for every $g_1, g_2 \in G$ and $N \in G/B$, we find that the triple $(\mathcal{N}(B, \psi), \triangleright, \alpha^\triangleright)$ does define a left $\mathrm{Vec}_G$-module category. It is a result of Ostrik that all indecomposable module categories over $\mathrm{Vec}_G$ are of this form [102, 103].

---

[12]Since we shall encounter both $\mathrm{Vec}_G$-module categories and $2\mathrm{Vec}_G$-module 2-categories at the same time in the following, we notate them via $\mathcal{N}$ and $\mathcal{M}$, respectively, to facilitate the distinction.

If $\mathsf{Vec}_G$-module categories admit an interpretation as 2-representations of the group $G$, then $\mathsf{Vec}_G$-module functors should be understood as 1-intertwiners between them. As previously, the notion of $\mathsf{Vec}_G$-module functor is defined in immediate analogy with that of $2\mathsf{Vec}_G$-module 2-functor. Concretely, given a pair of (left) $\mathsf{Vec}_G$-module categories $(\mathcal{N}, \rhd, \alpha^{\rhd})$ and $(\mathcal{N}', \rhd, \alpha^{\rhd})$, we define a *module functor* between them as a pair $(F, \omega)$ consisting of a functor $F : \mathcal{N} \to \mathcal{N}'$ and a natural isomorphism $\omega : F(- \rhd -) \xrightarrow{\sim} - \rhd F(-)$, which is required to satisfy a pentagon axiom involving both $\alpha^{\rhd}$ and $\alpha^{\rhd}$ akin to (53). There is also a notion of map between module functors, which within our context shall be interpreted as '2-intertwiners'. More precisely, given a pair of left $\mathsf{Vec}_G$-module functors $(F, \omega)$ and $(\tilde{F}, \tilde{\omega})$, we define a *module natural transformation* between them as a natural transformation $\theta : F \Rightarrow \tilde{F}$ satisfying $(1_{\mathbb{C}_g} \rhd \theta_M) \circ \omega_{\mathbb{C}_g, M} = \tilde{\omega}_{\mathbb{C}_g, M} \circ \theta_{\mathbb{C}_g \rhd M}$ for all $g \in G$ and $M \in \mathcal{M}$.

It follows from the definitions that, given a pair $(\mathcal{N}, \mathcal{N}')$ of $\mathsf{Vec}_G$-module categories, 1- and 2-intertwiners form a category that we denote by $\mathsf{Fun}_{\mathsf{Vec}_G}(\mathcal{N}, \mathcal{N}')$. Special attention is paid to Morita dual fusion categories $(\mathsf{Vec}_G)^{\star}_{\mathcal{N}(B, \psi)} := \mathsf{Fun}_{\mathsf{Vec}_G}(\mathcal{N}(B, \psi), \mathcal{N}(B, \psi))$ of $\mathsf{Vec}_G$ with respect to $\mathcal{N}(B, \psi)$ [55, 98]. In particular, these categories inherit a fusion structure from the composition of module functors [58], so that considering module endofunctors of indecomposable module categories is a way to construct new fusion categories. Treating $\mathsf{Vec}_G$ as a module category over itself, we have for instance $(\mathsf{Vec}_G)^{\star}_{\mathsf{Vec}_G} \cong \mathsf{Vec}_G$. Since the composition of module functors is a well-defined operation, we can further consider the 2-category $\mathsf{Mod}(\mathsf{Vec}_G)$ consisting of (left) $\mathsf{Vec}_G$-module categories and hom-categories of $\mathsf{Vec}_G$-module functors. This is another example of a *fusion 2-category*, still in the sense of Douglas and Reutter [47], where the fusion structure is obtained by defining a $\mathsf{Vec}_G$-module structure on $\mathcal{N} \boxtimes \mathcal{N}'$ via $\mathbb{C}_g \rhd (N \boxtimes N') := (\mathbb{C}_g \rhd N) \boxtimes (\mathbb{C}_g \rhd N')$ for every $g \in G$, $N \in \mathcal{N}$ and $N' \in \mathcal{N}'$ [73]. Explicit formulae for the fusion of indecomposable $\mathsf{Vec}_G$-module categories $\mathcal{N}(B, \psi)$ can be found in ref. [59] for the Abelian case and in ref. [73] for the non-Abelian one.

• We are now ready to refine the notion of 2-representation alluded to above. Firstly, we require a categorification of the notion of vector space, a natural candidate being the notion of 2-vector space introduced in sec. 3.1. Secondly, let $[\bullet, G, \bullet]$ be the 2-groupoid with unique simple object $\bullet$, 1-morphisms labelled by group elements in $G$, and no non-trivial 2-morphisms. Mimicking the definition of $\mathsf{Rep}(G)$, we would like to consider the 2-category $2\mathsf{Rep}(G) := 2\mathsf{Fun}([\bullet, G, \bullet], 2\mathsf{Vec})$ of pseudofunctors, pseudonatural transformations and modifications between $[\bullet, G, \bullet]$ and $2\mathsf{Vec}$. Unpacking the definition, one finds that an object $\rho$ in $2\mathsf{Rep}(G)$ is a map

$$
\begin{aligned}
\rho : [\bullet, G, \bullet] &\to 2\mathsf{Vec}, \\
: \quad \bullet &\mapsto \rho(\bullet) =: \mathsf{V}, \\
: \quad \bullet \xrightarrow{g} \bullet &\mapsto \mathsf{V} \xrightarrow{\rho(g)} \mathsf{V} \in \mathsf{End}(\mathsf{V}),
\end{aligned}
\tag{86}
$$

assigning to the unique object $\bullet$ a 2-vector space $\mathsf{V} := \rho(\bullet)$ and to every 1-morphism $g : \bullet \to \bullet$ a linear functor $\rho(g) : \mathsf{V} \to \mathsf{V}$, in sush a way that composition of 1-morphisms is only preserved up to natural 2-isomorphisms, i.e. $\rho$ further assigns to every pair of 1-morphisms labelled by $g_1, g_2 \in G$ a natural 2-isomorphism

$$
\rho_{g_1, g_2} : \rho(g_1) \circ \rho(g_2) \xRightarrow{\sim} \rho(g_1 g_2),
\tag{87}
$$

which is required to fulfill[13]

$$
\rho_{g_1, g_2 g_3} \cdot [1_{\rho(g_1)} \circ \rho_{g_2, g_3}] = \rho_{g_1 g_2, g_3} \cdot [\rho_{g_2, g_3} \circ 1_{\rho(g_3)}],
\tag{88}
$$

---

[13]As customary, we notate via $\circ$ and $\cdot$ the horizontal and vertical compositions of 2-morphisms, respectively.

for any $g_1, g_2, g_3 \in G$. Introducing the notations $\triangleright : \mathsf{Vec}_G \times \mathsf{V} \to \mathsf{V}$, whereby $\mathbb{C}_g \triangleright M := \rho(g)(M)$ for every $M \in \mathsf{V}$, and $\alpha^\triangleright_{\mathbb{C}_{g_1}, \mathbb{C}_{g_2}, M} := (\rho_{g_1, g_2})_M$, it follows from the 2-cocycle condition (88) that

$$[1_{\mathbb{C}_{g_1}} \triangleright \alpha^\triangleright_{\mathbb{C}_{g_2}, \mathbb{C}_{g_3}, M}] \circ \alpha^\triangleright_{\mathbb{C}_{g_1}, \mathbb{C}_{g_2} \odot \mathbb{C}_{g_3}, M} \circ 1_{\mathbb{C}_{g_1 g_2 g_3} \triangleright M} = \alpha^\triangleright_{\mathbb{C}_{g_1}, \mathbb{C}_{g_2}, \mathbb{C}_{g_3} \triangleright M} \circ \alpha^\triangleright_{\mathbb{C}_{g_1} \odot \mathbb{C}_{g_2}, \mathbb{C}_{g_3}, M}, \quad (89)$$

holds for any $g_1, g_2, g_3 \in G$ and $M \in \mathsf{V}$. Consequently, the triple $(\mathsf{V}, \triangleright, \alpha^\triangleright)$ thus constructed defines a left $\mathsf{Vec}_G$-module category. Similarly, we can readily check that pseudonatural transformations and modifications in $2\mathsf{Rep}(G)$ corresponds to $\mathsf{Vec}_G$-module functors and $\mathsf{Vec}_G$-module natural transformations, respectively. Putting everything together, we have an equivalence $2\mathsf{Rep}(G) \cong \mathsf{Mod}(\mathsf{Vec}_G)$, thereby justifying referring to $\mathsf{Vec}_G$-module categories as 2-representations of the group [3, 71]. The symmetric monoidal structure of $2\mathsf{Vec}$ endows $2\mathsf{Rep}(G)$ with the expected fusion structure according to

$$\begin{aligned} (\rho \odot \varrho)(\bullet) &:= \rho(\bullet) \boxtimes \varrho(\bullet) \equiv \mathsf{V} \boxtimes \tilde{\mathsf{V}}, \\ (\rho \odot \varrho)(g) &:= \rho(g) \boxtimes \varrho(g) \in \mathsf{End}(\mathsf{V} \boxtimes \tilde{\mathsf{V}}). \end{aligned} \quad (90)$$

The main reason to define 2-representations of $G$ as pseudofunctors $[\bullet, G, \bullet] \to 2\mathsf{Vec}$ is that it is readily generalisable to other scenarios relevant to our study. We present two such scenarios below.

• Let $\mathbb{G}$ be a *2-group* with homotopy groups $Q$ and $L$ in degree one and two, respectively [18]. Succinctly, a 2-group is a monoidal groupoid such that every object has a weak inverse. Concretely, the 2-group $\mathbb{G}$ has object-set $Q$, hom-sets $\mathsf{Hom}_{\mathbb{G}}(q, q) = L$ with composition rule[14]

$$(q \xrightarrow{l_1} q) \circ (q \xrightarrow{l_2} q) = (q \xrightarrow{l_1 + l_2} q), \quad (91)$$

and monoidal structure

$$(q_1 \xrightarrow{l_1} q_1) \odot (q_2 \xrightarrow{l_2} q_2) = (q_1 q_2 \xrightarrow{l_1 + \phi_{q_1}(l_2)} q_1 q_2), \quad (92)$$

for any $q_1, q_1 \in Q$ and $l_1, l_2 \in L$, where $\phi_- : Q \to \mathsf{Aut}(L)$. As for a group, we distinguish two ways to treat a 2-group as a 2-groupoid, namely $[Q, L, \bullet]$ and $[\bullet, Q, L]$. Let us focus for now on the former. We are interested in pseudofunctors between $[Q, L, \bullet]$ and $2\mathsf{Vec}$. Unpacking the definition, one finds that such a pseudofunctor is a map

$$\begin{aligned} \rho : [Q, L, \bullet] &\to 2\mathsf{Vec}, \\ : \quad q &\mapsto \rho(q) =: \mathsf{V}_q, \\ : \quad q \xrightarrow{l} q &\mapsto \mathsf{V}_q \xrightarrow{\rho(l)} \mathsf{V}_q \in \mathsf{End}(\mathsf{V}_q), \end{aligned} \quad (93)$$

assigning to every group element $q \in Q$ a 2-vector space $\mathsf{V}_q := \rho(q)$ and to every 1-morphism $l : q \to q$ a linear functor $\rho(l) : \mathsf{V}_q \to \mathsf{V}_q$ in such a way that composition of the 1-morphisms is only preserved up to natural 2-isomorphisms, i.e. $\rho$ further assigns to every pair of 1-morphisms labelled by $l_1, l_2 \in L$ a natural 2-isomorphism

$$\rho_{l_1, l_2} : \rho(l_1) \circ \rho(l_2) \overset{\sim}{\Rightarrow} \rho(l_1 + l_2), \quad (94)$$

which is required to fulfil eq. (88). In close analogy with the constructions presented so far, we deduce that $\rho$ amounts to a $Q$-graded 2-vector space $\mathsf{V} := \boxplus_{q \in Q} \mathsf{V}_q$ such that every homogeneous component $\mathsf{V}_q$ has the structure of a (left) $\mathsf{Vec}_L$-module category, or alternatively of a 2-representation of $L$. Pseudonatural transformations between pseudofunctors $[Q, L, \bullet] \to 2\mathsf{Vec}$

---

[14]Notice that we write the product rule in $L$ as an addition to emphasise the fact that it is an Abelian group.

provide the corresponding 1-morphisms, which amount to $Q$-grading preserving $\mathsf{Vec}_L$-module functors. More concretely, every group element $q \in Q$ together with an indecomposable $\mathsf{Vec}_L$-module category $\mathsf{V}$ furnishes a simple object $\mathsf{V}_q$ in $2\mathsf{Fun}([Q, L, \bullet], 2\mathsf{Vec})$. The hom-category between two such simple objects $\mathsf{V}_{q_1}$ and $\tilde{\mathsf{V}}_{q_2}$ is then provided by $\delta_{q_1,q_2} \mathsf{Fun}_{\mathsf{Vec}_Q}(\mathsf{V}_{q_1}, \tilde{\mathsf{V}}_{q_2})$, i.e., it is terminal unless $q_1 = q_2$. Finally, the convolution product of pseudofunctors $[Q, L, \bullet] \to 2\mathsf{Vec}$ endows $2\mathsf{Fun}([Q, L, \bullet], 2\mathsf{Vec})$ with a fusion structure, whereby $\mathsf{V}_{q_1} \odot \tilde{\mathsf{V}}_{q_2}$ is the $L$-graded 2-vector space with homogeneous components

$$(\mathsf{V}_{q_1} \odot \tilde{\mathsf{V}}_{q_2})_q = \bigboxplus_{x \in Q}(\mathsf{V}_{q_1})_x \boxtimes (\tilde{\mathsf{V}}_{q_2})_{x^{-1}q} = \delta_{q,q_1q_2} \mathsf{V}_{q_1} \boxtimes \tilde{\mathsf{V}}_{q_2}, \tag{95}$$

equipped with a $\mathsf{Vec}_L$-module structure defined by

$$\mathbb{C}_l \rhd (N \boxtimes N') := (\mathbb{C}_l \rhd N) \boxtimes (\mathbb{C}_{\phi_{q_1^{-1}}(l)} \rhd N'), \tag{96}$$

for any $N \in \mathsf{V}_{q_1}$, $N' \in \mathsf{V}'_{q_2}$ and $l \in L$. Henceforth, we denote by $2\mathsf{Vec}_{\mathbb{G}}$ this fusion 2-category and refer to it as the 2-category of $\mathbb{G}$-graded 2-vector spaces [47]. For our applications, the monoidal product of simple objects will not play a crucial role. We shall rather be interested in the monoidal product of simple 1-morphisms, which is of the same form as (96). Consider for instance the monoidal product of the identity 1-endomorphism $1_{\mathsf{V}_q}$ of a 1-simple object $\mathsf{V}_q$ with any simple 1-endomorphism of $\mathsf{Vec}_{\mathbb{1}_Q}$. By virtue of $(\mathsf{Vec}_L)^\star_{\mathsf{Vec}} \cong \hat{\mathsf{Rep}}(L)$, which establishes the Morita equivalence between $\mathsf{Vec}_L$ and $\mathsf{Rep}(L)$ [55], a simple 1-endomorphism of the simple object $\mathsf{Vec}_{\mathbb{1}_Q}$ in $2\mathsf{Vec}_{\mathbb{G}}$ is labelled by a character $\rho(-)$ of $L$. It follows in particular from eq. (96) that $1_{\mathsf{V}_q} \odot \rho(-) = \rho(\phi_{q^{-1}}(-)) \odot 1_{\mathsf{V}_q}$.

• In the notation of the above paragraph, let us finally consider the 2-category $2\mathsf{Fun}([\bullet, Q, L], 2\mathsf{Vec})$, where we recall that $[\bullet, Q, L]$ is the 2-groupoid with single object $\bullet$ and hom-category $\mathsf{Hom}_{[\bullet,Q,L]}(\bullet, \bullet) = \mathbb{G}$ such that horizontal and vertical compositions are provided by the monoidal product and the composition in $\mathbb{G}$, respectively. This 2-category was investigated in detail in ref. [7, 19, 56], or more recently in ref. [5]. As such, we shall keep our exposition brief and merely review the salient features of this 2-category following the description in terms of module categories and module functors proposed in ref. [45]. Unpacking the definition we find that an object in $2\mathsf{Fun}([\bullet, Q, L], 2\mathsf{Vec})$ is a map

$$
\begin{aligned}
\rho: \quad & [\bullet, Q, L] && \to 2\mathsf{Vec}, \\
: \quad & \bullet && \mapsto \rho(\bullet) =: \mathsf{V}, \\
: \quad & \bullet \xrightarrow{q} \bullet && \mapsto \mathsf{V} \xrightarrow{\rho(q)} \mathsf{V} \in \mathsf{End}(\mathsf{V}), \\
: \quad & \bullet \overset{q}{\underset{q}{\Downarrow l}} \bullet && \mapsto \mathsf{V} \overset{\rho(q)}{\underset{\rho(q)}{\rho(l)\Downarrow}} \mathsf{V} \in \mathsf{End}_{\mathsf{End}(\mathsf{V})}(\rho(q)),
\end{aligned}
\tag{97}
$$

assigning to the unique simple object $\bullet$ a 2-vector space $\mathsf{V}$, to every morphism $q: \bullet \to \bullet$ a linear functor $\rho(q): \mathsf{V} \to \mathsf{V}$, and to every 2-morphism $l: q \Rightarrow q$ a natural transformation. Moreover, vertical and horizontal compositions of 2-morphisms are strictly preserved, whereas the composition of 1-morphisms is only preserved up to natural 2-isomorphisms, i.e. $\rho$ further assigns to every pair of 1-morphisms labelled by $q_1, q_2 \in Q$ a natural 2-isomorphism

$$\rho_{q_1,q_2}: \rho(q_1) \circ \rho(q_2) \overset{\sim}{\Rightarrow} \rho(q_1 q_2), \tag{98}$$

which is required to fulfil eq. (88). Given two objects $\rho$ and $\varrho$, a 1-morphism $\theta: \rho \to \varrho$ between them is a pseudonatural transformation that assigns to $\bullet$ an object $\theta_\bullet$ in $\mathsf{Fun}(\rho(\bullet), \varrho(\bullet))$

and to every morphism $q : \bullet \to \bullet$ a *natural* 2-isomorphism defined by

$$\theta_q : \theta_\bullet \circ \rho(q) \overset{\sim}{\Rightarrow} \varrho(q) \circ \theta_\bullet \,, \tag{99}$$

such that $\theta_{\mathbb{1}_Q} = 1_{\theta_\bullet}$. Compatibility with the composition of 1-morphisms in $\mathbb{G}$ requires the following coherence relation to be satisfied:

$$[\varrho_{q_1,q_2} \circ 1_{\theta_\bullet}] \cdot [1_{\varrho(q_1)} \circ \theta_{q_2}] \cdot [\theta_{q_1} \circ 1_{\varrho(q_2)}] = \theta_{q_1 q_2} \cdot [1_{\theta_\bullet} \circ \rho_{q_1,q_2}] \,, \tag{100}$$

for every $q_1, q_2 \in Q$, whereas naturality stipulates that

$$\theta_q \cdot [1_{\theta_\bullet} \circ \rho(l)] = [\varrho(l) \circ 1_{\theta_\bullet}] \cdot \theta_q \,, \tag{101}$$

for every 2-morphism $l : q \Rightarrow q$. In the same vein as the previous derivations, we find that a 2-vector space $\mathsf{V}$ together with endofunctors $\rho(q) : \mathsf{V} \to \mathsf{V}$ and natural 2-isomorphisms $\rho_{q_1,q_2}$ satisfying (88) amounts to a (left) $\mathsf{Vec}_Q$-module category. As a natural transformation, $\rho(l) : \rho(q) \Rightarrow \rho(q)$ assigns to every simple object $N \in \mathsf{V}$ an endomorphism $\rho(q)(N) \to \rho(q)(N)$, which, together with $\rho(l_1) \cdot \rho(l_2) = \rho(l_1 \cdot l_2)$, implies that $\rho$ further assigns to every simple object $N \in \mathsf{V}$ a representation $\rho(-)_N : L \to \text{End}_\mathsf{V}(\rho(q)(N))$. Crucially, $\rho(l_1) \circ \rho(l_2) = \rho(l_1 \circ l_2)$ requires the following condition:

$$\rho(\phi_q(-))_N = \rho(-)_{\rho(q)(N)} \,, \quad \forall q \in Q \,, \text{ and } N \in \mathsf{V} \,. \tag{102}$$

Given the above, a 1-morphism $\rho \to \varrho$ in $2\mathsf{Fun}([\bullet, Q, L], 2\mathsf{Vec})$ amounts to a $\mathsf{Vec}_Q$-module functor $(\theta_\bullet, (\theta_-)_-)$ between the corresponding $\mathsf{Vec}_Q$-module categories, which, in virtue of the naturality condition (101), must satisfy the additional requirement

$$(\theta_q)_N \circ \theta_\bullet(\rho(l)_N) = \varrho(l)_{\theta_\bullet(N)} \circ (\theta_q)_N \,, \tag{103}$$

for every 2-morphism $l : q \Rightarrow q$ and $N \in \mathsf{V}$. Finally, the symmetric monoidal structure of $2\mathsf{Vec}$ endows $2\mathsf{Fun}([\bullet, Q, L], 2\mathsf{Vec})$ with a fusion structure. Henceforth, we denote by $2\mathsf{Rep}(\mathbb{G})$ this fusion 2-category and refer to it as the 2-category of 2-representations of the 2-group $\mathbb{G}$.

### 3.7 Morita duals

Guided by the derivations above, we shall now compute Morita duals of $2\mathsf{Vec}_G$ with respect to various choices of $2\mathsf{Vec}_G$-module 2-categories. In the context of this manuscript, this will establish that given a generic two-dimensional $G$-symmetric Hamiltonian, the models obtained by gauging the $G$ symmetry, or sub-symmetries thereof, are left invariant by symmetry operators organised into fusion 2-categories of some higher representations. Combined with the results of sec. 3.5, this provides an answer to the question, what does it mean for a lattice model to commute with symmetry operators labelled by higher representations?

• Choosing $G$ as a subgroup of itself and the trivial cocycle in $H^3(G, \mathsf{U}(1))$ yields the module 2-category $2\mathsf{Vec}$ via the forgetful functor $2\mathsf{Vec}_G \to 2\mathsf{Vec}$. The Morita dual $(2\mathsf{Vec}_G)^*_{2\mathsf{Vec}}$ was found in in ref. [42, 44] to be equivalent as a monoidal 2-category to $2\mathsf{Rep}(G)$, whose definition was given in sec. 3.6. Let us briefly review this derivation for completeness, we encourage the reader to consult ref. [44] for detail. By definition, an object in $2\mathsf{Fun}_{2\mathsf{Vec}_G}(2\mathsf{Vec}, 2\mathsf{Vec})$ consists of a 2-functor $F : 2\mathsf{Vec} \to 2\mathsf{Vec}$, which is fully determined by a 2-vector space $\mathsf{V} := F(2\mathsf{Vec})$, an adjoint natural 2-equivalence $\omega$ prescribed by

$$\omega_g : \mathsf{Vec}_g \triangleright F(\mathsf{Vec}) \overset{\sim}{\to} F(\mathsf{Vec}_g \triangleright \mathsf{Vec}) \in \mathsf{Fun}(\mathsf{V}, \mathsf{V}) \,, \quad \forall g \in G \,, \tag{104}$$

as well as an invertible modification $\Omega$ defined as per eq. (54) with components

$$\Omega_{g_1,g_2} \in \mathrm{Hom}_{\mathsf{Fun}(V,V)}(\omega_{g_1} \circ \omega_{g_2}, \omega_{g_1 g_2}), \quad \forall\, g_1, g_2 \in G. \tag{105}$$

Isomorphisms $\omega_g$ provide an action 2-functor $\triangleright : \mathsf{Vec}_G \times \mathsf{V} \to \mathsf{V}$ via $\mathbb{C}_g \triangleright M := \omega_g(M)$ for any $M \in \mathsf{V}$, whereas maps $\Omega_{g_1,g_2}$ yields natural isomorphisms $\alpha^\triangleright_{\mathbb{C}_{g_1},\mathbb{C}_{g_2},M} := (\Omega_{g_1,g_2})_M$. It follows from the associahedron axiom (55) that the triple $(\mathsf{V}, \triangleright, \alpha^\triangleright)$ defines a *left* $\mathsf{Vec}_G$-module category. Given a pair $(F, \omega, \Omega)$ and $(\tilde{F}, \tilde{\omega}, \tilde{\Omega})$ of $2\mathsf{Vec}_G$-module 2-endofunctors of $2\mathsf{Vec}$, a $2\mathsf{Vec}_G$-module natural transformation between them is given by a choice of natural transformation $\theta : F \Rightarrow \tilde{F}$ specified by a choice of functor $\hat{F} \in \mathsf{Fun}(V, \tilde{V})$ between the corresponding $\mathsf{Vec}_G$-module categories. The invertible modification $\Theta$ defined as per eq. (71) is prescribed by a collection of natural transformations

$$\Theta_g \in \mathrm{Hom}_{\mathsf{Fun}(V,\tilde{V})}(\hat{F} \circ \omega_g, \tilde{\omega}_g \circ \hat{F}), \quad \forall\, g \in G, \tag{106}$$

endowing $\hat{F}$ with a $\mathsf{Vec}_G$-module structure $\hat{\omega}_{\mathbb{C}_g,M} := (\Theta_g)_M$, so that 1-morphisms are given by $\mathsf{Vec}_G$-module functors $(\hat{F}, \hat{\omega})$ between the corresponding $\mathsf{Vec}_G$-module categories, as expected. Similarly, we can show that $2\mathsf{Vec}_G$-module natural 2-transformations are identified with $\mathsf{Vec}_G$-module natural transformations. Finally, it follows from the composition of $2\mathsf{Vec}_G$-module functors $(F, \omega, \Omega)$ and $(\tilde{F}, \tilde{\omega}, \tilde{\Omega})$ that the monoidal structure is obtained by defining a $\mathsf{Vec}_G$-module structure on $(F \circ \tilde{F})(\mathsf{Vec}) \equiv V \boxtimes \tilde{V}$ via $\omega_g \boxtimes \tilde{\omega}_g : V \boxtimes \tilde{V} \to V \boxtimes \tilde{V}$. Putting everything together, this shows the monoidal equivalence $(2\mathsf{Vec}_G)^\star_{2\mathsf{Vec}} \cong \mathsf{Mod}(\mathsf{Vec}_G) \cong 2\mathsf{Rep}(G)$.

In the context of our work, this computation together with the results of sec. 3.5 shows that gauging the $G$ symmetry of (2+1)d quantum theory results in a theory with a $2\mathsf{Rep}(G)$ symmetry. This result appeared in ref. [44] and was recovered in ref. [5, 24] in terms of separable algebras in fusion 2-categories [42]. Concretely, this means that a theory with a gauged $G$ symmetry host non-trivial topological surface operators labelled by indecomposable $\mathsf{Vec}_G$-module categories as well as topological line operators labelled by $\mathsf{Vec}_G$-module functors. Our construction further teaches us how these operators explicitly act on a lattice model. We have already seen an example in sec. 2, and we shall see further examples below, but in general the surface operator associated with the indecomposable left $\mathsf{Vec}_G$-module category $(\mathcal{N} \equiv \mathcal{N}(B, \psi), \triangleright, \alpha^\triangleright)$ is proportional to[15]

$$\sum_{\substack{\mathfrak{g} \in Z^1(\Sigma_\triangle, G) \\ \mathfrak{n} \in C^0_\mathfrak{g}(\Sigma_\triangle, \mathcal{N})}} \left( \prod_{(v_1 v_2 v_3) \subset \Sigma_\triangle} \alpha^\triangleright(\mathfrak{g}[v_1 v_2], \mathfrak{g}[v_2 v_3])(\mathfrak{n}[v_1])^{\epsilon(v_1 v_2 v_3)} \right) |\mathfrak{g}\rangle\langle\mathfrak{g}|, \tag{107}$$

such that $\alpha^\triangleright_{\mathbb{C}_{g_1},\mathbb{C}_{g_2},N} \equiv \alpha^\triangleright(g_1, g_2)(N) \cdot 1_{(g_1 g_2 \mathsf{r}(N))B}$ and $C^0_\mathfrak{g}(\Sigma_\triangle, \mathcal{N})$ refers to the collection of assignments $\mathfrak{n}$ of simple objects in $\mathcal{N}$ at every vertex of $\Sigma_\triangle$ such that $\mathfrak{n}[v_1] = \mathbb{C}_{\mathfrak{g}[v_1 v_2]} \triangleright \mathfrak{n}[v_2]$ for every $(v_1 v_2) \subset \Sigma_\triangle$. Let us emphasise that the conditions $\mathfrak{n}[v_1] = \mathbb{C}_{\mathfrak{g}[v_1 v_2]} \triangleright \mathfrak{n}[v_2]$ are with respect to the module structure of the $\mathsf{Vec}_G$-module 1-category $\mathcal{N}$. The same operators appeared in ref. [44] in the context of the (3+1)d gauge models of topological phases of matter. In the case where $B = \{\mathbb{1}_G\}$, it is convenient to think of assignments $\mathfrak{n} \in C^0_\mathfrak{g}(\Sigma_\triangle, \mathcal{N})$ as (virtual) matter fields $\mathfrak{n} \in C^0(\Sigma_\triangle, G)$ fulfilling $d\mathfrak{n} = \mathfrak{g}$, as we did in sec. 2. Note finally that we recover through $\mathrm{Hom}_{2\mathsf{Rep}(G)}(\mathsf{Vec}, \mathsf{Vec}) = \mathsf{Fun}_{\mathsf{Vec}_G}(\mathsf{Vec}, \mathsf{Vec}) \cong \mathsf{Rep}(G)$ that Wilson lines generate a 1-form symmetry for the $G$-gauged theory (see sec. 3.8 for more details). More generally, given a surface operator labelled by a $\mathsf{Vec}_G$-module category $\mathcal{N}(B, \psi)$, we can insert topological lines on it that organise into the Morita dual fusion 1-category

---

[15]In the notations of sec. 3.5, we are using the fact that simple morphisms $\mathfrak{f}[v_1' v_1]$ are given by simple objects in hom-categories that are equivalent $\mathcal{N}$.

$(\mathsf{Vec}_G)^\star_{\mathcal{N}(B,\psi)} = \mathsf{Fun}_{\mathsf{Vec}_G}(\mathcal{N}(B,\psi), \mathcal{N}(B,\psi))$. Combining the construction of sec. 3.5 and the computations above, these line operators can be implemented on the lattice in terms of matrix product operators whose building blocks evaluate to the module structure of the functors in $(\mathsf{Vec}_G)^\star_{\mathcal{N}(B,\psi)}$ [89–91].

We commented above that $(\mathsf{Vec}_G)^\star_{\mathsf{Vec}} \cong \mathsf{Rep}(G)$ signifies that the fusion 1-categories $\mathsf{Vec}_G$ and $\mathsf{Rep}(G)$ are Morita equivalent, which implies in particular the equivalence $\mathsf{Mod}(\mathsf{Vec}_G) \cong \mathsf{Mod}(\mathsf{Rep}(G))$ [55]. Interestingly, the fusion 2-category $\mathsf{Mod}(\mathsf{Rep}(G))$ can be thought of as the idempotent completion of the delooping of the *braided* fusion 1-category $\mathsf{Rep}(G)$, which encodes the line operators of the trivial surface operator. Physically, this idempotent completion amounts to including surface operators obtained by condensing suitable algebras of line operators in $\mathsf{Rep}(G)$, and as such these surface operators are often referred to as *condensation defects*. In this context, the surface operators in $2\mathsf{Rep}(G)$ are labelled by *algebra objects* in $\mathsf{Rep}(G)$. By definition, such an algebra object in $\mathsf{Rep}(G)$ is a $G$-algebra, i.e. an associative unital algebra equipped with a $G$-action by algebra automorphisms. We know from the Morita equivalence between $\mathsf{Rep}(G)$ and $\mathsf{Vec}_G$ that Morita classes of indecomposable algebra objects in $\mathsf{Rep}(G)$ are labelled by pairs $(B, \psi)$. Davydov then provided in ref. [41] a recipe to explicitly construct the corresponding $G$-algebras. We already showed in eq. (25) for the case of $G = \mathbb{Z}_2$ how to construct the non-trivial surface operator from this condensation perspective, and we shall provide additional comments along these lines in sec. 5.

Note finally that, more generally, picking a representative of a non-trivial cohomology class in $H^3(G, \mathsf{U}(1))$ yields a $2\mathsf{Vec}_G$-module 2-category $\mathcal{M}(G, \lambda) \equiv 2\mathsf{Vec}^\lambda$ that only differ from $2\mathsf{Vec}$ in the choice of module pentagonator $\pi^\triangleright$, which is such that $\pi^\triangleright_{\mathsf{Vec}_{g_1}, \mathsf{Vec}_{g_2}, \mathsf{Vec}_{g_3}, \mathbb{C}} = \lambda(g_1, g_2, g_3)$ for any $g_1, g_2, g_3 \in G$. Importantly, it follows immediately from the definitions that we still have $(2\mathsf{Vec}_G)^\star_{2\mathsf{Vec}^\lambda} \cong 2\mathsf{Rep}(G)$.

• The subsequent examples require the group $G$ to be isomorphic to a semi-direct product $Q \ltimes_\phi L$ with $L$ Abelian, in which the multiplication is given by

$$(q_1, l_1)(q_1, l_2) = (q_1 q_2, l_1 + \phi_{q_1}(l_2)), \tag{108}$$

for any $q_1, q_2 \in Q$ and $l_1, l_2 \in L$, where we are using the notation of sec. 3.6. Introducing projection maps $\varpi_Q : G \to Q$ and $\varpi_L : G \to L$, every group element in $G$ admits a decomposition of the form $g \equiv (\varpi_Q(g), \varpi_L(g)) \in Q \ltimes_\phi L$. Consider the $2\mathsf{Vec}_G$-module 2-category $\mathcal{M}(L, 1) \cong 2\mathsf{Vec}_Q$.[16] We find that the Morita dual $(2\mathsf{Vec}_G)^\star_{2\mathsf{Vec}_Q}$ is equivalent as a monoidal 2-category to the fusion 2-category $2\mathsf{Vec}_{\mathbb{G}} := 2\mathsf{Fun}([Q, L, \bullet], 2\mathsf{Vec})$ of $\mathbb{G}$-graded 2-vector spaces defined in the previous section. We sketch below the main steps of this derivation.

Any $2\mathsf{Vec}_G$-module 2-endofunctor of $2\mathsf{Vec}_Q$ is in particular an object in $(2\mathsf{Vec}_Q)^\star_{2\mathsf{Vec}_Q} \cong 2\mathsf{Vec}_Q$ so that an object $\mathsf{V}_{q_1} \in 2\mathsf{Vec}_Q$ determines a $2\mathsf{Vec}_G$-module endofunctor of $2\mathsf{Vec}_Q$ of the form $F(-) = - \odot \mathsf{V}_{q_1}$. The $2\mathsf{Vec}_G$-module structure on $F$ is then prescribed by a collection of functors

$$\omega_{g,q} \in \mathsf{Hom}_{2\mathsf{Vec}_Q}\big(\mathsf{Vec}_g \triangleright F(\mathsf{Vec}_q), F(\mathsf{Vec}_g \triangleright \mathsf{Vec}_q)\big) = \mathsf{Hom}_{2\mathsf{Vec}_Q}(\mathsf{Vec}_{\varpi_Q(g)q} \odot \mathsf{V}_{q_1}, \mathsf{Vec}_{\varpi_Q(g)q} \odot \mathsf{V}_{q_1}), \tag{109}$$

satisfying a coherence relation up to an invertible modification $\Omega$ defined as per eq. (54) with components

$$\Omega_{g_1, g_2, q} \in \mathsf{Hom}_{2\mathsf{Vec}_Q}\big(\omega_{g_1, \varpi_Q(g_2)q} \circ \omega_{g_2, q}, \omega_{g_1 g_2, q}\big), \quad \forall\, g_1, g_2 \in G, \text{ and } q \in G. \tag{110}$$

---

[16]By definition, $L \subseteq G$ is a normal subgroup so that $G/L$ isomorphic to $Q$ with the isomorphism being provided by the composition of the natural embedding $L \to G$ and the natural projection $G \to G/L$. It follows that we can identify $\mathcal{M}(L, 1)$ with $2\mathsf{Vec}_Q$ with the $2\mathsf{Vec}_G$-module structure being provided by $\mathsf{Vec}_g \triangleright \mathsf{Vec}_q := \mathsf{Vec}_{\varpi_Q(g)q}$, for all $g \in G$ and $q \in Q$.

As before, 1-morphisms between such objects correspond to $2\mathsf{Vec}_G$-module natural transformations between the corresponding 2-endofunctors of $2\mathsf{Vec}_Q$. Before analysing more carefully this 2-category $(2\mathsf{Vec}_G)^\star_{2\mathsf{Vec}_Q}$, let us immediately consider its fusion structure. Recall that the monoidal product of objects in $(2\mathsf{Vec}_G)^\star_{2\mathsf{Vec}_Q}$ is provided by the composition of the corresponding module 2-endofunctors. Let $(F, \omega, \Omega)$ and $(\tilde{F}, \tilde{\omega}, \tilde{\Omega})$ be two $2\mathsf{Vec}_G$-module 2-endofunctors of $2\mathsf{Vec}_Q$ such that $F(-) = - \odot V_{q_1}$ and $\tilde{F}(-) = - \odot \tilde{V}_{q_2}$, respectively. Composition yields a new module 2-endofunctor of $2\mathsf{Vec}_Q$ given by $(\tilde{F} \circ F)(-) = - \odot (V_{q_1} \odot \tilde{V}_{q_2})$, whose $2\mathsf{Vec}_G$-module structure is provided by

$$
\begin{aligned}
(\tilde{\omega} \circ \omega)_{g,q} : \mathsf{Vec}_g \triangleright (\tilde{F} \circ F)(\mathsf{Vec}_q) &\xrightarrow{\tilde{\omega}_{g,qq_1}} \tilde{F}(\mathsf{Vec}_g \triangleright F(\mathsf{Vec}_q)) \xrightarrow{\tilde{F}(\omega_{g,q})} (\tilde{F} \circ F)(\mathsf{Vec}_g \triangleright \mathsf{Vec}_q) \\
&= \tilde{F}(\omega_{g,q}) \circ \tilde{\omega}_{g,qq_1} \in \mathsf{Hom}_{2\mathsf{Vec}_Q}(\mathsf{Vec}_{\varpi_Q(g)q} \odot (V_{q_1} \odot \tilde{V}_{q_2}), \mathsf{Vec}_{\varpi_Q(g)q} \odot (V_{q_1} \odot \tilde{V}_{q_2})).
\end{aligned}
\tag{111}
$$

Let us now consider the monoidal equivalence given by[17]

$$
\begin{aligned}
(V_{q_1}, \omega_{g,q}, \Omega_{g_1,g_2,q}) &\mapsto (V_{q_1}, \omega_l := \omega_{(\mathbb{1}_Q,l),\mathbb{1}_Q}, \Omega_{l_1,l_2} := \Omega_{(\mathbb{1}_Q,l_1),(\mathbb{1}_Q,l_2),\mathbb{1}_Q}), \\
(V_{q_1}, \omega_l, \Omega_{l_1,l_2}) &\mapsto (V_{q_1}, \omega_{g,q} := \omega_{a_{g,q}}, \Omega_{g_1,g_2,q} := \Omega_{a_{g_1,\varpi_Q(g_2)q},a_{g_2,q}}),
\end{aligned}
\tag{112}
$$

where $a_{g,q} \in L$ was defined in eq. (45). Invoking this equivalence, we find that the $Q$-graded 2-vector space $V_{q_1}$ has the structure of a $\mathsf{Vec}_L$-module category with the module action and the module associator being provided by the collection of maps $\omega_l$ and $\Omega_{l_1,l_2}$, respectively. Furthermore, the fusion structure is now obtained by endowing $V_{q_1} \odot \tilde{V}_{q_2}$ with the $\mathsf{Vec}_L$-module structure

$$
(\tilde{\omega} \circ \omega)_l = (\tilde{\omega} \circ \omega)_{(\mathbb{1}_Q,l),\mathbb{1}_Q} = \omega_{(\mathbb{1}_Q,l),\mathbb{1}_Q} \odot \tilde{\omega}_{(\mathbb{1}_Q,l),q_1} = \omega_l \odot \tilde{\omega}_{\phi_{q_1^{-1}}(l)},
\tag{113}
$$

where we used the fact that $a_{(\mathbb{1}_Q,l),q_1} = \phi_{q_1^{-1}}(l)$, which agrees with eq. (96). We can now check that 1-morphisms in $(2\mathsf{Vec}_G)^\star_{2\mathsf{Vec}_Q}$ amounts to $Q$-grading preserving $\mathsf{Vec}_L$-module functors. Putting everything together, this motivates the equivalence $(2\mathsf{Vec}_G)^\star_{2\mathsf{Vec}_Q} \cong 2\mathsf{Vec}_\mathbb{G}$, where $\mathbb{G}$ is the 2-group defined in sec. 3.6.

Together with previous results, this computation shows that gauging the $L$ sub-symmetry of a $(Q \ltimes_\phi L)$-symmetric (2+1)d quantum theory results in a theory with a $2\mathsf{Vec}_\mathbb{G}$ symmetry. Although it is a little bit tedious to explicitly write down the lattice realisations of the corresponding topological surfaces and lines in general—but these can be obtained from the construction in sec. 3.5—we shall consider specific examples in sec. 5.

• Still assuming $G \simeq Q \ltimes_\phi L$, let us now consider the $2\mathsf{Vec}_G$-module 2-category $\mathcal{M}(Q, 1) \cong 2\mathsf{Vec}_{G/Q}$.[18] Even though $G/Q$ is not isomorphic to $L$ as a group, we have $|G/Q| = |L|$ and thus we label simple objects in $\mathcal{M}(Q, 1)$ by group elements in $L$. We thus write the $2\mathsf{Vec}_G$-structure of $\mathcal{M}(Q, 1)$ as $\mathsf{Vec}_g \triangleright \mathsf{Vec}_l := \mathsf{Vec}_{\varpi_L(g) + \phi_{\varpi_Q(g)}(l)}$ for every $g \in G$ and $l \in L$. Analogously to the previous computation, objects in $(2\mathsf{Vec}_G)^\star_{2\mathsf{Vec}_{G/Q}}$ are functors $F(-) = - \odot V$ with $V = \boxplus_{l_1 \in L} V_{l_1} \in 2\mathsf{Vec}_{G/Q}$ equipped with a $2\mathsf{Vec}_G$-module structure provided by

$$
\omega_{g,l} \in \mathsf{Hom}_{2\mathsf{Vec}_{G/Q}}(\mathsf{Vec}_g \triangleright (\mathsf{Vec}_l \odot V), \mathsf{Vec}_{\varpi_L(g) + \phi_{\varpi_Q}(l)} \odot V) = \mathsf{Hom}_{2\mathsf{Vec}_{G/Q}}(V, V),
\tag{114}
$$

---

[17]This equivalence essentially follows the isomorphism $H^n(G, \mathsf{Hom}(G/L, \mathsf{U}(1))) \simeq H^n(L, \mathsf{U}(1))$ provided by Shapiro's lemma.

[18]Note the slight abuse of notation. Since $Q$ is not a normal subgroup, the quotient $G/Q$ is not equipped with a group structure and thus the 2-category $2\mathsf{Vec}_{G/Q}$ is not monoidal.

satisfying a coherence relation to up an invertible modification $\Omega$ defined as per eq. (54) with components

$$\Omega_{g_1,g_2,l} \in \mathrm{Hom}_{2\mathrm{Vec}_{G/Q}}(\omega_{g_1,\varpi_L(g_1)+\phi_{\varpi_Q(g_1)}(l)} \circ \omega_{g_2,l}, \omega_{g_1 g_2,l}), \quad \forall\, g_1, g_2 \in G, \text{ and } l \in L. \quad (115)$$

Still in the same vein as the previous computation, let us consider the equivalence provided by

$$
\begin{aligned}
(V, \omega_{g,l}, \Omega_{g_1,g_2,l}) &\mapsto \left(V, \omega_q := \omega_{(q,0_L),0_L}, \Omega_{q_1,q_2} := \Omega_{(q_1,0_L),(q_2,0_L),0_L}\right), \\
(V, \omega_q, \Omega_{q_1,q_2}) &\mapsto \left(V, \omega_{g,l} := \omega_{a_{g,l}}, \Omega_{g_1,g_2,l} := \Omega_{a_{g_1,\varpi_L(g_1)+\phi_{\varpi_Q(g_1)}(l)}, a_{g_2,l}}\right),
\end{aligned}
\quad (116)
$$

where $a_{g,l} \in Q$ was defined in eq. (45). In particular, invoking this equivalence, we find that the $L$-graded 2-vector space $V$ has the structure of a $\mathrm{Vec}_Q$-module category with the module action $\rhd$ and the module associator $\alpha^{\rhd}$ being provided by the collections of maps $\omega_q$ and $\Omega_{q_1,q_2}$, respectively. Moreover, going back to eq. (114), we find that $\omega_q = \bigoplus_{l \in L} \omega_g|_{V_{l_1}}$ where $\omega_q|_{V_{l_1}} : V_{\phi_q(l_1)} \to V_{l_1}$. Associating to every object $N \in V$ a group element $l_1(N)$ in such a way that $N \in V_{l_1(N)}$, we must have $\mathbb{C}_q \rhd N := \omega_q|_{V_{l_1}}(N) \in V_{l_1}$ for every $N \in V_{\phi_q(l_1)}$ and thus $l_1(\mathbb{C}_q \rhd N) = \phi_{q^{-1}}(l_1(N))$. Finally, invoking the isomorphism $L \simeq L^\vee = \mathrm{Hom}(L, \mathrm{U}(1))$ and defining a $Q$-action on $L^\vee$ via $q \rhd \rho(-) = \rho(\phi_{q^{-1}}(-))$, we can equivalently state that an object in $(2\mathrm{Vec}_G)^\star_{2\mathrm{Vec}_{Q/G}}$ corresponds to a $\mathrm{Vec}_Q$-module category $V$ such that for every object $N \in V$, we assign a character $\rho(-)_N$ such that $\rho(\phi_q(-))_N = \rho(-)_{\mathbb{C}_q \rhd N}$, for every $q \in Q$. This is precisely the defining condition given in eq. (102). Analysing 1- and 2-morphisms under the same scope, it is then fairly immediate to obtain $(2\mathrm{Vec}_G)^\star_{2\mathrm{Vec}_{G/Q}} \cong 2\mathrm{Rep}(\mathbb{G})$, where $\mathbb{G}$ is the 2-group defined in sec. 3.6.

We summarise in the diagram below the Morita equivalences evoked in this section for a finite group $G \simeq Q \ltimes_\phi L$:

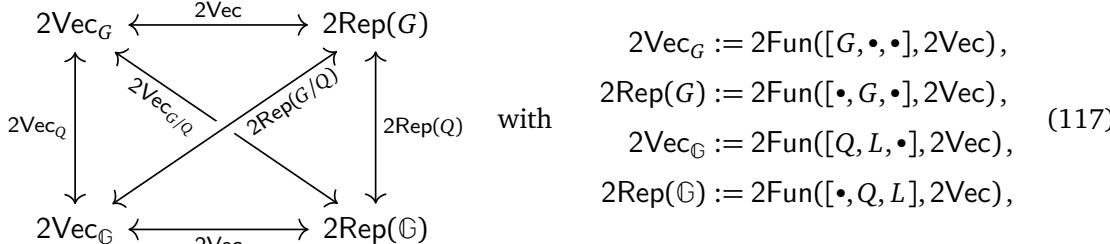

$$
\begin{aligned}
2\mathrm{Vec}_G &:= 2\mathrm{Fun}([G, \bullet, \bullet], 2\mathrm{Vec}), \\
2\mathrm{Rep}(G) &:= 2\mathrm{Fun}([\bullet, G, \bullet], 2\mathrm{Vec}), \\
2\mathrm{Vec}_{\mathbb{G}} &:= 2\mathrm{Fun}([Q, L, \bullet], 2\mathrm{Vec}), \\
2\mathrm{Rep}(\mathbb{G}) &:= 2\mathrm{Fun}([\bullet, Q, L], 2\mathrm{Vec}),
\end{aligned}
\quad (117)
$$

where fusion 2-categories connected by a double arrow are Morita equivalent with respect to the module 2-category labelling the arrow. Note that the equivalence $(2\mathrm{Vec}_G)^\star_{2\mathrm{Vec}} \cong 2\mathrm{Rep}(G)$ holds for arbitrary $G$ as demonstrated at the beginning of this section. Although we have not explicitly constructed all the Morita equivalences displayed above, we included them in this diagram for completeness.[19] We leave to future work a more systematic and general treatment of such Morita equivalences.

## 3.8 Gauging the transverse-field $G$-Ising model

Let us now illustrate some of the concepts presented in this section with a series of examples. Starting from a finite group generalisation of the transverse-field Ising model, we shall construct various dual models obtained by gauging sub-symmetries. Within our approach, this amounts to writing the initial model in terms of local operators (52), and then simply replacing the initial module 2-category by another one. By virtue of our construction, we already know

---

[19]Equivalence $(2\mathrm{Vec}_{\mathbb{G}})^\star_{2\mathrm{Vec}} \cong 2\mathrm{Rep}(\mathbb{G})$ was established in ref. [42].

that the resulting Hamiltonians will commute with symmetry operators encoded into Morita duals with respect to the corresponding module 2-categories. For now, the focus will be on deriving the various dual models using effective local operators (65) obtained after resolving kinematical constraints of the form (50). In the following sections, we shall choose specific groups and analyse in detail the dual symmetries by translating the symmetry operators defined in sec. 3.5 into explicit spin operators.

Given a finite group $G$, let $\mathcal{M} \equiv \mathcal{M}(A, \lambda)$ be an indecomposable $2\mathsf{Vec}_G$-module 2-category. We are interested in Hamiltonians of the form

$$\mathbb{H}^{\mathcal{M}} = \sum_{\mathsf{v} \subset \Sigma_{\triangle}} \sum_{n=1}^{4} \mathbb{h}_{\mathsf{v},n}^{\mathcal{M}}. \tag{118}$$

For any vertex $\mathsf{v} \subset \Sigma_{\triangle}$ and gauge field $\mathfrak{g} \in Z^1(\bigotimes_{\mathsf{v}}^{0}, G)$, the defining complex coefficients $h_{\mathsf{v},n}(\mathfrak{g})$ are chosen to be

$$
\begin{aligned}
h_{\mathsf{v},1}(\mathfrak{g}) &:= -J\delta_{\mathfrak{g}[\mathsf{v}'\mathsf{v}],\mathbb{1}}\delta_{\mathfrak{g}[\mathsf{v}\mathsf{v}+\hat{u}],\mathbb{1}}, \quad h_{\mathsf{v},2}(\mathfrak{g}) := -J\delta_{\mathfrak{g}[\mathsf{v}'\mathsf{v}],\mathbb{1}}\delta_{\mathfrak{g}[\mathsf{v}\mathsf{v}+\hat{v}],\mathbb{1}}, \\
h_{\mathsf{v},3}(\mathfrak{g}) &:= -J\delta_{\mathfrak{g}[\mathsf{v}'\mathsf{v}],\mathbb{1}}\delta_{\mathfrak{g}[\mathsf{v}\mathsf{v}+\hat{w}],\mathbb{1}}, \quad h_{\mathsf{v},4}(\mathfrak{g}) := -\frac{J\kappa}{|G|},
\end{aligned}
\tag{119}
$$

where the branching structure of $\bigotimes_{\mathsf{v}}^{0}$ is that given in eq. (37). This is all the data required to define a Morita class of Hamiltonian models. Specific matrix realisations of this Hamiltonian, i.e. representatives of the Morita class, are obtained by choosing specific $2\mathsf{Vec}_G$-module 2-categories $\mathcal{M}$. We consider below four different choices.

• Let us begin with the choice $\mathcal{M}(A = \{\mathbb{1}_G\}, 1) \cong 2\mathsf{Vec}_G$, i.e. the module 2-category $2\mathsf{Vec}_G$ over itself. Recall that local operators (65) act on the effective Hilbert space obtained after resolving the kinematical constraints (50). Since the kinematical constraints are such that degrees of freedom assigned to edges are fully determined by those assigned to vertices, we are left with a tensor product Hilbert space of the form $\bigotimes_{\mathsf{v}} \mathbb{C}[G] \ni |\mathfrak{m}\rangle$, where $\mathfrak{m} \in C^0(\Sigma_{\triangle}, G)$. In the notation of sec. 3.4, this is the statement that the assignment $\mathfrak{a}_{\mathfrak{g},\mathfrak{m}}$ is such that $\mathfrak{a}_{\mathfrak{g},\mathfrak{m}}[\mathsf{v}_1\mathsf{v}_2] = \mathbb{1}_G$ for every edge $(\mathsf{v}_1\mathsf{v}_2) \subset \bigotimes_{\mathsf{v}}^{0}$. It immediately follows from the definition of the effective local operators and the choice of coefficients (119) that $\mathbb{h}_{\mathsf{v},4}^{2\mathsf{Vec}_G}$ acts as

$$\mathbb{h}_{\mathsf{v},4}^{2\mathsf{Vec}_G} = \frac{-J\kappa}{|G|} \sum_{x \in G} L_{\mathsf{v}}^{x}, \tag{120}$$

where we recall that $L_{\mathsf{v}}^{x}: |\mathfrak{m}[\mathsf{v}]\rangle \mapsto |x\mathfrak{m}[\mathsf{v}]\rangle$. Similarly, we find that local operators $\mathbb{h}_{\mathsf{v},n=1,2,3}^{2\mathsf{Vec}_G}$ act as $-J\Pi_{\mathsf{v},\mathsf{v}+\hat{u}}^{\mathbb{1}_G}$, $-J\Pi_{\mathsf{v},\mathsf{v}+\hat{v}}^{\mathbb{1}_G}$ and $-J\Pi_{\mathsf{v},\mathsf{v}+\hat{w}}^{\mathbb{1}_G}$, respectively, where the vectors $(\hat{u}, \hat{v}, \hat{w})$ were introduced in (36) and $\Pi_{\mathsf{v}_1,\mathsf{v}_2}^{\mathbb{1}_G} := \sum_{\mathfrak{m}} \delta_{\mathfrak{m}[\mathsf{v}_1]^{-1}\mathfrak{m}[\mathsf{v}_2],\mathbb{1}_G} |\mathfrak{m}\rangle\langle\mathfrak{m}|$. Putting everything together, we obtain

$$\mathbb{H}^{2\mathsf{Vec}_G} = -J \sum_{\mathsf{e}} \Pi_{\mathsf{s}(\mathsf{e}),\mathsf{t}(\mathsf{e})}^{\mathbb{1}_G} - \frac{J\kappa}{|G|} \sum_{\mathsf{v}} \sum_{x \in G} L_{\mathsf{v}}^{x}, \tag{121}$$

which we recognise as the finite group generalisation of the transverse-field Ising model. We can readily check that this Hamiltonian possesses a $G$ symmetry, which is consistent with $(2\mathsf{Vec}_G)_{2\mathsf{Vec}_G}^{\star} \cong 2\mathsf{Vec}_G$. Let us now construct dual models resulting from gauging sub-symmetries.

• Within our framework, gauging the whole $G$ symmetry—or rather $2\mathsf{Vec}_G$ symmetry— amounts to choosing the $2\mathsf{Vec}_G$-module 2-category $\mathcal{M}(G, 1) \cong \mathsf{Vec}$. Clearly, with this choice, there are no degrees of freedom left at vertices so the effective microscopic Hilbert space is

spanned by states $|\mathfrak{g}\rangle \in \bigotimes_e \mathbb{C}[G]$, where $\mathfrak{g} \in Z^1(\Sigma_\triangle, G)$. The condition $d\mathfrak{g} = \mathbb{1}_G$ imposes kinematical constraints $\mathfrak{g}[v_1v_2]\mathfrak{g}[v_2v_3] = \mathfrak{g}[v_1v_3]$ for every triangle $(v_1v_2v_3) \subset \Sigma_\triangle$ so that $\mathfrak{g}$ defines a $G$-gauge field. In the notation of sec. 3.4, we have $\mathfrak{a}_{\mathfrak{g},m} = \mathfrak{g}$. Going back to the definition of the local operators (65), it readily follows from the flatness condition of $\mathfrak{g}$ that

$$\mathbb{h}_{v,4}^{2\mathsf{Vec}} = -\frac{J\kappa}{|G|} \sum_{x \in G} \Big(\prod_{e \to v} R_e^x\Big)\Big(\prod_{e \leftarrow v} L_e^x\Big) \equiv -\frac{J\kappa}{|G|}\sum_{x \in G} \mathbb{A}_v^x \equiv -J\kappa \mathbb{A}_v \,, \tag{122}$$

where $e \to v$ and $e \leftarrow v$ refer to edges $e \subset \Sigma_\triangle$ such that $t(e) = v$ and $s(e) = v$, respectively. Similarly, we find that the local operators $\mathbb{h}_{v,n=1,2,3}^{2\mathsf{Vec}_G}$ read $-J\mathbb{П}_{(v\,v+\hat{u})}^{\mathbb{1}_G}$, $-J\mathbb{П}_{(v\,v+\hat{v})}^{\mathbb{1}_G}$ and $-J\mathbb{П}_{(v\,v+\hat{w})}^{\mathbb{1}_G}$, respectively, where $\mathbb{П}_e^{\mathbb{1}_G} = \sum_{\mathfrak{g}} \delta_{\mathfrak{g}[e],\mathbb{1}_G} |\mathfrak{g}\rangle\langle\mathfrak{g}|$. Putting everything together, we obtain

$$\mathbb{H}^{2\mathsf{Vec}} = -J \sum_e \mathbb{П}_e^{\mathbb{1}_G} - J\kappa \sum_v \mathbb{A}_v \,, \tag{123}$$

which we recognise as the pure Ising $G$-gauge theory. We know from the general construction that this model has a $(2\mathsf{Vec}_G)_{2\mathsf{Vec}}^\star \cong 2\mathsf{Rep}(G)$ symmetry and we provided in eq. (107) a formula for constructing the corresponding surface and operators. In the following section, we shall study these symmetry operators on the lattice in more detail for specific choices of input group $G$, but let us make a few general comments in the meantime. Straightforward examples of surface operators are those associated with simple objects of the form $\mathsf{Vec}^\psi \in 2\mathsf{Rep}(G)$, where $\mathsf{Vec}^\psi$ is the $\mathsf{Vec}_G$-module category that only differs from $\mathsf{Vec}$ in the choice of module associator $\alpha^\triangleright$, which is such that $\alpha_{\mathbb{C}_{g_1},\mathbb{C}_{g_2},\mathbb{C}}^\triangleright = \psi(g_1, g_2)$ for every $g_1, g_2 \in G$. Going back to the general definition provided in sec. 3.5 we find these surface operators act diagonally by multiplication by the evaluation of the 3-cocycle characterising the corresponding module associator. In symbols, these are proportional to

$$\sum_{\mathfrak{g} \in Z^1(\Sigma_\triangle, G)} \Big(\prod_{(v_1v_2v_3)} \psi(\mathfrak{g}[v_1v_2], \mathfrak{g}[v_2v_3])^{\epsilon(v_1v_2v_3)}\Big)|\mathfrak{g}\rangle\langle\mathfrak{g}| \,. \tag{124}$$

In particular, the commutation relation with operators $\mathbb{A}_v$ introduced in eq. (123) follows from the 2-cocycle condition $d\psi = 1$. Topological lines living on such a surface operator were shown in sec. 3.5 to be labelled by 1-endomorphisms of $\mathsf{Vec}^\psi$ in $2\mathsf{Rep}(G)$, which correspond by definition to $\mathsf{Vec}_G$-module endofunctors of $\mathsf{Vec}^\psi$ in $\mathsf{Fun}_{\mathsf{Vec}_G}(\mathsf{Vec}^\psi, \mathsf{Vec}^\psi) \cong \mathsf{Rep}(G)$. Therefore, these amount to ordinary Wilson lines labelled by representations of $G$. Explicitly, the closed Wilson line operator labelled by $\rho \in \mathsf{Rep}(G)$ with support the closed path $\ell$ reads

$$\sum_{\mathfrak{g} \in Z^1(\Sigma_\triangle, G)} \mathrm{tr}\Big(\overset{\rightarrow}{\prod_{e \subset \ell}} \rho(\mathfrak{g}[e]^{\epsilon(e,\ell)})\Big)|\mathfrak{g}\rangle\langle\mathfrak{g}| \,. \tag{125}$$

The fact that these commute with the Hamiltonian eq. (121), and in particular with operators $\mathbb{A}_v$, follows from the gauge invariance of Wilson loop operators. These generate the 1-form $\mathsf{Rep}(G)$ symmetry of the model. We could also consider an open version of the surface operator (124) labelled by $\mathsf{Vec}^\psi$, together with topological lines living at the interface of this surface operator and the trivial one labelled by $\mathsf{Vec} \in 2\mathsf{Rep}(G)$. Mimicking the derivation of $(\mathsf{Vec}_G)_{\mathsf{Vec}}^\star \cong \mathsf{Rep}(G)$ [55], we immediately find that such topological lines are labelled by simple objects in $\mathsf{Fun}_{\mathsf{Vec}_G}(\mathsf{Vec}, \mathsf{Vec}^\psi) \cong \mathsf{Rep}^\psi(G)$, i.e. projective representations of the group $G$ with Schur's multiplier $\psi$. Explicit examples of such symmetry operators are presented in the next section.

Note finally that instead of considering the $2\mathsf{Vec}_G$-module 2-category $2\mathsf{Vec}$, we could have considered instead $\mathcal{M}(G, \lambda) \cong 2\mathsf{Vec}^\lambda$ where $\lambda$ is a non-trivial 3-cocycle in $H^3(G, U(1))$. Recall from sec. 3.7 that $2\mathsf{Vec}^\lambda$ only differs from $2\mathsf{Vec}$ in the choice of module pentagonator

$\pi^{\triangleright}$. Physically, this amounts to the $\lambda$-twisted gauging of the $G$ symmetry. In the case of $\mathbb{Z}_2$, and given the only non-trivial 3-cocycle in $H^3(\mathbb{Z}_2, U(1)) \simeq \mathbb{Z}_2$, the resulting model would precisely correspond to that considered in sec. 2. We already commented on the fact that $(2\text{Vec}_G)^\star_{2\text{Vec}^\lambda} \cong 2\text{Rep}(G)$ so the resulting twisted $G$-gauge theory would have the symmetry operators as $\mathbb{H}^{2\text{Vec}}$.

• In order to proceed with the next two cases, we further assume that the group $G$ is a semi-direct product of the form $G \simeq Q \ltimes_\phi L$ with $L$ Abelian. Recall that we write group elements as $g \equiv (\varpi_Q(g), \varpi_L(g)) \in Q \ltimes_\phi L$ and the multiplication is given by (108). Let us now consider the gauging of the $L$ sub-symmetry, which amounts to choosing the $2\text{Vec}_G$-module 2-category $\mathcal{M}(A = L, 1) \cong 2\text{Vec}_Q$. We know from sec. 3.4 that the effective microscopic Hilbert space is spanned by states $|\mathfrak{l}, \mathfrak{m}\rangle \in \bigotimes_e \mathbb{C}[L] \bigotimes_v \mathbb{C}[Q]$, where $\mathfrak{l} \in Z^1(\Sigma_\triangle, L)$. Let us now work out how the local operators $\mathbb{h}_{v,n}^{\mathcal{M}(L,1)}$ act on this Hilbert space. Going back to the definition, we have

$$\mathbb{h}_{v,4}^{\mathcal{M}(L,1)} = -\frac{J\kappa}{|G|} \sum_{\substack{\mathfrak{g} \in Z^1(\otimes_v^{\mathfrak{l}}, G) \\ \mathfrak{m} \in C_\mathfrak{g}^0(\otimes_v^{\mathfrak{l}}, \mathcal{M})}} \left| (\mathfrak{a}_{\mathfrak{g},\mathfrak{m}}, \mathfrak{m}) \left[ \otimes_v^1 \right] \right\rangle \left\langle (\mathfrak{a}_{\mathfrak{g},\mathfrak{m}}, \mathfrak{m}) \left[ \otimes_v^0 \right] \right|. \tag{126}$$

We shall first consider the operator associated with a fixed pair $(\mathfrak{g}, \mathfrak{m}) \in Z^1(\otimes_v^{\mathfrak{l}}, G) \times C_\mathfrak{g}^0(\otimes_v^{\mathfrak{l}}, \mathcal{M})$. Up to the scalar prefactor $\frac{-J\kappa}{|G|}$, this operator acts on the state $|\mathfrak{m}[v]\rangle$ as

$$|\mathfrak{m}[v]\rangle \mapsto |\mathfrak{m}[v']\rangle = |\text{Vec}_{\mathfrak{g}[v'v]} \triangleright \mathfrak{m}[v]\rangle = |\varpi_Q(\mathfrak{g}[v'v]) \mathfrak{m}[v]\rangle, \tag{127}$$

while it acts on a state $|\mathfrak{l}[vv_1]\rangle \equiv |\mathfrak{a}_{\mathfrak{g},\mathfrak{m}}[vv_1]\rangle = |\mathfrak{a}_{\mathfrak{g}[vv_1],\mathfrak{m}[v_1]}\rangle$ as

$$|\mathfrak{l}[vv_1]\rangle \mapsto |\mathfrak{a}_{\mathfrak{g}[v'v]\mathfrak{g}[vv_1],\mathfrak{m}[v_1]}\rangle = |\mathfrak{l}[vv_1] + \mathfrak{a}_{\mathfrak{g},\mathfrak{m}}[v'v]\rangle, \tag{128}$$

where we made use of (the Abelian version of) eq. (45). Similarly, it acts on a state $|\mathfrak{l}[v_1v]\rangle \equiv |\mathfrak{a}_{\mathfrak{g},\mathfrak{m}}[v_1v]\rangle = |\mathfrak{a}_{\mathfrak{g}[v_1v],\mathfrak{m}[v]}\rangle$ as

$$|\mathfrak{l}[v_1v]\rangle \mapsto |\mathfrak{a}_{\mathfrak{g}[v_1v]\mathfrak{g}[v'v]^{-1}, \mathfrak{g}[v'v] \triangleright \mathfrak{m}[v]}\rangle = |\mathfrak{l}[v_1v] + \mathfrak{a}_{\mathfrak{g},\mathfrak{m}}[vv']\rangle. \tag{129}$$

Invoking eq. (63), we have $\mathfrak{a}_{\mathfrak{g},\mathfrak{m}}[v_1v_2] = \phi_{\mathfrak{m}[v_1]^{-1}}\big(\varpi_L(\mathfrak{g}[v_1v_2])\big)$ so that

$$\begin{aligned} \mathfrak{a}_{\mathfrak{g},\mathfrak{m}}[v'v] &= \phi_{\mathfrak{m}[v']^{-1}}\big(\varpi_L(\mathfrak{g}[v'v])\big), \\ \mathfrak{a}_{\mathfrak{g},\mathfrak{m}}[vv'] &= -\phi_{\mathfrak{m}[v']^{-1}}\big(\varpi_L(\mathfrak{g}[v'v])\big), \end{aligned} \tag{130}$$

where we used the fact that $\mathfrak{g}[vv'] = \mathfrak{g}[v'v]^{-1} \equiv \big(\varpi_Q(\mathfrak{g}[v'v])^{-1}, -\phi_{\varpi_Q(\mathfrak{g}[v'v])^{-1}}(\varpi_L(\mathfrak{g}[v'v]))\big)$. These expressions in turn allow us to rewrite the actions (128) and (129) more explicitly. Keeping in mind that $\mathfrak{m}[v'] = \varpi_Q(\mathfrak{g}[v'v])\mathfrak{m}[v]$, we obtain the following expression for the local operator $\mathbb{h}_{v,4}^{\mathcal{M}(L,1)}$:

$$\mathbb{h}_{v,4}^{\mathcal{M}(L,1)} = -\frac{J\kappa}{|G|} \sum_{x \in G} \Big( \prod_{e \to v} {}^\phi R_{e,v}^x \Big) \Big( \prod_{e \leftarrow v} {}^\phi L_{e,v}^x \Big) L_v^{\varpi_Q(x)} \equiv -\frac{J\kappa}{|G|} \sum_{x \in G} {}^\phi \mathbb{A}_v^x \equiv -J\kappa\, {}^\phi \mathbb{A}_v, \tag{131}$$

where

$$\begin{aligned} {}^\phi L_{e,v}^x &: |\mathfrak{l}[e]\rangle \mapsto |\mathfrak{l}[e] + \phi_{\mathfrak{m}[v]^{-1}}(\varpi_L(x))\rangle, \\ {}^\phi R_{e,v}^x &: |\mathfrak{l}[e]\rangle \mapsto |\mathfrak{l}[e] - \phi_{\mathfrak{m}[v]^{-1}}(\varpi_L(x))\rangle. \end{aligned} \tag{132}$$

Moreover, it immediately follows from the definitions that that local operators $\mathbb{h}_{v,n=1,2,3}^{\mathcal{M}(L,1)}$ act as $-J\mathbb{\Pi}_{(vv+\hat{u})}^{\mathbb{1}_Q, 0_L}$, $-J\mathbb{\Pi}_{(vv+\hat{v})}^{\mathbb{1}_Q, 0_L}$ and $-J\mathbb{\Pi}_{(vv+\hat{w})}^{\mathbb{1}_Q, 0_L}$, respectively, where

$$\mathbb{\Pi}_{(v_1v_2)}^{\mathbb{1}_Q, 0_L} := \sum_{\mathfrak{m}, \mathfrak{l}} \delta_{\mathfrak{m}[v_1]^{-1}\mathfrak{m}[v_2], \mathbb{1}_Q}\, \delta_{\mathfrak{l}[v_1v_2], 0_L} |\mathfrak{l}, \mathfrak{m}\rangle\langle\mathfrak{l}, \mathfrak{m}|. \tag{133}$$

Putting everything together, we obtain[20]

$$\mathbb{H}^{\mathcal{M}(L,1)} = -J \sum_{\mathsf{e}} \mathbb{\Pi}_{\mathsf{e}}^{\mathbb{1}_Q, 0_L} - J\kappa \sum_{\mathsf{v}} {}^{\phi}\mathbb{A}_{\mathsf{v}} \,. \tag{134}$$

We know from the general construction of sec. 3.5 that this Hamiltonian must commute with symmetry operators encoded into the Morita dual $(2\mathrm{Vec}_G)^\star_{\mathcal{M}(L,1)}$, which was shown in sec. 3.7 to be equivalent to the fusion 2-category $2\mathrm{Vec}_{\mathbb{G}}$ of $\mathbb{G}$-graded 2-vector spaces. In particular, the model possesses a 1-form $\mathrm{Rep}(L)$ symmetry, which is acted upon by a 0-form $2\mathrm{Vec}_Q$ symmetry that is not on-site. Instead of explaining the lattice implementations of these symmetry operators in the general case, we shall provide explicit parametrisations in terms of spin operators in sec. 5 for the case of the symmetric group $\mathcal{S}_3$ of degree 3.

• Finally, we consider gauging the $Q$ sub-symmetry, which amounts to choosing the $2\mathrm{Vec}_G$-module 2-category $\mathcal{M}(A = Q, 1) \cong 2\mathrm{Vec}_{G/Q}$. The effective microscopic Hilbert space is spanned by states $|\mathfrak{q}, \mathfrak{m}\rangle \in \bigotimes_{\mathsf{e}} \mathbb{C}[Q] \bigotimes_{\mathsf{v}} \mathbb{C}[L]$ where $\mathfrak{q} \in Z^1(\Sigma_\triangle, Q)$.[21] Mimicking the previous derivation, given a fixed pair $(\mathfrak{g}, \mathfrak{m}) \in Z^1(\otimes_{\mathsf{v}}^{\mathbb{I}}, G) \times C^0_{\mathfrak{g}}(\otimes_{\mathsf{v}}^{\mathbb{I}}, \mathcal{M})$ and up to the scalar prefactor $\frac{-J\kappa}{|G|}$, we have an operator that acts on the state $|\mathfrak{m}[\mathsf{v}]\rangle$ as

$$|\mathfrak{m}[\mathsf{v}]\rangle \mapsto |\mathfrak{m}[\mathsf{v}']\rangle = |\mathrm{Vec}_{\mathfrak{g}[\mathsf{v}'\mathsf{v}]} \triangleright \mathfrak{m}[\mathsf{v}]\rangle = \big|\varpi_L(\mathfrak{g}[\mathsf{v}'\mathsf{v}]) + \phi_{\varpi_Q(\mathfrak{g}[\mathsf{v}'\mathsf{v}])}(\mathfrak{m}[\mathsf{v}])\big\rangle \,, \tag{135}$$

while it acts on states $|\mathfrak{q}[\mathsf{vv}_1]\rangle \equiv |\mathfrak{a}_{\mathfrak{g},\mathfrak{m}}[\mathsf{vv}_1]\rangle$ and $|\mathfrak{q}[\mathsf{v}_1\mathsf{v}]\rangle \equiv ||\mathfrak{a}_{\mathfrak{g},\mathfrak{m}}[\mathsf{vv}_1]\rangle$ as

$$
\begin{aligned}
|\mathfrak{q}[\mathsf{vv}_1]\rangle &\mapsto \big|\varpi_Q(\mathfrak{g}[\mathsf{v}'\mathsf{v}])\mathfrak{q}[\mathsf{vv}_1]\big\rangle \,, \quad \text{and} \\
|\mathfrak{q}[\mathsf{v}_1\mathsf{v}]\rangle &\mapsto \big|\mathfrak{q}[\mathsf{v}_1\mathsf{v}]\varpi_Q(\mathfrak{g}[\mathsf{v}'\mathsf{v}])^{-1}\big\rangle \,,
\end{aligned}
\tag{136}
$$

respectively. In the latter equations, we used the fact $\mathfrak{a}_{\mathfrak{g},\mathfrak{m}}[\mathsf{v}_1\mathsf{v}_2] = a_{\mathfrak{g}[\mathsf{v}_1\mathsf{v}_2],\mathfrak{m}[\mathsf{v}_2]} = \varpi_Q(\mathfrak{g}[\mathsf{v}_1\mathsf{v}_2])$. We thus obtain the following expression for the local operators $\mathbb{h}^{\mathcal{M}(Q,1)}_{\mathsf{v},4}$:

$$\mathbb{h}^{\mathcal{M}(Q,1)}_{\mathsf{v},4} = -\frac{J\kappa}{|G|} \sum_{x \in G} \Big(\prod_{\mathsf{e} \to \mathsf{v}} R_{\mathsf{e}}^{\varpi_Q(x)}\Big) {}^{\phi}L_{\mathsf{v}}^x \Big(\prod_{\mathsf{e} \leftarrow \mathsf{v}} L_{\mathsf{e}}^{\varpi_Q(x)}\Big) \equiv -\frac{J\kappa}{|G|} \sum_{x \in G} {}^{\phi}\widetilde{\mathbb{A}}_{\mathsf{v}}^x \equiv -J\kappa \, {}^{\phi}\widetilde{\mathbb{A}}_{\mathsf{v}} \,, \tag{137}$$

where

$$\phi L_{\mathsf{v}}^x : |\mathfrak{m}[\mathsf{v}]\rangle \mapsto \big|\varpi_L(x) + \phi_{\varpi_Q(x)}(\mathfrak{m}[\mathsf{v}])\big\rangle \,. \tag{138}$$

Similarly, we find that local operators $\mathbb{h}^{\mathcal{M}(Q,1)}_{\mathsf{v},n=1,2,3}$ acts as $-J\mathbb{\Pi}^{0_L,\mathbb{1}_Q}_{(\mathsf{vv}+\hat{u})}$, $-J\mathbb{\Pi}^{0_L,\mathbb{1}_Q}_{(\mathsf{vv}+\hat{v})}$ and $-J\mathbb{\Pi}^{0_L,\mathbb{1}_Q}_{(\mathsf{vv}+\hat{w})}$, respectively, where

$$\mathbb{\Pi}^{0_L,\mathbb{1}_Q}_{(\mathsf{v}_1\mathsf{v}_2)} := \sum_{\mathfrak{m},\mathfrak{q}} \delta_{\mathfrak{m}[\mathsf{v}_1],\mathfrak{m}[\mathsf{v}_2]} \, \delta_{\mathfrak{q}[\mathsf{v}_1\mathsf{v}_2],\mathbb{1}_Q} |\mathfrak{q},\mathfrak{m}\rangle\langle\mathfrak{q},\mathfrak{m}| \,. \tag{139}$$

Putting everything together, we obtain

$$\mathbb{H}^{\mathcal{M}(Q,1)} = -J \sum_{\mathsf{e}} \mathbb{\Pi}_{\mathsf{e}}^{0_L,\mathbb{1}_Q} - J\kappa \sum_{\mathsf{v}} {}^{\phi}\widetilde{\mathbb{A}}_{\mathsf{v}} \,. \tag{140}$$

---

[20]An alternative Hamiltonian resulting from the gauging of a normal subgroup sub-symmetry of $\mathbb{H}^{2\mathrm{Vec}_G}$ is often found in the literature, see e.g. ref. [115, 117, 119]. The Hamiltonian found in these references is related to ours via unitary transformation $|\phi_{\mathfrak{m}}(\mathfrak{l}), \mathfrak{m}\rangle\langle\mathfrak{l}, \mathfrak{m}|$, where $\phi_{\mathfrak{m}}(\mathfrak{l})[\mathsf{e}] = \phi_{\mathfrak{m}[s(\mathsf{e})]}(\mathfrak{l}[\mathsf{e}])$. The point of this additional unitary is for the remaining 0-form $2\mathrm{Vec}_Q$ to be on-site so it can be subsequently gauged following the canonical approach. However, the resulting model then possesses a *twisted* 1-form $\mathrm{Rep}(L)$ symmetry.

[21]Recall that even though $G/Q$ is not isomorphic to $L$ as a group, we can still identify objects $M$ in $\mathcal{M}(Q, 1)$ with group elements in $l \in L$ such that $M = lQ$.

We know from the general construction of sec. 3.5 that this Hamiltonian must commute with symmetry operators encoded into the Morita dual $(2\mathsf{Vec}_G)^\star_{\mathcal{M}(Q,1)}$, which was shown in sec. 3.7 to be equivalent to the fusion 2-category $2\mathsf{Rep}(\mathbb{G})$ of 2-representations of the 2-group $\mathbb{G}$. As for the previous example, we shall refrain from describing the lattice implementations of the corresponding topological surfaces and topological lines in the general case, and shall rather focus in sec. 5 on the specific case of the symmetric group $\mathcal{S}_3$.

**Back to the transverse-field Ising model:** We conclude this section by specialising once more to the case of the transverse-field ($\mathbb{Z}_2$-)Ising model. Let us focus on Hamiltonian (10) obtained by choosing the $2\mathsf{Vec}_{\mathbb{Z}_2}$-module category $2\mathsf{Vec}$. We established that by construction this model has a $2\mathsf{Rep}(\mathbb{Z}_2)$ symmetry. There are two simple objects in $2\mathsf{Rep}(\mathbb{Z}_2)$ provided by the two indecomposable $\mathsf{Vec}_{\mathbb{Z}_2}$-module categories, namely $\mathsf{Vec}$ and $\mathsf{Vec}_{\mathbb{Z}_2}$. The corresponding surface operators were notated via $\mathcal{U}^{\text{triv.}}$ and $\mathcal{U}^{\mathbb{Z}_2}$ in sec. 2, respectively. It follows from the alternative definition provided in eq. (21) and the preceding paragraph that $\mathcal{U}^{\mathbb{Z}_2}$ indeed corresponds to surface operator (107) for $\mathcal{N} = \mathsf{Vec}_{\mathbb{Z}_2}$. Line operators living on the surface operator $\mathcal{U}^{\text{triv.}}$ are now identified with simple 1-morphisms in $\mathsf{Hom}_{2\mathsf{Rep}(\mathbb{Z}_2)}(\mathsf{Vec}, \mathsf{Vec}) \cong \mathsf{Rep}(\mathbb{Z}_2)$. Line operators labelled by the non-trivial representation of $\mathbb{Z}_2$ as defined in eq. (125) readily correspond to (22). Similarly, we recover line operators on the surface operator $\mathcal{U}^{\mathbb{Z}_2}$ as simple 1-morphisms in $\mathsf{Hom}_{2\mathsf{Rep}(\mathbb{Z}_2)}(\mathsf{Vec}_{\mathbb{Z}_2}, \mathsf{Vec}_{\mathbb{Z}_2}) \cong \mathsf{Vec}_{\mathbb{Z}_2}$. What about line operators at the junctions of surface operators $\mathcal{U}^{\mathbb{Z}_2}$ and $\mathcal{U}^{\text{triv.}}$? We established in sec. 2 that such lines are unique up to isomorphisms. Within the framework of this section, these correspond to the unique simple objects in the hom-categories $\mathsf{Hom}_{2\mathsf{Rep}(\mathbb{Z}_2)}(\mathsf{Vec}_{\mathbb{Z}_2}, \mathsf{Vec}) = \mathsf{Fun}_{\mathsf{Vec}_{\mathbb{Z}_2}}(\mathsf{Vec}_{\mathbb{Z}_2}, \mathsf{Vec}) \cong \mathsf{Vec}$ and $\mathsf{Fun}_{\mathsf{Vec}_{\mathbb{Z}_2}}(\mathsf{Vec}, \mathsf{Vec}_{\mathbb{Z}_2}) \cong \mathsf{Vec}$. Moreover, composition of $\mathsf{Vec}_{\mathbb{Z}_2}$-module functors $\mathsf{Fun}_{\mathsf{Vec}_{\mathbb{Z}_2}}(\mathsf{Vec}_{\mathbb{Z}_2}, \mathsf{Vec}) \times \mathsf{Fun}_{\mathsf{Vec}_{\mathbb{Z}_2}}(\mathsf{Vec}, \mathsf{Vec}_{\mathbb{Z}_2}) \to \mathsf{Rep}(\mathbb{Z}_2)$ informs us that composing the corresponding line operators yields a line operator living on $\mathcal{U}^{\text{triv.}}$ labelled by the regular representation in $\mathsf{Rep}(\mathbb{Z}_2)$, which is compatible with eq. (32). Similarly, composition of $\mathsf{Vec}_{\mathbb{Z}_2}$-module functors $\mathsf{Fun}_{\mathsf{Vec}_{\mathbb{Z}_2}}(\mathsf{Vec}, \mathsf{Vec}_{\mathbb{Z}_2}) \times \mathsf{Fun}_{\mathsf{Vec}_{\mathbb{Z}_2}}(\mathsf{Vec}_{\mathbb{Z}_2}, \mathsf{Vec}) \to \mathsf{Vec}_{\mathbb{Z}_2}$ informs us that composing the corresponding line operators yields a line operator living on $\mathcal{U}^{\mathbb{Z}_2}$ labelled by the object $\mathbb{C}_0 \oplus \mathbb{C}_1$ in $\mathsf{Vec}_{\mathbb{Z}_2}$, which is compatible with eq. (34). Finally, the monoidal structure of $2\mathsf{Rep}(\mathbb{Z}_2)$ is such that $\mathsf{Vec}_{\mathbb{Z}_2} \odot \mathsf{Vec}_{\mathbb{Z}_2} \cong \mathsf{Vec}_{\mathbb{Z}_2} \boxplus \mathsf{Vec}_{\mathbb{Z}_2}$, which amounts to (19) when $\Sigma$ is the two-torus.

# 4 Example: Doubled transverse-field Ising model

*In this section, we study in detail the symmetry structure of the model obtained by gauging the $\mathbb{Z}_2^2$ symmetry of the doubled transverse-field Ising model.*

## 4.1 Symmetric Hamiltonian and gauging

The starting point is the doubled transverse-field Ising model on a triangulation $\Sigma_\triangle$ of a closed oriented surface $\Sigma$. Pairs of qubit degrees of freedom are assigned to vertices $\mathsf{v} \subset \Sigma_\triangle$. We identify such an assignment with a choice of 0-cochain $\mathfrak{m} \in C^0(\Sigma_\triangle, \mathbb{Z}_2^2)$ so the microscopic Hilbert space is provided by the tensor product $\bigotimes_{\mathsf{v}} \mathbb{C}[\mathbb{Z}_2^2] \simeq \mathbb{C}^4$, on which two sets of Pauli operators denoted as $\sigma_{\mathsf{v}}^{\mu,I}$ with $I = 1, 2$ act. The doubled transverse-field Ising model is then defined via the Hamiltonian

$$\mathbb{H}^{2\mathsf{Vec}_{\mathbb{Z}_2^2}} = -\sum_{I=1}^{2} \left( J_{I,1} \sum_{\mathsf{e}} \sigma_{\mathsf{s}(\mathsf{e})}^{z,I} \sigma_{\mathsf{t}(\mathsf{e})}^{z,I} + J_{I,2} \sum_{\mathsf{v}} \sigma_{\mathsf{v}}^{x,I} \right). \qquad (141)$$

The model has a (0-form) global $\mathbb{Z}_2^2$ symmetry implemented by surface operators

$$\mathcal{O}^g = \prod_{\mathsf{v}} (\sigma_{\mathsf{v}}^{x,1})^{g_1} (\sigma_{\mathsf{v}}^{x,2})^{g_2}, \tag{142}$$

for every $g \equiv (g_1, g_2) \in \mathbb{Z}_2^2$. Fusion rules of these surface operators are dictated by the multiplication rule in $\mathbb{Z}_2^2$.

In the language of the previous section, the symmetry structure of this model is encapsulated in the fusion 2-category $2\mathsf{Vec}_{\mathbb{Z}_2^2}$, whose four simple objects correspond to the surface operators $\mathcal{O}^{(0,0)}$, $\mathcal{O}^{(0,1)}$, $\mathcal{O}^{(1,0)}$ and $\mathcal{O}^{(1,1)}$, respectively. Moreover, recall that for any simple object $\mathsf{Vec}_g$ in $2\mathsf{Vec}_{\mathbb{Z}_2^2}$, its endo-category is equivalent to $\mathsf{Vec}$, whose unique simple object corresponds to the identity line operator living on $\mathcal{O}^g$. Finally, since there are no 1-morphisms in $2\mathsf{Vec}_{\mathbb{Z}_2^2}$ between distinct simple objects, there are no topological lines between distinct surface operators.

Note that for conciseness we only consider a minimal $\mathbb{Z}_2^2$-symmetric transverse-field Ising model, which realises the symmetric paramagnetic and symmetry-broken gapped phases. In particular, this Hamiltonian does not realise any SPT phase.[22] However, as was explained in the previous section, details of the Hamiltonian are irrelevant to the ensuing analysis of the gauging procedure and the symmetry structure of the gauged model, so that the following derivations hold for any model with the same $\mathbb{Z}_2^2$ symmetry.

Given the input fusion 2-category $2\mathsf{Vec}_{\mathbb{Z}_2^2}$, Hamiltonian (141) is implicitly defined with respect to the module 2-category $2\mathsf{Vec}_{\mathbb{Z}_2^2}$ over itself. In this case, gauging the $\mathbb{Z}_2^2$ symmetry simply amounts to choosing instead the $2\mathsf{Vec}_{\mathbb{Z}_2^2}$-module 2-category $2\mathsf{Vec}$. That being said, since this Hamiltonian is merely a doubled version of that considered in sec. 2, we can immediately infer from the procedure outlined there the resulting dual Hamiltonian:

$$\mathbb{H}^{2\mathsf{Vec}} = -\sum_{I=1}^{2} \left( J_{I,1} \sum_{\mathsf{e}} \sigma_{\mathsf{e}}^{z,I} + J_{I,2} \sum_{\mathsf{v}} \prod_{\mathsf{e} \supset \mathsf{v}} \sigma_{\mathsf{e}}^{x,I} \right), \tag{144}$$

which acts on the physical Hilbert space spanned by states $|\mathfrak{g}\rangle$, where $\mathfrak{g} \equiv (\mathfrak{g}_1, \mathfrak{g}_2) \in Z^1(\Sigma_\triangle, \mathbb{Z}_2^2)$. Recall that we chose the basis such that

$$\sigma_{\mathsf{e}}^{z,I} |\mathfrak{g}\rangle = (-1)^{\mathfrak{g}_I[\mathsf{e}]} |\mathfrak{g}\rangle. \tag{145}$$

In this basis, the first term in the Hamiltonian (144) acts diagonally, while an arbitrary combination of operators $\prod_{\mathsf{e} \supset \mathsf{v}} \sigma_{\mathsf{e}}^{x,I}$ indexed by a 0-cochain $\mathfrak{r} \equiv (\mathfrak{r}_1, \mathfrak{r}_2) \in C^0(\Sigma_\triangle, \mathbb{Z}_2^2)$ acts as

$$\mathbb{A}^{\mathfrak{r}} := \prod_{\mathsf{v} \subset \Sigma_\triangle} \prod_{\mathsf{e} \supset \mathsf{v}} \bigotimes_{I=1}^{2} (\sigma_{\mathsf{e}}^{x,I})^{\mathfrak{r}_I[\mathsf{v}]} = \sum_{\mathfrak{g}} |\mathfrak{g} + \mathrm{d}\mathfrak{r}\rangle \langle \mathfrak{g}|. \tag{146}$$

We know from the results of sec. 3.2 that Hamiltonian (144) must commute with various surface and line operators that are organised into the Morita dual fusion 2-category

---

[22]The classification of SPT phases with 0-form global $\mathbb{Z}_2^2$ symmetry is given by the cohomology group $H^3(\mathbb{Z}_2^2, U(1)) \simeq \mathbb{Z}_2^3$. Therefore, there are eight distinct topological phases which can be labelled by $p \equiv (p_1, p_2, p_3) \in \mathbb{Z}_2^3$. The corresponding fixed-point Hamiltonians, which we denote as $H_p$, have the form

$$\mathbb{H}_{(1,0,0)} = -\sum_{\mathsf{v}} \sigma_{\mathsf{v}}^{x,1} \prod_{(\mathsf{v}\mathsf{v}_1\mathsf{v}_2)} \exp\left( \frac{\mathrm{i}\pi}{4}(1 - \sigma_{\mathsf{v}_1}^{z,1}\sigma_{\mathsf{v}_2}^{z,1}) \right), \quad \mathbb{H}_{(0,1,0)} = -\sum_{\mathsf{v}} \sigma_{\mathsf{v}}^{x,2} \prod_{(\mathsf{v}\mathsf{v}_1\mathsf{v}_2)} \exp\left( \frac{\mathrm{i}\pi}{4}(1 - \sigma_{\mathsf{v}_1}^{z,2}\sigma_{\mathsf{v}_2}^{z,2}) \right),$$

$$\mathbb{H}_{(0,0,1)} = -\sum_{\mathsf{v}} \sigma_{\mathsf{v}}^{x,1} \prod_{(\mathsf{v}\mathsf{v}_1\mathsf{v}_2)} \exp\left( \frac{\mathrm{i}\pi}{4}(1 - \sigma_{\mathsf{v}_1}^{z,1}\sigma_{\mathsf{v}_2}^{z,1}) \right) - \sum_{\mathsf{v}} \sigma_{\mathsf{v}}^{x,2} \prod_{(\mathsf{v}\mathsf{v}_1\mathsf{v}_2)} \exp\left( \frac{\mathrm{i}\pi}{4}(1 - \sigma_{\mathsf{v}_1}^{z,2}\sigma_{\mathsf{v}_2}^{z,2}) \right). \tag{143}$$

$(2\text{Vec}_{\mathbb{Z}_2^2})^\star_{\text{Vec}} \cong 2\text{Rep}(\mathbb{Z}_2^2)$. Recall from sec. 3.6 that simple objects in $2\text{Rep}(\mathbb{Z}_2^2)$ are provided by indecomposable $\text{Vec}_{\mathbb{Z}_2^2}$-module categories $\mathcal{N}(B, \psi)$, which are conveniently labelled by tuples $(B, \psi)$ consisting of a subgroup $B \subseteq \mathbb{Z}_2^2$ and a 2-cocycle $\psi$ in $H^2(B, U(1))$. Therefore, we count six simple objects in $2\text{Rep}(\mathbb{Z}_2^2)$ labelled by the tuples $(\mathbb{Z}_2^2, 1)$, $(\mathbb{Z}_2^2, \psi)$, $(\mathbb{Z}_2^{(1)}, 1)$, $(\mathbb{Z}_2^{(2)}, 1)$, $(\mathbb{Z}_2^{(\text{diag.})}, 1)$ and $(\mathbb{Z}_1, 1)$, respectively, where $\psi$ refers here to a normalised representative of the non-trivial cohomology class in $H^2(\mathbb{Z}_2^2, U(1)) \simeq \mathbb{Z}_2$. Each such simple object provides a surface operator commuting with (144). Furthermore, there are various line operators within each surface operator as well as at interfaces between surface operators associated with distinct simple objects in $2\text{Rep}(\mathbb{Z}_2^2)$.[23] The remainder of this section is dedicated to explicitly constructing these various operators.

## 4.2  $2\text{Rep}(\mathbb{Z}_2^2)$ symmetry: invertible surface operators

We begin our detailed analysis of the symmetry structure of Hamiltonian (144) by enumerating the *invertible* surface operators. Firstly, there is of course the identity operator[24]

$$\mathcal{U}^{\text{triv.}} = \prod_{\mathsf{e}} \text{id}_{\mathsf{e}} \,, \tag{147}$$

which corresponds to the identity object $\mathcal{N}(\mathbb{Z}_2^2, 1) \cong \text{Vec}$ in $2\text{Rep}(\mathbb{Z}_2^2)$. As explained in sec. 3.8, line operators living on this trivial operator form the hom-category

$$\text{Hom}_{2\text{Rep}(\mathbb{Z}_2^2)}(\text{Vec}, \text{Vec}) \cong \text{Fun}_{\text{Vec}_{\mathbb{Z}_2^2}}(\text{Vec}, \text{Vec}) \cong \text{Rep}(\mathbb{Z}_2^2) \,. \tag{148}$$

We provided in eq. (125) a general formula for such line operators but we can make it more explicit by specialising to $G = \mathbb{Z}_2^2$. Given a 1-cycle $\ell$ on $\Sigma_\triangle$ and an irreducible representation $(\rho_1, \rho_2) \in \text{Rep}(\mathbb{Z}_2^2)$, we may define such a line operator as

$$\sum_{\mathfrak{g}} \Big( \prod_{\mathsf{e} \subset \ell} \prod_I \rho_I(\mathfrak{g}_I[\mathsf{e}]) \Big) |\mathfrak{g}\rangle\langle\mathfrak{g}| \,, \tag{149}$$

which readily commutes with (144). For instance, choosing both $\rho_1$ and $\rho_2$ to be the non-trivial irreducible representation of $\mathbb{Z}_2$, the operator above can be equivalently defined as $\prod_{\mathsf{e} \subset \ell} \sigma_{\mathsf{e}}^{z,1} \sigma_{\mathsf{e}}^{z,2}$. More generally, an operator corresponding to a network of lines in $\text{Rep}(\mathbb{Z}_2^2)$ can be defined as

$$\mathcal{U}^{\text{triv.}}(\mathfrak{f}) = \sum_{\mathfrak{g}} (-1)^{\int_{\Sigma_\triangle}(\mathfrak{f}_1 \smile \mathfrak{g}_1 + \mathfrak{f}_2 \smile \mathfrak{g}_2)} |\mathfrak{g}\rangle\langle\mathfrak{g}| \,, \tag{150}$$

where $\mathfrak{f} = (\mathfrak{f}_1, \mathfrak{f}_2) \in Z^1(\Sigma_\triangle, \mathbb{Z}_2^2)$. It follows directly from the definition that the composition such lines satisfies

$$\mathcal{U}^{\text{triv.}}(\mathfrak{f}_1 \circ \mathfrak{f}_2) = \mathcal{U}^{\text{triv.}}(\mathfrak{f}_1 + \mathfrak{f}_2) \,, \tag{151}$$

which does amount to the monoidal structure in $\text{Rep}(\mathbb{Z}_2^2)$. Similarly, the fusion of lines read

$$\mathcal{U}^{\text{triv.}}(\mathfrak{f}_1) \odot \mathcal{U}^{\text{triv.}}(\mathfrak{f}_2) = \mathcal{U}^{\text{triv.}}(\mathfrak{f}_1 + \mathfrak{f}_2) \,, \tag{152}$$

as predicted by the monoidal structure in $2\text{Rep}(\mathbb{Z}_2^2)$. The second and final invertible surface corresponds to the simple object $\mathcal{N}(\mathbb{Z}_2^2, \psi) \cong \text{Vec}^\psi$ in $2\text{Rep}(\mathbb{Z}_2^2)$, which as a $\text{Vec}_{\mathbb{Z}_2^2}$-module category only differs from Vec in the choice of module associator. We provided in eq. (124) a general expression for the corresponding type of surface operator. Writing $\psi(g, g') := (-1)^{g_1 g_2'}$,

---

[23]Note that two objects that have a non-trivial 1-morphism between them are said to belong to the same Schur component. Physically, this means that there is a condensation process relating both objects [47, 70, 105].

[24]Notice that, in a way that is reminiscent of the construction of the corresponding module categories, we notate the surface operator associated with the tuple $(B, \psi)$ via $\mathcal{U}^{G/B, \psi}$.

we find it is a non-trivial operator that acts diagonally on basis states as

$$\mathcal{U}^{\psi}[\Sigma_{\triangle}] = \sum_{\mathfrak{g}} (-1)^{\int_{\Sigma_{\triangle}} \mathfrak{g}_1 \smile \mathfrak{g}_2} |\mathfrak{g}\rangle\langle\mathfrak{g}| . \tag{153}$$

This operator can be shown to commute with the Hamiltonian. On the one hand, it is diagonal in the chosen computational basis, thus it clearly commutes with the first term in (144). On the other hand, the SPT being non-anomalous is gauge invariant, and thus commutes with the second term. The operator can also be defined with non-trivial line insertions which are also labelled by $\mathfrak{f} \in Z^1(\Sigma_{\triangle}, \mathbb{Z}_2^2)$ as

$$\mathcal{U}^{\psi}(\mathfrak{f})[\Sigma_{\triangle}] = \sum_{\mathfrak{g}} (-1)^{\int_{\Sigma_{\triangle}} \mathfrak{g}_1 \smile \mathfrak{g}_2 + \mathfrak{f}_1 \smile \mathfrak{g}_1 + \mathfrak{f}_2 \smile \mathfrak{g}_2} |\mathfrak{g}\rangle\langle\mathfrak{g}| , \tag{154}$$

which satisfy composition rules analogous to (151). It follows that such lines also encoded into $\mathsf{Rep}(\mathbb{Z}_2^2)$. As evoked in sec. 3.8, this is explained by the fact that $\mathsf{Fun}_{\mathsf{Vec}_{\mathbb{Z}_2^2}}(\mathsf{Vec}^{\psi}, \mathsf{Vec}^{\psi}) \cong \mathsf{Rep}(\mathbb{Z}_2^2)$. The operators $\mathcal{U}^{\psi}[\Sigma_{\triangle}]$ and $\mathcal{U}^{\mathrm{triv.}}$ satisfy $\mathbb{Z}_2$ fusion rules of the form

$$(\mathcal{U}^{\psi} \odot \mathcal{U}^{\psi})[\Sigma_{\triangle}] = \mathcal{U}^{\mathrm{triv.}}, \quad (\mathcal{U}^{\psi} \odot \mathcal{U}^{\mathrm{triv.}})[\Sigma_{\triangle}] = \mathcal{U}^{\psi}[\Sigma_{\triangle}] = (\mathcal{U}^{\psi} \odot \mathcal{U}^{\mathrm{triv.}})[\Sigma_{\triangle}] , \tag{155}$$

which readily follows from eq. (153). Similarly, composition rules for surface operators with line operators inserted take the form

$$\begin{aligned}
(\mathcal{U}^{\psi}(\mathfrak{f}_1) \odot \mathcal{U}^{\psi}(\mathfrak{f}_2))[\Sigma_{\triangle}] &= \mathcal{U}^{\mathrm{triv.}}(\mathfrak{f}_1 + \mathfrak{f}_2)[\Sigma_{\triangle}] , \\
(\mathcal{U}^{\psi}(\mathfrak{f}_1) \odot \mathcal{U}^{\mathrm{triv.}}(\mathfrak{f}_2))[\Sigma_{\triangle}] &= \mathcal{U}^{\psi}(\mathfrak{f}_1 + \mathfrak{f}_2)[\Sigma_{\triangle}] , \\
(\mathcal{U}^{\mathrm{triv.}}(\mathfrak{f}_1) \odot \mathcal{U}^{\psi}(\mathfrak{f}_2))[\Sigma_{\triangle}] &= \mathcal{U}^{\psi}(\mathfrak{f}_1 + \mathfrak{f}_2)[\Sigma_{\triangle}] .
\end{aligned} \tag{156}$$

Interestingly, one may also define the operator $\mathcal{U}^{\psi}$ on an open sub-complex $\Xi_{\triangle} \subseteq \Sigma_{\triangle}$. Equivalently, this is the statement that there is a topological line operator between the operators $\mathcal{U}^{\mathrm{triv.}}$ and $\mathcal{U}^{\psi}$. Naively, the operator $\mathcal{U}^{\psi}[\Xi_{\triangle}]$ simply defined by restricting definition (153) to the open sub-complex $\Xi_{\triangle}$ does not commute with the Hamiltonian (144) since

$$\begin{aligned}
\mathbb{A}^{\mathfrak{r}} \mathcal{U}^{\psi}[\Xi_{\triangle}] &= \sum_{\mathfrak{g}} \exp\left( i\pi \int_{\Xi_{\triangle}} \mathfrak{g}_1 \smile \mathfrak{g}_2 \right) |\mathfrak{g} + d\mathfrak{r}\rangle\langle\mathfrak{g}| , \\
\mathcal{U}^{\psi}[\Xi_{\triangle}] \mathbb{A}^{\mathfrak{r}} &= \sum_{\mathfrak{g}} \exp\left( i\pi \int_{\Xi_{\triangle}} \mathfrak{g}_1 \smile \mathfrak{g}_2 + i\pi \oint_{\partial\Xi_{\triangle}} \zeta(\mathfrak{g}, \mathfrak{r}) \right) |\mathfrak{g} + d\mathfrak{r}\rangle\langle\mathfrak{g}| ,
\end{aligned} \tag{157}$$

where $d\zeta(\mathfrak{g}, \mathfrak{r}) = (\mathfrak{g}_1 + d\mathfrak{r}) \smile (\mathfrak{g}_2 + d\mathfrak{r}) - \mathfrak{g}_1 \smile \mathfrak{g}_2$. Such a lack of commutation is remedied by appending a line operator on the boundary $\partial\Xi_{\triangle}$, which has the form

$$\mathcal{U}^{\psi|\mathrm{triv.}}[\partial\Xi_{\triangle}] = \sum_{\mathfrak{g}} \mathcal{Z}_{\mathrm{anom.}}(\mathfrak{g})[\partial\Xi_{\triangle}] |\mathfrak{g}\rangle\langle\mathfrak{g}| , \tag{158}$$

where the amplitude $\mathcal{Z}_{\mathrm{anom.}}(\mathfrak{g})[\partial\Xi_{\triangle}]$ can be understood as the partition function of a quantum mechanical (i.e., $(0+1)$-dimensional) system with an anomalous $\mathbb{Z}_2^2$ symmetry encoded into the (unique) irreducible projective representation with Schur multiplier $\psi$. Concretely, this operator can be constructed by considering auxiliary $\mathbb{Z}_2^2$-valued vertex degrees of freedom $\mathfrak{p}, \mathfrak{q}$ on $\partial\Xi_{\triangle}$ such that

$$\mathcal{Z}_{\mathrm{anom.}}(\mathfrak{g})[\partial\Xi_{\triangle}] = \sum_{\mathfrak{b}, \mathfrak{n}} \exp\left( i\pi \oint_{\partial\Xi_{\triangle}} \left( \delta^{I,J} \mathfrak{b}_I \smile (d\mathfrak{n}_J + \mathfrak{g}_J) + d\mathfrak{n}_1 \smile d\mathfrak{n}_2 \right) \right) , \tag{159}$$

which has the required property

$$\mathcal{Z}_{\text{anom.}}(\mathfrak{g} + \mathrm{d}\mathfrak{r})[\partial \Xi_\triangle] = \exp\left(i\pi \oint_{\partial \Xi_\triangle} \zeta(\mathfrak{g},\mathfrak{r})\right) \mathcal{Z}_{\text{anom.}}(\mathfrak{g})[\partial \Xi_\triangle]. \tag{160}$$

It follows that the combined operator

$$\mathcal{U}^{\psi \to \text{triv.}}[\Xi_\triangle] := \mathcal{U}^\psi[\Xi_\triangle] \cdot \mathcal{U}^{\psi|\text{triv.}}[\partial \Xi_\triangle], \tag{161}$$

commutes with the Hamiltonian and is therefore a symmetry operator. Graphically, we depict such a configuration as

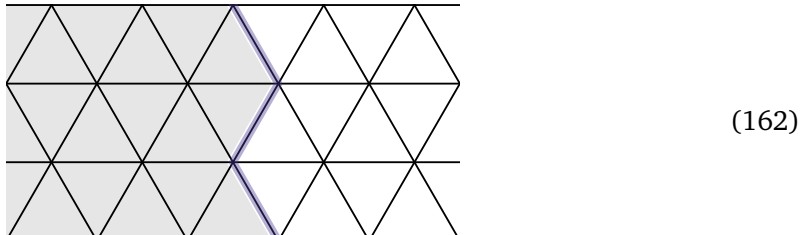

$$\tag{162}$$

where the gray area depicts $\Xi_\triangle$ and the blue coloured edges the support of the line operator. From a category theoretic standpoint, recall that line operators at the interface of $\mathcal{U}^\psi$ and $\mathcal{U}^{\text{triv.}}$ are organised into the fusion category $\mathsf{Fun}_{\mathsf{Vec}_{\mathbb{Z}_2^2}}(\mathsf{Vec}, \mathsf{Vec}^\psi) \cong \mathsf{Rep}^\psi(\mathbb{Z}_2^2)$ of $\psi$-projective representations of $\mathbb{Z}_2^2$, which is compatible with the above construction. Similarly, we define an operator $\mathcal{U}^{\text{triv.} \to \psi}[\Xi_\triangle]$.

Finally, the composition of the topological line $\mathcal{U}^{\psi|\text{triv.}}[\partial \Xi_\triangle]$ follows from the tensor product of representations. Since the line $\mathcal{U}^{\psi|\text{triv.}}[\partial \Xi_\triangle]$ carries a $\psi$-projective representation of $\mathbb{Z}_2^2$, composing it with itself must yield an object labelled by the trivial representation in $\mathsf{Rep}(\mathbb{Z}_2^2)$. We can compose this line between the trivial surface operators $\mathcal{U}^{\text{triv.}}$ and the non-trivial operator $\mathcal{U}^\psi$ in two ways so as to recover the trivial representation line either within the trivial surface or within the non-trivial surface operator, i.e.

$$\begin{aligned} \mathcal{U}^{\psi \to \text{triv.}}[\Xi_\triangle] \circ \mathcal{U}^{\text{triv.} \to \psi}[\Sigma_\triangle/\Xi_\triangle] &= \mathcal{U}^\psi[\Sigma_\triangle], \\ \mathcal{U}^{\text{triv.} \to \psi}[\Xi_\triangle] \circ \mathcal{U}^{\psi \to \text{triv.}}[\Sigma_\triangle/\Xi_\triangle] &= \mathcal{U}^{\text{triv.}}, \end{aligned} \tag{163}$$

which is mathematically encoded into the composition of the corresponding $\mathsf{Vec}_{\mathbb{Z}_2^2}$-module functors. This concludes our analysis of the invertible surface operators.

## 4.3 $2\mathsf{Rep}(\mathbb{Z}_2^2)$ symmetry: non-invertible surface operators

We continue our analysis of the symmetry structure of Hamiltonian (144) with the study of surface operators that have *non-invertible* fusion rules. In terms of simple objects in $2\mathsf{Rep}(\mathbb{Z}_2^2)$, these are the ones labelled by the tuples $(\mathbb{Z}_2^{(1)}, 1)$, $(\mathbb{Z}_2^{(2)}, 1)$, $(\mathbb{Z}_2^{\text{diag.}}, 1)$ and $(\mathbb{Z}_1, 1)$. Let us focus for now on the first three surface operators. Mimicking the definition of the non-invertible surface operator described in sec. 2, one finds:

$$\mathcal{U}^{\mathbb{Z}_2^{(1)}}[\Sigma_\triangle] = \frac{1}{2^{\#(\Sigma_\triangle)}} \sum_{\mathfrak{g},\mathfrak{n},\mathfrak{b}} (-1)^{\int_{\Sigma_\triangle} \mathfrak{b} \smile (\mathrm{d}\mathfrak{n} + \mathfrak{g}_1)} |\mathfrak{g}\rangle\langle\mathfrak{g}| = \frac{1}{2^{\chi(\Sigma_\triangle)}} \sum_{\mathfrak{g},\mathfrak{n}} \delta_{\mathrm{d}\mathfrak{n},\mathfrak{g}_1} |\mathfrak{g}\rangle\langle\mathfrak{g}|,$$

$$\mathcal{U}^{\mathbb{Z}_2^{(2)}}[\Sigma_\triangle] = \frac{1}{2^{\#(\Sigma_\triangle)}} \sum_{\mathfrak{g},\mathfrak{n},\mathfrak{b}} (-1)^{\int_{\Sigma_\triangle} \mathfrak{b} \smile (\mathrm{d}\mathfrak{n} + \mathfrak{g}_2)} |\mathfrak{g}\rangle\langle\mathfrak{g}| = \frac{1}{2^{\chi(\Sigma_\triangle)}} \sum_{\mathfrak{g},\mathfrak{n}} \delta_{\mathrm{d}\mathfrak{n},\mathfrak{g}_2} |\mathfrak{g}\rangle\langle\mathfrak{g}|, \tag{164}$$

$$\mathcal{U}^{\mathbb{Z}_2^{(\text{diag.})}}[\Sigma_\triangle] = \frac{1}{2^{\#(\Sigma_\triangle)}} \sum_{\mathfrak{g},\mathfrak{n},\mathfrak{b}} (-1)^{\int_{\Sigma_\triangle} \mathfrak{b} \smile (\mathrm{d}\mathfrak{n} + \mathfrak{g}_1 + \mathfrak{g}_2)} |\mathfrak{g}\rangle\langle\mathfrak{g}| = \frac{1}{2^{\chi(\Sigma_\triangle)}} \sum_{\mathfrak{g},\mathfrak{n}} \delta_{\mathrm{d}\mathfrak{n},\mathfrak{g}_1 + \mathfrak{g}_2} |\mathfrak{g}\rangle\langle\mathfrak{g}|,$$

where the summation variables are $\mathfrak{n} \in C^0(\Sigma_\triangle, \mathbb{Z}_2)$ and $\mathfrak{b} \in C^1(\Sigma_\triangle, \mathbb{Z}_2)$. As in sec. 2, we have defined $\#(\Sigma_\triangle) = 2^{|\Sigma_\triangle^0| + |\Sigma_\triangle^2|}$ and $\chi(\Sigma_\triangle)$ is the Euler characteristic of $\Sigma_\triangle$. Summing over $\mathfrak{b}$, imposes a constraint that pins $d\mathfrak{n}$ on each edge of the triangulation to be $\mathfrak{g}_1, \mathfrak{g}_2$ and $\mathfrak{g}_1 + \mathfrak{g}_2$ for the three operators $\mathcal{U}^{\mathbb{Z}_2^{(1)}}, \mathcal{U}^{\mathbb{Z}_2^{(2)}}$ and $\mathcal{U}^{\mathbb{Z}_2^{(\text{diag.})}}$, respectively. These operators should be thought of as an explicit version of the general operator (107). There, $\mathfrak{n}$ refers to an assignment of simple objects in the $\mathsf{Vec}_{\mathbb{Z}_2^2}$-module categories $\mathcal{N}(\mathbb{Z}_2^{(1)}, 1), \mathcal{N}(\mathbb{Z}_2^{(2)}, 1)$ and $\mathcal{N}(\mathbb{Z}_2^{(\text{diag.})})$, which are all equivalent to $\mathsf{Vec}_{\mathbb{Z}_2}$ as categories, satisfying $\mathfrak{n}[v_1] = \mathbb{C}_{\mathfrak{g}[v_1 v_2]} \triangleright \mathfrak{n}[v_2]$ for every edge $(v_1 v_2) \subset \Sigma_\triangle$. In particular, the $\mathsf{Vec}_{\mathbb{Z}_2^2}$-module structure on $\mathcal{N}(\mathbb{Z}_2^{(\text{diag.})}, 1)$ is given by $\mathbb{C}_g \triangleright N := (g_1 + g_2) + N$ mod 2, for any $g \equiv (g_1, g_2) \in \mathbb{Z}_2^2$ and $N \in \mathsf{Vec}_{\mathbb{Z}_2}$. This amounts to the condition $d\mathfrak{n} = \mathfrak{g}_1 + \mathfrak{g}_2$ in the equation above.

Summing over $\mathfrak{n}$ instead of $\mathfrak{b}$ in eq. (164) imposes $d\mathfrak{b} = 0$. Summing over equivalence classes of 1-cocycles, i.e., $\mathfrak{f} \in H^1(\Sigma_\triangle, \mathbb{Z}_2)$, one finds

$$\mathcal{U}^{\mathbb{Z}_2^{(1)}}[\Sigma_\triangle] = \frac{1}{|H^0(\Sigma_\triangle, \mathbb{Z}_2)|} \sum_{\mathfrak{g},\mathfrak{f}} (-1)^{\int_{\Sigma_\triangle} \mathfrak{f} \smile \mathfrak{g}_1} |\mathfrak{g}\rangle\langle\mathfrak{g}|,$$

$$\mathcal{U}^{\mathbb{Z}_2^{(2)}}[\Sigma_\triangle] = \frac{1}{|H^0(\Sigma_\triangle, \mathbb{Z}_2)|} \sum_{\mathfrak{g},\mathfrak{f}} (-1)^{\int_{\Sigma_\triangle} \mathfrak{f} \smile \mathfrak{g}_2} |\mathfrak{g}\rangle\langle\mathfrak{g}|, \tag{165}$$

$$\mathcal{U}^{\mathbb{Z}_2^{(\text{diag.})}}[\Sigma_\triangle] = \frac{1}{|H^0(\Sigma_\triangle, \mathbb{Z}_2)|} \sum_{\mathfrak{g},\mathfrak{f}} (-1)^{\int_{\Sigma_\triangle} \mathfrak{f} \smile (\mathfrak{g}_1 + \mathfrak{g}_2)} |\mathfrak{g}\rangle\langle\mathfrak{g}|.$$

From these expressions, it is clear that these operators are condensation defects of the $\mathsf{Rep}(\mathbb{Z}_2^2)$ lines described previously. It turns out that this alternative form of the operators is particularly convenient to demonstrate the commutativity of these operators with the Hamiltonian. It follows from the operators being diagonal in the chosen basis and invariance under the action of $\mathbb{A}^{\mathfrak{r}}$.

Still in analogy with the non-invertible surface operator considered in sec. 2, the surfaces (164) can be defined with topological lines inserted. Recall that these must be encoded into Morita duals of $\mathsf{Vec}_{\mathbb{Z}_2^2}$ with respect to the corresponding module categories. For instance, lines living on $\mathcal{U}^{\mathbb{Z}_2^{(1)}}$ form the fusion 1-category

$$\mathsf{Fun}_{\mathsf{Vec}_{\mathbb{Z}_2^2}}(\mathsf{Vec}_{\mathbb{Z}_2^{(1)}}, \mathsf{Vec}_{\mathbb{Z}_2^{(1)}}) \cong \mathsf{Vec}_{\mathbb{Z}_2^{(1)}} \boxtimes \mathsf{Rep}(\mathbb{Z}_2^{(2)}). \tag{166}$$

The corresponding surface operator with a network of lines inserted can be constructed by choosing 1-cocycles $\tilde{\mathfrak{f}}_1, \mathfrak{f}_2 \in Z^1(\Sigma_\triangle, \mathbb{Z}_2)$ as

$$\mathcal{U}^{\mathbb{Z}_2^{(1)}}(\tilde{\mathfrak{f}}_1, \mathfrak{f}_2)[\Sigma_\triangle] = \frac{1}{2^{\#(\Sigma_\triangle)}} \sum_{\mathfrak{g},\mathfrak{n},\mathfrak{b}} (-1)^{\int_{\Sigma_\triangle} \mathfrak{b} \smile (d\mathfrak{n} + \mathfrak{g}_1 + \tilde{\mathfrak{f}}_1) + \mathfrak{f}_2 \smile \mathfrak{g}_2} |\mathfrak{g}\rangle\langle\mathfrak{g}|$$
$$= \frac{1}{2^{\chi(\Sigma_\triangle)}} \sum_{\mathfrak{g},\mathfrak{n}} \delta_{d\mathfrak{n},\mathfrak{g}_1 + \tilde{\mathfrak{f}}_1} (-1)^{\int_{\Sigma_\triangle} \mathfrak{f}_2 \cup \mathfrak{g}_2} |\mathfrak{g}\rangle\langle\mathfrak{g}|, \tag{167}$$

where $\tilde{\mathfrak{f}}_1$, which twists the cocycle condition on $\mathfrak{n}$, corresponds to the $\mathsf{Vec}_{\mathbb{Z}_2^{(1)}}$ lines, whereas $\mathfrak{f}_2$ corresponds to the usual Wilson lines. By analogy, the topological lines living on the surface operators $\mathcal{U}^{\mathbb{Z}_2^{(2)}}[\Sigma_\triangle]$ and $\mathcal{U}^{\mathbb{Z}_2^{(\text{diag.})}}[\Sigma_\triangle]$ form the fusion 1-categories $\mathsf{Vec}_{\mathbb{Z}_2^{(2)}} \boxtimes \mathsf{Rep}(\mathbb{Z}_2^{(1)})$ and $\mathsf{Vec}_{\mathbb{Z}_2^{(1)}} \boxtimes \mathsf{Rep}(\mathbb{Z}_2^{(\text{diag.})})$, respectively, where a networks of lines within these two operators take

the form

$$
\mathcal{U}^{\mathbb{Z}_2^{(2)}}(\mathfrak{f}_1,\tilde{\mathfrak{f}}_2)[\Sigma_\triangle] = \frac{1}{2^{\#(\Sigma_\triangle)}} \sum_{\mathfrak{g},\mathfrak{n},\mathfrak{b}} (-1)^{\int_{\Sigma_\triangle} \mathfrak{f}_1 \smile \mathfrak{g}_1 + \mathfrak{b} \smile (d\mathfrak{n}+\mathfrak{g}_2+\tilde{\mathfrak{f}}_2)} |\mathfrak{g}\rangle\langle\mathfrak{g}| \,,
$$

$$
\mathcal{U}^{\mathbb{Z}_2^{(\mathrm{diag.})}}(\mathfrak{f},\tilde{\mathfrak{f}})[\Sigma_\triangle] = \frac{1}{2^{\#(\Sigma_\triangle)}} \sum_{\mathfrak{g},\mathfrak{n},\mathfrak{b}} (-1)^{\int_{\Sigma_\triangle} \mathfrak{b} \smile (d\mathfrak{n}+(\mathfrak{g}_1+\mathfrak{g}_2)+\tilde{\mathfrak{f}})+\mathfrak{f} \smile (\mathfrak{g}_1+\mathfrak{g}_2)} |\mathfrak{g}\rangle\langle\mathfrak{g}| \,. \tag{168}
$$

Let us now describe the final symmetry surface operator in the fusion 2-category $2\mathrm{Rep}(\mathbb{Z}_2^2)$. It is that labelled by the $\mathsf{Vec}_{\mathbb{Z}_2^2}$-module category $\mathcal{N}(\mathbb{Z}_1,1) \cong \mathsf{Vec}_{\mathbb{Z}_2^2}$:

$$
\mathcal{U}^{\mathbb{Z}_2^2}[\Sigma_\triangle] = \frac{1}{2^{2\#(\Sigma_\triangle)}} \prod_{I=1}^{2} \sum_{\mathfrak{g}} \sum_{\mathfrak{n}_I,\mathfrak{b}_I} (-1)^{\int_{\Sigma_\triangle} \delta^{I,J} \mathfrak{b}_I \smile (d\mathfrak{n}_J+\mathfrak{g}_J)} |\mathfrak{g}\rangle\langle\mathfrak{g}| = \frac{1}{2\chi(\Sigma_\triangle)} \prod_{I=1}^{2} \sum_{\mathfrak{n}_I} \delta_{d\mathfrak{n}_I,\mathfrak{g}_I} |\mathfrak{g}\rangle\langle\mathfrak{g}| \,. \tag{169}
$$

Similarly, $\mathcal{U}^{\mathbb{Z}_2^2}[\Sigma_\triangle]$ can be defined with line operator insertions:

$$
\mathcal{U}^{\mathbb{Z}_2^2}(\tilde{\mathfrak{f}}_1,\tilde{\mathfrak{f}}_2)[\Sigma_\triangle] = \frac{1}{2^{2\#(\Sigma_\triangle)}} \prod_{I} \sum_{\mathfrak{g},\mathfrak{n}_I,\mathfrak{b}_I} (-1)^{\int_{\Sigma_\triangle} \delta^{I,J} \mathfrak{b}_I \smile (d\mathfrak{n}_J+\mathfrak{g}_J+\tilde{\mathfrak{f}}_J)} |\mathfrak{g}\rangle\langle\mathfrak{g}| \,. \tag{170}
$$

The commutation of $\mathcal{U}^{\mathbb{Z}_2^2}(\tilde{\mathfrak{f}}_1,\tilde{\mathfrak{f}}_2)[\Sigma_\triangle]$ with the Hamiltonian can be demonstrated as before. Having described all the surface operators associated with simple objects in $2\mathrm{Rep}(\mathbb{Z}_2^2)$, let now compute their fusion rules. For instance, we have

$$
\begin{aligned}
(\mathcal{U}^{\mathbb{Z}_2^{(1)}} \odot \mathcal{U}^{\mathbb{Z}_2^{(1)}})[\Sigma_\triangle] &= \frac{1}{2^{2\#(\Sigma_\triangle)}} \sum_{\substack{\mathfrak{g},\mathfrak{b},\mathfrak{n} \\ \mathfrak{g}',\mathfrak{b}',\mathfrak{n}'}} (-1)^{\int_{\Sigma_\triangle} \mathfrak{b} \smile (d\mathfrak{n}+\mathfrak{g}_1)+\mathfrak{b}' \smile (d\mathfrak{n}'+\mathfrak{g}_1')} |\mathfrak{g}\rangle\langle\mathfrak{g}|\mathfrak{g}'\rangle\langle\mathfrak{g}'| \\
&= \left( \frac{1}{2^{\#(\Sigma_\triangle)}} \sum_{\mathfrak{b}',\mathfrak{n}_+} (-1)^{\int_{\Sigma_\triangle} \mathfrak{b}' \smile d\mathfrak{n}_+} \right) \frac{1}{2^{\#(\Sigma_\triangle)}} \sum_{\mathfrak{g},\mathfrak{b}_+,\mathfrak{n}} (-1)^{\int_{\Sigma_\triangle} \mathfrak{b}_+ \smile (d\mathfrak{n}+\mathfrak{g}_1)} |\mathfrak{g}\rangle\langle\mathfrak{g}| \\
&= \mathcal{Z}_{2\mathrm{d}}[\Sigma_\triangle] \cdot \mathcal{U}^{\mathbb{Z}_2^{(1)}}[\Sigma_\triangle] \,,
\end{aligned} \tag{171}
$$

where in the second line, we have defined $\mathfrak{b}_+ = \mathfrak{b}+\mathfrak{b}'$ and $\mathfrak{n}_+ = \mathfrak{n}+\mathfrak{n}'$. The pre-factor $\mathcal{Z}_{2\mathrm{d}}[\Sigma_\triangle]$ in the fusion rule outcome is the partition function of the pure two-dimensional $\mathbb{Z}_2$ gauge theory on $\Sigma_\triangle$ as in sec. 2 and app. A. In the case where $\Sigma$ is a two-torus, we recover exactly the fusion structure of $2\mathrm{Rep}(\mathbb{Z}_2^2)$ according to which

$$
\mathsf{Vec}_{\mathbb{Z}_2^{(1)}} \odot \mathsf{Vec}_{\mathbb{Z}_2^{(1)}} \cong \mathsf{Vec}_{\mathbb{Z}_2^{(1)}} \boxplus \mathsf{Vec}_{\mathbb{Z}_2^{(1)}} \,. \tag{172}
$$

The remaining fusion rules can be computed analogously:

$$
\left( \mathcal{U}^{\mathbb{Z}_2^{(I)}} \odot \mathcal{U}^{\mathbb{Z}_2^{(J)}} \right)[\Sigma_\triangle] = \begin{cases} \mathcal{Z}_{2\mathrm{d}}[\Sigma_\triangle] \cdot \mathcal{U}^{\mathbb{Z}_2^{(I)}}[\Sigma_\triangle], & \text{if } I=J \,, \\ \mathcal{U}^{\mathbb{Z}_2^2}[\Sigma_\triangle], & \text{otherwise,} \end{cases} \tag{173}
$$

where $I,J \in \{1,2,\mathrm{diag.}\}$. Finally, the fusion rules between the symmetry operator $\mathcal{U}^{\mathbb{Z}_2^2}[\Sigma_\triangle]$ and the three other non-invertible surfaces are given by

$$
\left( \mathcal{U}^{\mathbb{Z}_2^2} \odot \mathcal{U}^{\mathbb{Z}_2^I} \right)[\Sigma_\triangle] = \left( \mathcal{U}^{\mathbb{Z}_2^J} \odot \mathcal{U}^{\mathbb{Z}_2^2} \right)[\Sigma_\triangle] = \mathcal{Z}_{2\mathrm{d}}[\Sigma] \times \mathcal{U}^{\mathbb{Z}_2^2}[\Sigma_\triangle] \,. \tag{174}
$$

Finally, fusing $\mathcal{U}^{\mathbb{Z}_2^2}$ with itself yields

$$
\left( \mathcal{U}^{\mathbb{Z}_2^2} \odot \mathcal{U}^{\mathbb{Z}_2^2} \right)[\Sigma_\triangle] = (\mathcal{Z}_{2\mathrm{d}}[\Sigma_\triangle])^2 \cdot \mathcal{U}^{\mathbb{Z}_2^2}[\Sigma_\triangle] \,, \tag{175}
$$

where the coefficient $(\mathcal{Z}_{2d}[\Sigma_\triangle])^2$ amounts to the partition function of the pure two-dimensional $\mathbb{Z}_2^2$ topological gauge theory on $\Sigma_\triangle$.

In order to conclude our analysis of the symmetry structure of (144) as encoded into $2\mathsf{Rep}(\mathbb{Z}_2^2)$, we are left to consider topological lines between distinct surfaces as well as the corresponding composition rules. It largely mimics the case presented in sec. 2. A topological surface operator can be defined on a triangulation of the form $\Sigma_\triangle = (\Sigma_\triangle \backslash \Xi_\triangle) \sqcup_{\partial \Xi_\triangle} \Xi_\triangle$, which locally looks like $\mathcal{U}^{\mathbb{Z}_2^{(I)}}$ and $\mathcal{U}^{\mathbb{Z}_2^{(J)}}$ in the regions $\Sigma_\triangle \backslash \Xi_\triangle$ and $\Xi_\triangle$, respectively. Such an operator has the form

$$\mathcal{U}^{\mathbb{Z}_2^{(I)}, \mathbb{Z}_2^{(J)}}[\Sigma_\triangle \backslash \Xi_\triangle, \Xi_\triangle] = \frac{1}{2^{\#(\Sigma_\triangle)}} \sum_{\substack{\mathfrak{g},\mathfrak{n},\mathfrak{b} \\ \tilde{\mathfrak{n}},\tilde{\mathfrak{b}}}} (-1)^{\int_{\Sigma_\triangle \backslash \Xi_\triangle} \mathfrak{b} \smile (d\mathfrak{n} + \mathfrak{g}_I) + \int_{\Xi_\triangle} \tilde{\mathfrak{b}} \smile (d\tilde{\mathfrak{n}} + \mathfrak{g}_J)} |\mathfrak{g}\rangle\langle\mathfrak{g}|, \qquad (176)$$

where $I, J \in \{1, 2, \mathrm{diag.}\}$ and $\mathfrak{g}_{\mathrm{diag.}} := \mathfrak{g}_1 + \mathfrak{g}_2$. Moreover, we imposed in the previous equation Dirichlet boundary conditions $\mathfrak{b}[\partial \Xi_\triangle] = \tilde{\mathfrak{b}}[\partial \Xi_\triangle] = 0$ along the interface. Similarly, one may define a topological surface operator that interpolates between $\mathcal{U}^{\mathbb{Z}_2^2}$ and $\mathcal{U}^{\mathbb{Z}_2^{(I)}}$ by defining $\mathcal{U}^{\mathbb{Z}_2^2}$ on $\Sigma_\triangle \backslash \Xi_\triangle$, $\mathcal{U}^{\mathbb{Z}_2^{(I)}}$ on $\Xi_\triangle$, and imposing suitable Dirichlet boundary conditions along the interface $\partial \Xi_\triangle$.

Let us now compute the composition rules between topological interfaces separating regions with locally distinct symmetry operators. To do so, we consider a setup closely resembling that of sec. 2. Let $\Xi_\triangle$ be a thin annular strip of single lattice spacing width supporting a surface operator $\mathcal{U}^{\mathbb{Z}_2^{(J)}}$, while the rest of the lattice $\Sigma_\triangle \backslash \Xi_\triangle$ supports $\mathcal{U}^{\mathbb{Z}_2^{(I)}}$. We denote the left and right boundaries of $\Xi_\triangle$ by $\partial_L \Xi_\triangle$ and $\partial_R \Xi_\triangle$, respectively. The corresponding composition of lines is then given by the operator

$$\bigoplus_{\mathfrak{f}} \mathcal{U}^{\mathbb{Z}_2^{(I)}}(\mathfrak{f})[\Sigma_\triangle], \qquad (177)$$

where the sum is over the four simple topological lines of $\mathcal{U}^{\mathbb{Z}_2^{(I)}}[\Sigma_\triangle]$ traversing the (relative) homology cycle of $\Xi_\triangle$ with Dirichlet conditions $\mathfrak{b}[\partial_L \Xi_\triangle] = \mathfrak{b}[\partial_R \Xi_\triangle] = 0$ imposed. These fusion rules are reminiscent of the $\mathbb{Z}_2^2$ *Tambara-Yamagami* fusion category.

# 5 Example: transverse-field $\mathcal{S}_3$-Ising model

*In this section, we consider the higher-categorical symmetry structures of the models obtained by gauging various sub-symmetries of the transverse-field $\mathcal{S}_3$-Ising model. We shall focus on features specific to dealing with a non-Abelian group.*

## 5.1 Symmetric Hamiltonian

For our final series of examples, we consider a transverse-field Ising model with a non-Abelian symmetry group, namely the symmetric group $\mathcal{S}_3$ of degree 3. In sec. 3.8, we explained how to perform the gauging of various sub-symmetries of this model for an arbitrary group. Moreover, we elucidated there the symmetry structures of the resulting models in terms of fusion 2-categories of higher representations of groups, and categorifications thereof. Although the construction of sec. 3.5 provides a general recipe to realise on the lattice operators commuting with dual Hamiltonians, it remains somewhat formal. The goal of this section is to describe these symmetry operators more explicitly in the spirit of sec. 4.

Let us begin by reviewing the group structure of $\mathcal{S}_3$. The permutation group $\mathcal{S}_3$ is the group with presentation

$$\mathcal{S}_3 = \langle r, s \,|\, r^2 = s^3 = (rs)^2 = \mathbb{1} \rangle. \tag{178}$$

Both the cyclic groups $\mathbb{Z}_2 = \langle r \,|\, r^2 = \mathbb{1}\rangle$ and $\mathbb{Z}_3 = \langle s \,|\, s^3 = \mathbb{1}\rangle$ are subgroups of $\mathcal{S}_3$. Due to the action of $\mathbb{Z}_2$ on $\mathbb{Z}_3$ given by $\phi_- : \mathbb{Z}_2 \to \mathrm{Aut}(\mathbb{Z}_3)$ such that $\phi_r(s) = s^2$, we have an isomorphism $\mathcal{S}_3 \simeq \mathbb{Z}_2 \ltimes_\phi \mathbb{Z}_3$. Therefore, it is a group of the form considered in sec. 3.8.

The microscopic Hilbert space of the transverse-field $\mathcal{S}_3$-model is given by the tensor product $\bigotimes_\mathsf{v} \mathbb{C}[\mathcal{S}_3] \ni |\mathfrak{m}\rangle$, where $\mathfrak{m}$ is an assignment of group elements in $\mathcal{S}_3$ to every vertex of $\Sigma_\triangle$. Due to the semi-direct product structure of $\mathcal{S}_3$, the local Hilbert space can be rather spanned by states $|\mathfrak{m}[\mathsf{v}]\rangle \equiv |\varpi_{\mathbb{Z}_2}(\mathfrak{m}[\mathsf{v}]), \varpi_{\mathbb{Z}_3}(\mathfrak{m}[\mathsf{v}])\rangle$, that is a pair of qubit and qutrit degrees of freedom at every vertex. Given the above, it is useful to define the following operators

$$\sigma^x = \begin{pmatrix} 0 & 1 \\ 1 & 0 \end{pmatrix}, \quad \sigma^z = \begin{pmatrix} 1 & 0 \\ 0 & -1 \end{pmatrix}, \quad \Sigma^x = \begin{pmatrix} 0 & 0 & 1 \\ 1 & 0 & 0 \\ 0 & 1 & 0 \end{pmatrix},$$

$$\Sigma^z = \begin{pmatrix} 1 & 0 & 0 \\ 0 & \omega & 0 \\ 0 & 0 & \omega^2 \end{pmatrix}, \quad \Gamma = \begin{pmatrix} 1 & 0 & 0 \\ 0 & 0 & 1 \\ 0 & 1 & 0 \end{pmatrix}. \tag{179}$$

The $\sigma$ matrices satisfy the Pauli algebra $\sigma^x \sigma^z = -\sigma^z \sigma^x$ as before, while the $\Sigma$ matrices represent the $\mathbb{Z}_3$ clock and shift operators, which satisfy $\Sigma^z \Sigma^x = \omega \Sigma^x \Sigma^z$ and $(\Sigma^x)^3 = (\Sigma^z)^3 = \mathrm{id}$ with $\omega = \exp(\frac{2\pi i}{3})$. Additionally, the operator $\Gamma$ implements the action of $\mathbb{Z}_2$ on $\mathbb{Z}_3$ by automorphisms. This operator satisfies the relations $\Gamma \Sigma^x \Gamma = \Sigma^{x\dagger}$ and $\Gamma \Sigma^z \Gamma = \Sigma^{z\dagger}$. We work in the basis such that

$$(\mathrm{id} \otimes \Sigma^z_\mathsf{v})|\mathfrak{m}[\mathsf{v}]\rangle = \omega^{\varpi_{\mathbb{Z}_3}(\mathfrak{m}[\mathsf{v}])}|\mathfrak{m}[\mathsf{v}]\rangle,$$
$$(\sigma^z_\mathsf{v} \otimes \mathrm{id})|\mathfrak{m}[\mathsf{v}]\rangle = (-1)^{\varpi_{\mathbb{Z}_2}(\mathfrak{m}[\mathsf{v}])}|\mathfrak{m}[\mathsf{v}]\rangle, \tag{180}$$

effectively identifying group elements in $\mathbb{Z}_2$ with $\{0, 1\}$ and those in $\mathbb{Z}_3$ with $\{0, 1, 2\}$. The $L^g_\mathsf{v}$ operators which act by left multiplication (see e.g. below eq. (120)) have the following explicit form in this basis:

$$L^r_\mathsf{v} : |\mathfrak{m}[\mathsf{v}]\rangle \mapsto (\sigma^x \otimes \Gamma)_\mathsf{v}|\mathfrak{m}[\mathsf{v}]\rangle = |r \cdot \mathfrak{m}[\mathsf{v}]\rangle,$$
$$L^s_\mathsf{v} : |\mathfrak{m}[\mathsf{v}]\rangle \mapsto (\mathrm{id} \otimes \Sigma^x)_\mathsf{v}|\mathfrak{m}[\mathsf{v}]\rangle = |s \cdot \mathfrak{m}[\mathsf{v}]\rangle. \tag{181}$$

The qubit and qutrit degrees of freedom are subject to the $2\mathsf{Vec}_{\mathcal{S}_3}$-symmetric Hamiltonian whose expression we reproduce below:

$$\mathbb{H}^{2\mathsf{Vec}_G} = -J\sum_e \mathbb{\Pi}^{\mathbb{1}_{\mathcal{S}_3}}_{s(e),t(e)} - \frac{J\kappa}{6} \sum_\mathsf{v} \sum_{g \in G} L^g_\mathsf{v}. \tag{182}$$

Both terms appearing in this Hamiltonian can be rewritten more explicitly in terms of the matrices introduced above. On the one hand, we have

$$\mathbb{\Pi}^{\mathbb{1}_{\mathcal{S}_3}}_{s(e),t(e)} = \frac{1}{6}\Big(\mathrm{id} + \sigma^z_{s(e)}\sigma^z_{t(e)}\Big) \otimes \Big(\mathrm{id} + \Sigma^z_{s(e)}(\Sigma^z_{t(e)})^\dagger + (\Sigma^z_{s(e)})^\dagger \Sigma^z_{t(e)}\Big), \tag{183}$$

which implicitly makes use of the fact that $1 + \omega + \bar\omega = 0$. Such a term is analogous to the ferromagnetic term in the $\mathbb{Z}_2$-symmetric transverse field Ising model. It energetically favors homogenous configurations in the computational basis. In the limit $\kappa \to 0$, the ground state spontaneously breaks the $\mathcal{S}_3$ global symmetry with $\mathfrak{m}[\mathsf{v}] = \mathfrak{m}_0$, for all $\mathsf{v}$, and there are $|\mathcal{S}_3|$ ground states associated with different choices of $\mathfrak{m}_0 \in \mathcal{S}_3$. On the other hand, the term

proportional to $\kappa$ is a combination of operators $L_v^g$ which act on the basis by left multiplication. The linear combination of $L_v^g$ operators has the following explicit form:

$$\frac{1}{6}\sum_{g\in\mathcal{S}_3}L_v^g = \frac{1}{6}\sum_{g\in\mathcal{S}_3}(\sigma^x\otimes\Gamma)^{\varpi_{\mathbb{Z}_2}(g)}(\mathrm{id}\otimes\Sigma^x)^{\varpi_{\mathbb{Z}_3}(g)}, \tag{184}$$

which acts as a projector onto the one-dimensional subspace of $\mathbb{C}[\mathcal{S}_3]$ (at the vertex v) transforming in the trivial representation of $\mathcal{S}_3$. This term is analogous to the paramagnetic term in the $\mathbb{Z}_2$-symmetric transverse field Ising model and favours a unique ground state that preserves the full $\mathcal{S}_3$ symmetry. One can readily check that this model has a global 0-form $\mathcal{S}_3$ symmetry implemented by surface operators

$$\mathcal{O}^g = \prod_v R_v^g, \tag{185}$$

where $R_v^g$ acts on the basis state at vertex v by right multiplication. These operators satisfy the fusion rules provided by the multiplication rules in $\mathcal{S}_3$, i.e., $\mathcal{O}^{g_1}\odot\mathcal{O}^{g_2} = \mathcal{O}^{g_1g_2}$. In order to rewrite operators $R_v^g$ more explicitly, it is convenient to introduce the following *controlled* gate:

$$\begin{aligned}
c\Sigma^x : \mathbb{C}^2\otimes\mathbb{C}^3 &\to \mathbb{C}^2\otimes\mathbb{C}^3, \\
: |q,l\rangle &\mapsto \big(\mathrm{id}\otimes(\Sigma^x)^{1+q}\big)|q,l\rangle,
\end{aligned} \tag{186}$$

where the qubit plays the role of the control. These controlled gates satisfy in particular the following commutation relation

$$c\Sigma^x(\sigma^x\otimes\Gamma) = (\sigma^x\otimes\Gamma)c\Sigma^x, \tag{187}$$

and adaptations thereof. The explicit action of right multiplication operators $R_v^g$ now reads:

$$\begin{aligned}
R_v^r : |\mathfrak{m}[v]\rangle &\mapsto (\sigma^x\otimes\mathrm{id})_v|\mathfrak{m}[v]\rangle = |\mathfrak{m}[v]\cdot r\rangle, \\
R_v^s : |\mathfrak{m}[v]\rangle &\mapsto (c\Sigma^x)_v^\dagger|\mathfrak{m}[v]\rangle = |\mathfrak{m}[v]\cdot s^2\rangle.
\end{aligned} \tag{188}$$

Given this action, commutation of $\mathcal{O}^g$ with the Hamiltonian is ensured by the various commutation relations listed above.

## 5.2  Gauging sub-symmetries

Given a finite group isomorphic to a semi-direct product, we explained in sec. 3.8 how to obtain three dual Hamiltonians. In the present context, these are obtained by gauging the $\mathcal{S}_3$, $\mathbb{Z}_3$ and $\mathbb{Z}_2$ sub-symmetries of the $\mathcal{S}_3$-symmetric Hamiltonian, respectively. In the language of sec. 3.8, these amount to choosing the $2\mathrm{Vec}_{\mathcal{S}_3}$-module 2-categories $\mathcal{M}(\mathcal{S}_3,1)\cong 2\mathrm{Vec}$, $\mathcal{M}(\mathbb{Z}_3,1)\cong 2\mathrm{Vec}_{\mathbb{Z}_2}$ and $\mathcal{M}(\mathbb{Z}_2,1)\cong 2\mathrm{Vec}_{\mathcal{S}_3/\mathbb{Z}_2}$, respectively. In preparation for the analysis of the symmetry structures, we provide below more explicit expressions for the terms appearing in the definitions of these Hamiltonians.

• Let us first consider gauging the full $\mathcal{S}_3$ symmetry. The resulting Hamiltonian was found in eq. (123) to be of the form

$$\mathbb{H}^{2\mathrm{Vec}} = -J\sum_e \mathbb{\Pi}_e^{\mathbb{1}_{\mathcal{S}_3}} - J\kappa\sum_v \mathbb{A}_v. \tag{189}$$

The edge term explicitly reads

$$\mathbb{\Pi}_e^{\mathbb{1}_{\mathcal{S}_3}} = \frac{1}{6}\big(\mathrm{id}+\sigma_e^z\big)\otimes\big(\mathrm{id}+\Sigma_e^z+(\Sigma_e^z)^\dagger\big), \tag{190}$$

whereas the contribution of the generators of $\mathcal{S}_3$ to the vertex term $\mathbb{A}_{\mathsf{v}} = \frac{1}{6}\sum_{g\in\mathcal{S}_3}\mathbb{A}_{\mathsf{v}}^g$ can be depicted as

$$
\mathbb{A}_{\mathsf{v}}^r \equiv \left\langle\begin{array}{c}\sigma^x\Gamma \;\; \sigma^x\Gamma \\ \sigma^x \;\;\longleftrightarrow\;\; \sigma^x\Gamma \\ \sigma^x \;\; \sigma^x\end{array}\right\rangle , \quad \mathbb{A}_{\mathsf{v}}^s \equiv \left\langle\begin{array}{c}\Sigma^x \;\; \Sigma^x \\ \mathrm{c}\Sigma^{x\dagger}\;\longleftrightarrow\;\Sigma^x \\ \mathrm{c}\Sigma^{x\dagger}\,\mathrm{c}\Sigma^{x\dagger}\end{array}\right\rangle ,
\tag{191}
$$

where we omitted $\otimes$ symbols for convenience. The operators $\mathbb{A}_{\mathsf{v}}^g$ for the remaining $g \in \mathcal{S}_3$ can be obtained by suitably composing $\mathbb{A}_{\mathsf{v}}^r$ and $\mathbb{A}_{\mathsf{v}}^s$. The model (189) describes an $\mathcal{S}_3$ lattice gauge theory. The first term suppresses the $\mathcal{S}_3$ fluctuations. In the limit $\kappa \to 0$, the model is in the *confined* phase, which is the dual analogue of the symmetry-broken phase in the pre-gauged model. Instead, the second term is responsible for gauge fluctuations. In the $\kappa \to \infty$ limit, the model is in the *deconfined* phase whose renormalisation group fixed point is provided by the Hamiltonian realisation of $\mathcal{S}_3$ Dijkgraaf-Witten theory with trivial cohomological twist [51,52].

• In the same vein, let us now consider gauging the $\mathbb{Z}_3$ sub-symmetry. The resulting Hamiltonian was found in eq. (134) to be of the form

$$
\mathbb{H}^{2\mathsf{Vec}_{\mathbb{Z}_2}} = -J\sum_{\mathsf{e}} \mathbb{\Pi}_{\mathsf{e}}^{0_{\mathbb{Z}_2},0_{\mathbb{Z}_3}} - J\kappa \sum_{\mathsf{v}} {}^{\phi}\mathbb{A}_{\mathsf{v}} .
\tag{192}
$$

The edge term explicitly reads

$$
\mathbb{\Pi}_{\mathsf{e}}^{0_{\mathbb{Z}_2},0_{\mathbb{Z}_3}} = \frac{1}{6}\Big(\mathrm{id} + \sigma_{\mathsf{s(e)}}^z \sigma_{\mathsf{t(e)}}^z\Big) \otimes \Big(\mathrm{id} + \Sigma_{\mathsf{e}}^z + (\Sigma_{\mathsf{e}}^z)^\dagger\Big) .
\tag{193}
$$

In order to rewrite the vertex term more explicitly, we require the controlled gates introduced previously. We shall apply here these gates between a qutrit assigned to an edge and a qubit assigned to a vertex. It is convenient to graphically depict such controlled gates by means of a dotted line connecting a control qubit identified by 'c' and a target qutrit. The contribution of the generators of $\mathcal{S}_3$ to the vertex term ${}^{\phi}\mathbb{A}_{\mathsf{v}} = \frac{1}{6}\sum_{g\in\mathcal{S}_3}{}^{\phi}\mathbb{A}_{\mathsf{v}}^g$ can now be depicted as

$$
{}^{\phi}\mathbb{A}_{\mathsf{v}}^r = \sigma_{\mathsf{v}}^x \equiv \left\langle\;\; \sigma^x \;\;\right\rangle , \quad {}^{\phi}\mathbb{A}_{\mathsf{v}}^s = \prod_{\mathsf{e}\leftarrow\mathsf{v}}\mathrm{c}\Sigma_{\mathsf{e}}^x \prod_{\mathsf{e}\to\mathsf{v}}\big(\mathrm{c}\Sigma_{\mathsf{e}}^x\big)^\dagger \equiv \left\langle\begin{array}{c}\Sigma^x \;\; \Sigma^x \\ \Sigma^{x\dagger}\;\longleftrightarrow\;\Sigma^x \\ \Sigma^{x\dagger}\;\Sigma^{x\dagger}\end{array}\right\rangle ,
\tag{194}
$$

where all the gates on the r.h.s. are controlled by the qubit living at the vertex $\mathsf{v}$ the operator acts on. The model (192) describes a $\mathbb{Z}_3$ lattice gauge theory coupled to a $\mathbb{Z}_2$-Ising matter model. In the limit $\kappa \to 0$, the $\mathbb{Z}_3$ gauge sector is in the confined phase while the $\mathbb{Z}_2$ matter sector is in the ferromagnetic phase. Conversely, in the $\kappa \to \infty$ limit, the gauge sector is in the deconfined phase while the matter sector is in the paramagnetic phase. In this limit the model describes up to a unitary (see footnote 20) a $\mathbb{Z}_3$ topological gauge theory enriched by a global $\mathbb{Z}_2$ symmetry [115,117,119].

• Finally, let us consider gauging the $\mathbb{Z}_2$ sub-symmetry. The resulting Hamiltonian was found in eq. (140) to be of the form

$$
\mathbb{H}^{2\mathsf{Vec}_{\mathcal{S}_3/\mathbb{Z}_2}} = -J\sum_{\mathsf{e}} \mathbb{\Pi}_{\mathsf{e}}^{0_{\mathbb{Z}_3},0_{\mathbb{Z}_2}} - J\kappa \sum_{\mathsf{v}} {}^{\phi}\widetilde{\mathbb{A}}_{\mathsf{v}} .
\tag{195}
$$

The edge term explicitly reads

$$
\mathbb{\Pi}_{\mathsf{e}}^{0_{\mathbb{Z}_3},0_{\mathbb{Z}_2}} = \frac{1}{6}\Big(\mathrm{id} + \sigma_{\mathsf{e}}^z\Big) \otimes \Big(\mathrm{id} + \Sigma_{\mathsf{s(e)}}^z (\Sigma_{\mathsf{t(e)}}^z)^\dagger + (\Sigma_{\mathsf{s(e)}}^z)^\dagger \Sigma_{\mathsf{t(e)}}^z\Big) ,
\tag{196}
$$

whereas the contribution of the generators of $\mathcal{S}_3$ to the vertex term ${}^{\phi}\widetilde{\mathbb{A}}_{\mathsf{v}} = \frac{1}{6}\sum_{g\in\mathcal{S}_3}{}^{\phi}\widetilde{\mathbb{A}}_{\mathsf{v}}^g$ can be depicted as

$$
{}^{\phi}\widetilde{\mathbb{A}}_{\mathsf{v}}^s = \left\langle\!\!\!\!\!\!\!\!\!\!\!\! \Sigma^x \!\!\!\!\!\!\!\!\!\!\!\!\right\rangle , \quad {}^{\phi}\widetilde{\mathbb{A}}_{\mathsf{v}}^r = \left\langle\!\!\!\!\!\!\!\! \begin{array}{c} \sigma^x \quad \sigma^x \\ \sigma^x - \Gamma - \sigma^x \\ \sigma^x \quad \sigma^x \end{array} \!\!\!\!\!\!\!\!\right\rangle .
\tag{197}
$$

The two phases of this Hamiltonian can be interpreted in the same vein as for the other models. Having described the models (189), (192) and (195), which are obtained by gauging the different subgroups of $\mathcal{S}_3$, we detail below the symmetry structures corresponding to each of these (partially) gauged models.

## 5.3 $2\mathsf{Rep}(\mathcal{S}_3)$ symmetry

Let us study the symmetry structure of Hamiltonian (189) acting on a Hilbert space spanned by states $|\mathfrak{g}\rangle \in \bigotimes_{\mathsf{e}}\mathbb{C}[G]$, where $\mathfrak{g} \in Z^1(\Sigma_{\triangle}, G)$. Following the general discussions in sec. 3.2 and 3.8, we know that Hamiltonian (189) must have a symmetry structure embodying the fusion 2-category $2\mathsf{Rep}(\mathcal{S}_3)$ of 2-representations of $\mathcal{S}_3$. In particular, this means that (189) hosts topological surfaces associated with simple objects in $2\mathsf{Rep}(\mathcal{S}_3)$. Recall from sec. 3.6 that simple objects in $2\mathsf{Rep}(\mathcal{S}_3)$ are provided by indecomposable $\mathsf{Vec}_{\mathcal{S}_3}$-module categories $\mathcal{N}(B,\psi)$, which are conveniently labelled by tuples $(B,\psi)$ consisting of a subgroup $B \subseteq \mathcal{S}_3$ and a 2-cocycle $\psi$ in $H^2(B,\mathsf{U}(1))$. Since $H^2(B,\mathsf{U}(1))$ is trivial for any $B \subseteq \mathcal{S}_3$, we count four simple objects in $2\mathsf{Rep}(\mathcal{S}_3)$ associated with each subgroup of $\mathcal{S}_3$, namely $\mathcal{N}(\mathcal{S}_3,1) \cong \mathsf{Vec}$, $\mathcal{N}(\mathbb{Z}_3,1) \cong \mathsf{Vec}_{\mathbb{Z}_2}$, $\mathcal{N}(\mathbb{Z}_2,1) \cong \mathsf{Vec}_{\mathcal{S}_3/\mathbb{Z}_2}$ and $\mathcal{N}(\mathbb{Z}_1,1) \cong \mathsf{Vec}_{\mathcal{S}_3}$. We provided in (107) a general formula to construct the corresponding surface operators, but let us unpack it further here in the spirit of sec. 2.

Generally speaking, defining topological surfaces associated with simple objects in $2\mathsf{Rep}(\mathcal{S}_3)$ requires introducing virtual degrees of freedom $\mathfrak{n}[\mathsf{v}]$ at every vertex $\mathsf{v} \subset \Sigma_{\triangle}$, whose configuration space is given by the set of isomorphism classes of simple objects in the corresponding $\mathsf{Vec}_{\mathcal{S}_3}$-module category. Concretely, given the topological surface associated with $\mathcal{N}(B,1)$, virtual degrees of freedom are valued in the collection $G/B$ of left cosets. These virtual degrees of freedom are then coupled to the physical degrees of freedom via the conditions spelt out below (107) involving the $\mathsf{Vec}_{\mathcal{S}_3}$-module structure of $\mathcal{N}(B,1)$, which we recall is given by the natural action of $\mathcal{S}_3$ on the left cosets.

Suppose for instance that the subgroup $B$ is the whole group $\mathcal{S}_3$. There is a single left coset in $\mathcal{S}_3/\mathcal{S}_3 \simeq \mathbb{Z}_1$, namely $\mathcal{S}_3$ itself, on which $\mathcal{S}_3$ acts invariantly. It follows that the resulting operator acts as the identity. We denote it by $\mathcal{U}^{\text{triv.}}$ in accordance with sec. 2 and 4:

$$
\mathcal{U}^{\text{triv.}} = \prod_{\mathsf{e}} \mathrm{id}_{\mathsf{e}} .
\tag{198}
$$

This topological surface, which is associated with the $\mathsf{Vec}_{\mathcal{S}_3}$-module category $\mathsf{Vec}$, is the only invertible one for this model.

Let us now consider non-invertible topological surfaces. We first focus on that associated with the $\mathsf{Vec}_{\mathcal{S}_3}$-module category $\mathcal{N}(\mathbb{Z}_3,1) \cong \mathsf{Vec}_{\mathbb{Z}_2}$. By definition, simple objects in $\mathcal{N}(\mathbb{Z}_3,1)$ are left cosets in $\mathcal{S}_3/\mathbb{Z}_3 \simeq \big\{\{1,s,s^2\},\{r,sr,s^2r\}\big\} \simeq \mathbb{Z}_2$. The corresponding surface operator amounts to summing over configurations $\mathfrak{n} \in C^0(\Sigma_{\triangle},\mathbb{Z}_2)$ of virtual degrees of freedom, which are coupled to the physical degrees of freedom by imposing conditions $\mathfrak{n}[\mathsf{v}_1] = \mathbb{C}_{\mathfrak{g}[\mathsf{v}_1\mathsf{v}_2]} \triangleright \mathfrak{n}[\mathsf{v}_2] = \varpi_{\mathbb{Z}_2}(\mathfrak{g}[\mathsf{v}_1\mathsf{v}_2]) + \mathfrak{n}[\mathsf{v}_2]$ at every edge $(\mathsf{v}_1\mathsf{v}_2) \subset \Sigma_{\triangle}$. These conditions

can be enforced explicitly by introducing Lagrange multiplier $\mathfrak{b} \in C^1(\Sigma_\triangle, \mathbb{Z}_2)$ as follows:

$$
\begin{aligned}
\mathcal{U}^{\mathbb{Z}_2}[\Sigma_\triangle] &= \frac{1}{2^{\#(\Sigma_\triangle)}} \sum_{\mathfrak{g}, \mathfrak{n}, \mathfrak{b}} \exp\left( \pi i \int_{\Sigma_\triangle} \mathfrak{b} \smile \left( d\mathfrak{n} - \varpi_{\mathbb{Z}_2}(\mathfrak{g}) \right) \right) |\mathfrak{g}\rangle\langle\mathfrak{g}| \\
&= \frac{1}{2^{\chi(\Sigma_\triangle)}} \sum_{\mathfrak{g}, \mathfrak{n}} \prod_{(v_1 v_2) \subset \Sigma_\triangle} \delta_{\mathbb{C}_{\mathfrak{g}[v_1 v_2]} \triangleright \mathfrak{n}[v_2], \mathfrak{n}[v_1]} |\mathfrak{g}\rangle\langle\mathfrak{g}|,
\end{aligned}
\tag{199}
$$

where $(d\mathfrak{n})[v_1 v_2] = \mathfrak{n}[v_1] - \mathfrak{n}[v_2]$.

The topological surface associated with the $\mathsf{Vec}_{\mathcal{S}_3}$-module category $\mathcal{N}(\mathbb{Z}_2, 1) \cong \mathsf{Vec}_{\mathcal{S}_3/\mathbb{Z}_2}$ is constructed similarly. By definition, simple objects in $\mathcal{N}(\mathbb{Z}_2, 1)$ are provided by left cosets in $\mathcal{S}_3/\mathbb{Z}_2 \simeq \{\{1, r\}, \{r, sr\}, \{s^2, s^2 r\}\}$. The non-normal subgroup $\mathbb{Z}_2 \subset \mathcal{S}_3$ acts trivially on $\mathcal{S}_3/\mathbb{Z}_3$, while the remaining elements permute the elements in $\mathcal{S}_3/\mathbb{Z}_2$. The corresponding surface operator amounts to summing over configurations $\mathfrak{n} \in C^0(\Sigma_\triangle, \mathbb{Z}_3)$ of virtual degrees of freedom, which are coupled to the physical degrees of freedom by imposing conditions $\mathfrak{n}[v_1] = \mathbb{C}_{\mathfrak{g}[v_1 v_2]} \triangleright \mathfrak{n}[v_2] = \varpi_{\mathbb{Z}_3}(\mathfrak{g}[v_1 v_2]) + \phi_{\varpi_{\mathbb{Z}_2}(\mathfrak{g}[v_1 v_2])}(\mathfrak{n}[v_2])$ at every edge $(v_1 v_2) \subset \Sigma_\triangle$, where we are using that $\mathcal{S}_3/\mathbb{Z}_2 \simeq \mathbb{Z}_3$ as a set. These conditions can be enforced explicitly by introducing Lagrange multiplier $\mathfrak{b} \in C^1(\Sigma_\triangle, \mathbb{Z}_3)$ as follows:

$$
\begin{aligned}
\mathcal{U}^{\mathcal{S}_3/\mathbb{Z}_2}[\Sigma_\triangle] &= \frac{1}{3^{\#(\Sigma_\triangle)}} \sum_{\mathfrak{g}, \mathfrak{n}, \mathfrak{b}} \exp\left( \frac{2\pi i}{3} \int_{\Sigma_\triangle} \mathfrak{b} \smile \left( d_\mathfrak{g} \mathfrak{n} - \varpi_{\mathbb{Z}_3}(\mathfrak{g}) \right) \right) |\mathfrak{g}\rangle\langle\mathfrak{g}| \\
&= \frac{1}{3^{\chi(\Sigma_\triangle)}} \sum_{\mathfrak{g}, \mathfrak{n}} \prod_{(v_1 v_2) \subset \Sigma_\triangle} \delta_{\mathbb{C}_{\mathfrak{g}[v_1 v_2]} \triangleright \mathfrak{n}[v_2], \mathfrak{n}[v_1]} |\mathfrak{g}\rangle\langle\mathfrak{g}|,
\end{aligned}
\tag{200}
$$

where we have used the twisted differential

$$
(d_\mathfrak{g} \mathfrak{n})[v_1 v_2] := \mathfrak{n}[v_1] - \phi_{\varpi_{\mathbb{Z}_2}(\mathfrak{g}[v_1 v_2])}(\mathfrak{n}[v_2]).
\tag{201}
$$

Finally, the remaining topological surface associated with the $\mathsf{Vec}_{\mathcal{S}_3}$-module category $\mathcal{N}(\mathbb{Z}_1, 1) \cong \mathsf{Vec}_{\mathcal{S}_3}$ can be simply expressed as

$$
\mathcal{U}^{\mathcal{S}_3}[\Sigma_\triangle] = \frac{1}{|\mathcal{S}_3|^{\chi(\Sigma_\triangle)}} \sum_{\mathfrak{g}, \mathfrak{n}} \prod_{(v_1 v_2) \subset \Sigma_\triangle} \delta_{\mathfrak{g}[v_1 v_2] \mathfrak{n}[v_2], \mathfrak{n}[v_1]} |\mathfrak{g}\rangle\langle\mathfrak{g}|.
\tag{202}
$$

Let us now briefly confirm that all these surface operators do commute with (189), and are thus part of the symmetry structure of the model. Firstly, edge terms $\mathbb{T}_\mathsf{e}^{\mathbb{1}_{\mathcal{S}_3}}$ straightforwardly commute with all the topological surfaces since all the operators are diagonal in the chosen computational basis. Vertex operators $\mathbb{A}_\mathsf{v}$ perform averagings over the group of gauge transformations. An arbitrary combination of gauge transformations indexed by an assignment $\mathfrak{x} \in C^0(\Sigma_\triangle, \mathcal{S}_3)$ acts as $|\mathfrak{g}\rangle \mapsto |{}^\mathfrak{x}\mathfrak{g}\rangle$, where ${}^\mathfrak{x}\mathfrak{g}[\mathsf{e}] = \mathfrak{x}[\mathsf{s}(\mathsf{e})] \cdot \mathfrak{g}[\mathsf{e}] \cdot \mathfrak{x}[\mathsf{t}(\mathsf{e})]^{-1}$. Commutation with the surface operators is simply obtained by absorbing the gauge transformation of $\mathfrak{g}$ into a redefinition of $\mathfrak{n}$, which is summed over.

To summarise, we have thus far obtained symmetry surface operators associated with each simple of object of $2\mathsf{Rep}(\mathcal{S}_3)$. Let us now compute the corresponding fusion rules. We now by construction that these must correspond to the monoidal structure of $2\mathsf{Rep}(\mathcal{S}_3)$. Briefly, fusion rules follow from the fact that acting consecutively with two surface operators amounts to taking a Cartesian product of the local configuration space assigned to each vertex. This Cartesian product can be subsequently decomposed into disjoint unions of isomorphism classes of $\mathcal{S}_3$-sets according to the *Burnside ring* multiplication rule [73]. Concretely, let $\mathcal{N}(B, 1)$ and $\mathcal{N}(B', 1)$ be two indecomposable $\mathsf{Vec}_{\mathcal{S}_3}$-module categories, the fusion of the corresponding

topological surfaces read

$$
\begin{aligned}
&\left(\mathcal{U}^{G/B} \odot \mathcal{U}^{G/B'}\right)[\Sigma_\triangle] \\
&= \left|\frac{B \times B'}{G \times G}\right|^{\chi(\Sigma_\triangle)} \sum_{\mathfrak{g},\mathfrak{g}',n,n'} \prod_{(v_1 v_2) \subset \Sigma_\triangle} \delta_{\mathbb{C}_{\mathfrak{g}[v_1 v_2]} \triangleright n[v_2], n[v_1]} \, \delta_{\mathbb{C}_{\mathfrak{g}'[v_1 v_2]} \triangleright n'[v_2], n'[v_1]} |\mathfrak{g}\rangle\langle\mathfrak{g}|\mathfrak{g}'\rangle\langle\mathfrak{g}'| \\
&= \left|\frac{B \times B'}{G \times G}\right|^{\chi(\Sigma_\triangle)} \sum_{\mathfrak{g},n,n'} \prod_{(v_1 v_2) \subset \Sigma_\triangle} \delta_{\mathbb{C}_{\mathfrak{g}[v_1 v_2]} \triangleright (n,n')[v_2], (n,n')[v_1]} |\mathfrak{g}\rangle\langle\mathfrak{g}|,
\end{aligned} \tag{203}
$$

which is a topological surface with virtual degrees of freedom valued in the Cartesian product $\mathcal{S}_3/B \times \mathcal{S}_3/B'$ that is acted upon diagonally by $\mathcal{S}_3$. Decomposing this Cartesian product into a disjoint union of $\mathcal{S}_3$-sets then yields the decomposition of the topological surface into simple ones. For instance, when taking the fusion of topological surfaces $\mathcal{U}^{\mathcal{S}_3/\mathbb{Z}_2} \odot \mathcal{U}^{\mathcal{S}_3/\mathbb{Z}_2}$, one has

$$
\begin{aligned}
\mathcal{S}_3/\mathbb{Z}_2 \times \mathcal{S}_3/\mathbb{Z}_2 &\simeq \left\{\{1,r\},\{s,sr\},\{s^2,s^2 r\}\right\} \times \left\{\{1,r\},\{s,sr\},\{s^2,s^2 r\}\right\} \\
&\simeq \left\{\left(\{1,r\},\{1,r\}\right), \left(\{s,sr\},\{s,sr\}\right), \left(\{s^2,s^2 r\},\{s^2,s^2 r\}\right)\right\} \\
&\quad \sqcup \left\{\left(\{1,r\},\{s,sr\}\right), \left(\{1,r\},\{s^2,s^2 r\}\right), \left(\{s,sr\},\{1,r\}\right), \right. \\
&\qquad \left. \left(\{s,sr\},\{s^2,s^2 r\}\right), \left(\{s^2,s^2 r\},\{1,r\}\right), \left(\{s^2,s^2 r\},\{s,sr\}\right)\right\},
\end{aligned} \tag{204}
$$

which decomposes into a three dimensional (diagonal) orbit and a six-dimensional off-diagonal orbit under $\mathcal{S}_3$. Then it follows from (203) that

$$
\left(\mathcal{U}^{\mathcal{S}_3/\mathbb{Z}_2} \odot \mathcal{U}^{\mathcal{S}_3/\mathbb{Z}_2}\right)[\Sigma_\triangle] = \frac{1}{3^{\chi(\Sigma_\triangle)}} \cdot \mathcal{U}^{\mathcal{S}_3/\mathbb{Z}_2}[\Sigma_\triangle] \boxplus \left(\frac{2}{3}\right)^{\chi(\Sigma_\triangle)} \cdot \mathcal{U}^{\mathcal{S}_3}[\Sigma_\triangle]. \tag{205}
$$

Similarly, the fusion rules for all the remaining operators on a general (path-connected) $\Sigma$ read

$$
\begin{aligned}
&\left(\mathcal{U}^{\mathcal{S}_3} \odot \mathcal{U}^{\mathcal{S}_3}\right)[\Sigma_\triangle] = 6^{1-\chi(\Sigma_\triangle)} \cdot \mathcal{U}^{\mathcal{S}_3}[\Sigma_\triangle], \quad \left(\mathcal{U}^{\mathcal{S}_3/\mathbb{Z}_2} \odot \mathcal{U}^{\mathcal{S}_3}\right)[\Sigma_\triangle] = 3^{1-\chi(\Sigma_\triangle)} \cdot \mathcal{U}^{\mathcal{S}_3}[\Sigma_\triangle], \\
&\left(\mathcal{U}^{\mathbb{Z}_2} \odot \mathcal{U}^{\mathbb{Z}_2}\right)[\Sigma_\triangle] = 2^{1-\chi(\Sigma_\triangle)} \cdot \mathcal{U}^{\mathbb{Z}_2}[\Sigma_\triangle], \quad \left(\mathcal{U}^{\mathcal{S}_3/\mathbb{Z}_2} \odot \mathcal{U}^{\mathbb{Z}_2}\right)[\Sigma_\triangle] = \mathcal{U}^{\mathcal{S}_3}[\Sigma_\triangle], \\
&\left(\mathcal{U}^{\mathbb{Z}_2} \odot \mathcal{U}^{\mathcal{S}_3}\right)[\Sigma_\triangle] = 2^{1-\chi(\Sigma_\triangle)} \cdot \mathcal{U}^{\mathcal{S}_3}[\Sigma_\triangle].
\end{aligned} \tag{206}
$$

Note that $p^{1-\chi(\Sigma_\triangle)}$ is the partition function for the $\mathbb{Z}_p$ gauge theory on a general path connected manifold $\Sigma$ with $\Sigma_\triangle$. Choosing $\Sigma$ to be a two-torus, we recover the monoidal structure of $2\mathsf{Rep}(\mathcal{S}_3)$ [73].

Let us finally describe the topological lines living on the various surface operators constructed above. We established in sec. 3.7 that give a surface operator associated with a $\mathsf{Vec}_{\mathcal{S}_3}$-module category $\mathcal{N}(B,1)$, we can insert topological lines that form the Morita dual fusion 1-category $(\mathsf{Vec}_{\mathcal{S}_3})^\star_{\mathcal{N}(B,1)} = \mathsf{Fun}_{\mathsf{Vec}_{\mathcal{S}_3}}(\mathcal{N}(B,1),\mathcal{N}(B,1))$. Concretely, we have

$$
\begin{aligned}
(\mathsf{Vec}_{\mathcal{S}_3})^\star_{\mathsf{Vec}} &\cong \mathsf{Rep}(\mathcal{S}_3), \quad (\mathsf{Vec}_{\mathcal{S}_3})^\star_{\mathsf{Vec}_{\mathbb{Z}_2}} \cong \mathsf{Vec}_{\mathcal{S}_3}, \\
(\mathsf{Vec}_{\mathcal{S}_3})^\star_{\mathsf{Vec}_{\mathcal{S}_3/\mathbb{Z}_2}} &\cong \mathsf{Rep}(\mathcal{S}_3), \quad (\mathsf{Vec}_{\mathcal{S}_3})^\star_{\mathsf{Vec}_{\mathcal{S}_3}} \cong \mathsf{Vec}_{\mathcal{S}_3}.
\end{aligned} \tag{207}
$$

Generally speaking, the topological line associated with a simple object of such a Morita dual fusion 1-category can be realised on the lattice in terms of matrix product operators whose building blocks evaluate to the matrix elements of the module structure of the corresponding $\mathsf{Vec}_{\mathcal{S}_3}$-module functors [44, 89]. In particular, we have $\mathsf{Fun}_{\mathsf{Vec}_{\mathcal{S}_3}}(\mathsf{Vec},\mathsf{Vec}) \cong \mathsf{Rep}(\mathcal{S}_3)$, which indicates topological lines of the identity surface operator are labelled by irreducible representations of $\mathcal{S}_3$ and amount to ordinary Wilson lines. Recall that $\mathsf{Rep}(\mathcal{S}_3)$ has three simple objects, namely the trivial $\underline{0}$, the sign $\underline{1}$ and the standard representation $\underline{2}$. We provided in eq. (125) the corresponding lattice operators. These implement a non-invertible 1-form

symmetry of the model. Indeed, commutation with edge terms $\mathbb{\Pi}_e^{\mathbb{1}_{S_3}}$ follows from the fact that both operators act diagonally on basis states $|\mathfrak{g}\rangle$, while commutation with vertex terms $\mathbb{A}_v$ amounts to the gauge invariance of Wilson lines. Finally, it follows straightforwardly from (125) that composition of lines amounts to the monoidal structure of $\mathsf{Rep}(\mathcal{S}_3)$ with $\underline{0}$ the unit and $\underline{1} \otimes \underline{1} \simeq \underline{0}$, $\underline{1} \otimes \underline{2} \simeq \underline{2}$ and $\underline{2} \otimes \underline{2} \simeq \underline{0} \oplus \underline{1} \oplus \underline{2}$.

Finally, we mentioned in sec. 3.5 that we could recover the topological surfaces considered above as condensation defects obtained by condensing suitable algebras of topological lines in $\mathsf{Rep}(\mathcal{S}_3)$. Specifically, topological surfaces in $2\mathsf{Rep}(\mathcal{S}_3)$ can be identified with (separable) algebra objects in $\mathsf{Rep}(\mathcal{S}_3)$. We further commented in sec. 3.5 that these algebra objects are given by $\mathcal{S}_3$-algebras, i.e., associative unital algebras equipped with an $\mathcal{S}_3$-action. Since none of the subgroups of $\mathcal{S}_3$ has a non-trivial second cohomology group, these admit a simple definition: Given a subgroup $B \subseteq \mathcal{S}_3$, the corresponding $\mathcal{S}_3$-algebra is provided by the permutation representation $\mathbb{C}[G/B]$ with pointwise multiplication. Besides, we have $\mathbb{C}[\mathcal{S}_3/\mathcal{S}_3] \simeq \underline{0}$, $\mathbb{C}[\mathcal{S}_3/\mathbb{Z}_3] \simeq \underline{0} \oplus \underline{1}$, $\mathbb{C}[\mathcal{S}_3/\mathbb{Z}_2] \simeq \underline{0} \oplus \underline{2}$ and $\mathbb{C}[\mathcal{S}_3] \simeq \underline{0} \oplus \underline{1} \oplus \underline{2}$. Concretely, this means for instance that we can reconstruct the topological surface $\mathcal{U}^{\mathcal{S}_3/\mathbb{Z}_2}$ by inserting a network of topological lines labelled by $\underline{0} \oplus \underline{2} \in \mathsf{Rep}(\mathcal{S}_3)$.

## 5.4 $2\mathsf{Vec}_{\mathbb{G}}$ symmetry

We now turn to the symmetry structure of Hamiltonian (192) acting on a Hilbert space spanned by states $|\mathfrak{l}, \mathfrak{m}\rangle \in \bigotimes_e \mathbb{C}[\mathbb{Z}_3] \bigotimes_v \mathbb{C}[\mathbb{Z}_2]$, where $\mathfrak{l} \in Z^1(\Sigma_\triangle, \mathbb{Z}_3)$ and $\mathfrak{m} \in C^0(\Sigma_\triangle, \mathbb{Z}_2)$. Following the general discussions in sec. 3.2 and 3.8, we know that Hamiltonian (189) must have a symmetry structure embodying the fusion 2-category $2\mathsf{Vec}_{\mathbb{G}}$ of 2-vector spaces graded by the 2-group $\mathbb{G}$ with homotopy groups $\mathbb{Z}_2$ and $\mathbb{Z}_3$ in degree one and two, respectively, as defined in sec. 3.6. In particular, Hamiltonian (192) must commute with topological surfaces labelled by $\mathbb{Z}_2$-graded vector spaces of the form $\mathsf{V}_q$, where $q \in \mathbb{Z}_2$ and $\mathsf{V}_q$ has the structure of a $\mathsf{Vec}_{\mathbb{Z}_3}$-module category. We thus count four topological surfaces identified with $\mathsf{Vec}_0$, $\mathsf{Vec}_1$, $(\mathsf{Vec}_{\mathbb{Z}_3})_0$ and $(\mathsf{Vec}_{\mathbb{Z}_3})_1$. Firstly, as always, there is the identity operator

$$\mathcal{U}^{\mathrm{triv.}} = \prod_e \mathrm{id}_e, \tag{208}$$

which is here identified with $\mathsf{Vec}_0$. Next, there is a non-invertible surface operator defined as

$$\mathcal{U}^{\mathbb{Z}_3}[\Sigma_\triangle] = \frac{1}{3^{\#(\Sigma_\triangle)}} \sum_{\mathfrak{l},\mathfrak{m},\mathfrak{n},\mathfrak{b}} \exp\left(\frac{2\pi i}{3} \int_{\Sigma_\triangle} \mathfrak{b} \smile (d\mathfrak{n} - \mathfrak{l})\right) |\mathfrak{l}, \mathfrak{m}\rangle\langle \mathfrak{l}, \mathfrak{m}|, \tag{209}$$

where $\mathfrak{b} \in Z^1(\Sigma_\triangle, \mathbb{Z}_3)$ and $\mathfrak{n} \in C^0(\Sigma_\triangle, \mathbb{Z}_2)$. This is the surface operator identified with $(\mathsf{Vec}_{\mathbb{Z}_3})_0$. Summing over $\mathfrak{n}$, one obtains the presentation of this operator as a condensation defect of $\mathsf{Rep}(\mathbb{Z}_3)$ lines. The third operator, which is identified with $\mathsf{Vec}_1$, implements the 0-form $\mathbb{Z}_2$ symmetry that is left over from the initial $\mathcal{S}_3$ 0-form symmetry after gauging of the $\mathbb{Z}_3$ sub-symmetry. This $\mathbb{Z}_2$ operator is somewhat unconventional owing to the fact that $\mathbb{Z}_2$ acted non-trivially on $\mathbb{Z}_3$ via the outer automorphism $\phi$ in $\mathcal{S}_3$. Concretely, the 0-form $\mathbb{Z}_2$ symmetry operator is

$$\mathcal{O}^1 = \prod_v \sigma_v^x \prod_e \Gamma_e. \tag{210}$$

Commutation with Hamiltonian (192), and more specifically with the vertex terms, then follows from eq. (187). This operator was obtained by applying the general recipe of sec. 3.5: Recall that $2\mathsf{Vec}_{\mathbb{G}}$ arises as the Morita dual $(2\mathsf{Vec}_{\mathcal{S}_3})^\star_{2\mathsf{Vec}_{\mathbb{Z}_2}}$ of $2\mathsf{Vec}_{\mathcal{S}_3}$ with respect to $2\mathsf{Vec}_{\mathbb{Z}_2}$. In this context, the operator $\mathcal{O}^1$ is associated with the module 2-endofunctor $- \odot \mathsf{Vec}_1$. As such, it acts on degrees of freedom at vertices valued in the set of isomorphism classes of simple objects of $2\mathsf{Vec}_{\mathbb{Z}_2}$ by mulitplication by the non-trivial element in $\mathbb{Z}_2$ (hence $\sigma_v^x$), whereas

the action on edge degrees of freedom simply follows from the identification of the effective degrees of freedom according to the formula $\mathfrak{a}_{\mathfrak{g},\mathfrak{m}}[v_1 v_2] = \phi_{\mathfrak{m}[v_1]^{-1}}\big(\varpi_L(\mathfrak{g}[v_1 v_2])\big)$. The fourth and final operator denoted by $\mathcal{U}^{\mathbb{Z}_3,1}[\Sigma_\triangle]$, which is identified with the simple object $(\mathsf{Vec}_{\mathbb{Z}_3})_1$, can be simply defined as the fusion $(\mathcal{O}^1 \odot \mathcal{U}^{\mathbb{Z}_2})[\Sigma_\triangle]$. The fusion rules of these various topological surfaces can be computed using the explicit formulas above following methods employed for the other examples. We write below the non-trivial fusion rules:

$$
\begin{aligned}
\big(\mathcal{U}^{\mathbb{Z}_3} \odot \mathcal{U}^{\mathbb{Z}_3}\big)[\Sigma_\triangle] &= 3^{1-\chi(\Sigma_\triangle)} \cdot \mathcal{U}^{\mathbb{Z}_3}[\Sigma_\triangle], &\quad \big(\mathcal{U}^{\mathbb{Z}_3} \odot \mathcal{O}^1\big)[\Sigma_\triangle] &= \mathcal{U}^{\mathbb{Z}_3,r}[\Sigma_\triangle], \\
\big(\mathcal{U}^{\mathbb{Z}_3} \odot \mathcal{U}^{\mathbb{Z}_3,r}\big)[\Sigma_\triangle] &= 3^{1-\chi(\Sigma_\triangle)} \cdot \mathcal{U}^{\mathbb{Z}_3}[\Sigma_\triangle], &\quad \big(\mathcal{O}^1 \odot \mathcal{U}^{\mathbb{Z}_3,r}\big)[\Sigma_\triangle] &= \mathcal{U}^{\mathbb{Z}_3}[\Sigma_\triangle], \\
\big(\mathcal{U}^{\mathbb{Z}_3,r} \odot \mathcal{U}^{\mathbb{Z}_3,r}\big)[\Sigma_\triangle] &= 3^{1-\chi(\Sigma_\triangle)} \cdot \mathcal{U}^{\mathbb{Z}_3}[\Sigma_\triangle], &\quad \mathcal{O}^r \odot \mathcal{O}^1 &= \mathcal{U}^{\text{triv.}},
\end{aligned}
\tag{211}
$$

which matches the monoidal structure of $2\mathsf{Vec}_{\mathbb{G}}$ as defined in sec. 3.6.

Let us now analyse the topological lines living on the four topological surfaces described above. We know that these are labelled by simple objects in the endo-categories of $2\mathsf{Vec}_{\mathbb{G}}$, and we explained in sec. 3.6 that these amount to $\mathbb{Z}_2$-grading preserving $\mathsf{Vec}_{\mathbb{Z}_3}$-module functors. In particular, this means that the operator $\mathcal{O}^1$ must act on the whole space and cannot have support on an (open) sub-region of $\Sigma_\triangle$, while the topological lines on $\mathcal{U}^{\text{triv.}}$ are labelled by simple objects $(\mathsf{Vec}_{\mathbb{Z}_3})^\star_{\mathsf{Vec}} \cong \mathsf{Rep}(\mathbb{Z}_3)$. The characterisation in terms of module functors also indicates that topological lines living on $\mathcal{U}^{\mathbb{Z}_3}$ and $\mathcal{U}^{\mathbb{Z}_3,1}$ are labelled by simple objects in $(\mathsf{Vec}_{\mathbb{Z}_3})^\star_{\mathsf{Vec}_{\mathbb{Z}_3}} \cong \mathsf{Vec}_{\mathbb{Z}_3}$ and can be constructed mimicking previous constructions.

Let us focus on the topological (Wilson) lines of $\mathcal{U}^{\text{triv.}}$. Given a 1-cycle $\ell$ on $\Sigma_\triangle$ and an irreducible representation $\rho \in \mathsf{Rep}(\mathbb{Z}_3)$, we may define such a line operator as

$$
\sum_{\mathfrak{l},\mathfrak{m}} \Big(\prod_{e \subset \ell} \rho(\mathfrak{l}[e])\Big) |\mathfrak{l},\mathfrak{m}\rangle\langle\mathfrak{l},\mathfrak{m}| \,.
\tag{212}
$$

For instance, given the non-trivial irreducible representation such that $\rho(s) = \omega$, this operator can be equivalently defined as $\prod_{e \subset \ell} \Sigma^z_e$. It is then straightforward to confirm that these commute with (192). Furthermore, composition of these lines is provided by the monoidal structure of $\mathsf{Rep}(\mathbb{Z}_3)$. Interestingly, the 0-form operator $\mathcal{O}^1$ acts non-trivially on these topological lines via the action of $\mathbb{Z}_2$ on $\mathbb{Z}_3^\vee$, mapping a topological line labelled by $\rho$ to one labelled by the dual representation $\rho^\star$. Indeed, due to the presence of the $\Gamma$ matrices in eq. (210), we have for instance

$$
\mathcal{O}^1\Big(\prod_{e \subset \ell} \Sigma^z_e\Big) = \Big(\prod_{e \subset \ell} \Sigma^{z\,\dagger}_e\Big)\mathcal{O}^1 \,,
\tag{213}
$$

and vice versa. We explained this feature below eq. (96) in terms of the monoidal structure of $2\mathsf{Vec}_{\mathbb{G}}$. Similarly, the operator $\mathcal{O}^1$ acts non-trivially on the topological lines living on the topological surfaces $\mathcal{U}^{\mathbb{Z}_3}$ and $\mathcal{U}^{\mathbb{Z}_3,1}$, which can for instance be traced back to the fact that these surfaces results from condensing topological lines in $\mathsf{Rep}(\mathbb{Z}_3)$. This concludes our analysis of the symmetry structure of Hamiltonian (192) as encoded into $2\mathsf{Vec}_{\mathbb{G}}$.

## 5.5 $2\mathsf{Rep}(\mathbb{G})$-symmetry

We finally describe the symmetry structure of Hamiltonian (195), obtained by gauging the non-normal $\mathbb{Z}_2$ subgroup of $\mathcal{S}_3$ in the transverse field $\mathcal{S}_3$-Ising model (182). This model acts on a Hilbert space spanned by states $|\mathfrak{q},\mathfrak{m}\rangle \in \bigotimes_e \mathbb{C}[\mathbb{Z}_2] \bigotimes_v \mathbb{C}[\mathbb{Z}_3]$, where $\mathfrak{q} \in Z^1(\Sigma_\triangle, \mathbb{Z}_2)$ and $\mathfrak{m} \in C^0(\Sigma_\triangle, \mathbb{Z}_3)$. Following the general discussions in sec. 3.2 and 3.8, we know that Hamiltonian (189) must have a symmetry structure embodying the fusion 2-category $2\mathsf{Rep}(\mathbb{G})$ of 2-representations of the same 2-group $\mathbb{G}$ considered above. Invoking the results of sec. 3.7, Hamiltonian (195) must commute in particular with topological surfaces labelled by tuples

$(\mathsf{V}, \{l(N)\}_{N \in \mathsf{V}})$ consisting of an indecomposable $\mathsf{Vec}_{\mathbb{Z}_2}$-module category $\mathsf{V}$ and a collection $\{l(N)\}_{N \in \mathsf{V}}$ of group elements in $\mathbb{Z}_3$ for each simple object in $\mathsf{V}$ such that $l(\mathbb{C}_q \triangleright N) = \phi_{q^{-1}}(l(N))$ for every $q \in \mathbb{Z}_2$. Concretely, we count three simple topological surfaces identified with tuples $(\mathsf{Vec}, \{0\})$, $(\mathsf{Vec}_{\mathbb{Z}_2}, (0,0))$ and $(\mathsf{Vec}_{\mathbb{Z}_2}, (1,-1))$.[25] Note that by exchanging the roles of 1 and 2 in $\mathbb{Z}_3$, we find a simple object $(\mathsf{Vec}_{\mathbb{Z}_2}, (-1,1))$ that is equivalent to $(\mathsf{Vec}_{\mathbb{Z}_2}, (1,-1))$.

As usual, we begin with the identity operator

$$\mathcal{U}^{\mathrm{triv.}} = \prod_{\mathsf{e}} \mathrm{id}_{\mathsf{e}}, \tag{214}$$

which is now identified with $(\mathsf{Vec}, \{0\})$. Next, there is a non-invertible surface operator defined as

$$\mathcal{U}^{\mathbb{Z}_2,0}[\Sigma_\triangle] = \frac{1}{2^{\#(\Sigma_\triangle)}} \sum_{\mathfrak{q},\mathfrak{m},\mathfrak{b},\mathfrak{n}} \left( i\pi \int_{\Sigma_\triangle} \mathfrak{b} \smile (d\mathfrak{n} + \mathfrak{q}) \right) |\mathfrak{q},\mathfrak{m}\rangle\langle\mathfrak{q},\mathfrak{m}|, \tag{215}$$

where $\mathfrak{n} \in C^0(\Sigma_\triangle, \mathbb{Z}_2)$ and $\mathfrak{b} \in C^1(\Sigma_\triangle, \mathbb{Z}_2)$. This is the surface operator identified with $(\mathsf{Vec}_{\mathbb{Z}_2}, (0,0))$, which is an ordinary condensation defect as we described for the transverse field $\mathbb{Z}_2$-Ising model in sec. 2. The third and fourth operators, which are identified with $(\mathsf{Vec}_{\mathbb{Z}_2}, (1,-1))$ and $(\mathsf{Vec}_{\mathbb{Z}_2}, (-1,1))$, respectively, implement the 0-form $\mathbb{Z}_3$ symmetry that is left over from the initial $\mathcal{S}_3$ 0-form symmetry after gauging the $\mathbb{Z}_2$ sub-symmetry. Conventionally, a $\mathbb{Z}_3$ 0-form symmetry generator would take the form $\prod_\mathsf{v} \Sigma_\mathsf{v}^x$, but this operator clearly does not commute vertex terms depicted eq. (197) due to the presence of the $\Gamma$ matrix. Instead, one may define the following topological surface operators

$$\mathcal{U}^{\mathbb{Z}_2,\pm 1}[\Sigma_\triangle] = \frac{1}{2^{\#(\Sigma_\triangle)}} \sum_{\mathfrak{q},\mathfrak{m},\mathfrak{b},\mathfrak{n}} \exp\left( i\pi \int_{\Sigma_\triangle} \mathfrak{b} \smile (d\mathfrak{n} + \mathfrak{q}) \right) |\mathfrak{q},\mathfrak{m} \pm \phi_\mathfrak{n}(1)\rangle\langle\mathfrak{q},\mathfrak{m}|, \tag{216}$$

where $\phi_\mathfrak{n}(1)[\mathsf{v}] := \phi_{\mathfrak{n}[\mathsf{v}]}(1)$ for all $\mathsf{v} \subset \Sigma_\triangle$, which is identified with $(\mathsf{Vec}_{\mathbb{Z}_2}, (\pm 1, \mp 1))$. In contrast, the conventional 0-form $\mathbb{Z}_3$ operators would be given by $\mathcal{O}^{\pm 1} = \sum_{\mathfrak{q},\mathfrak{m}} |\mathfrak{q},\mathfrak{m} \pm 1\rangle\langle\mathfrak{q},\mathfrak{m}|$. It follows that $\mathcal{U}^{\mathbb{Z}_2,\pm 1}$ locally acts like $\mathcal{O}^{\pm 1}$ or $\mathcal{O}^{\mp 1}$ depending on the configuration $\mathfrak{n}$ of virtual degrees of freedom. More precisely, $\mathcal{U}^{\mathbb{Z}_2,1}$ is a $\mathbb{Z}_2$ condensation defect that additionally acts as $\Sigma_\mathsf{v}^x$ at a vertex $\mathsf{v} \subset \Sigma_\triangle$ if $\mathfrak{n}[\mathsf{v}] = 0$ and $\Sigma_\mathsf{v}^{x\dagger}$ if $\mathfrak{n}[\mathsf{v}] = 1$. Commutation of this surface operator with Hamiltonian (195) can be demonstrated as follows: The only non-trivial commutation to check is that with vertex terms $^\phi\widetilde{\mathbb{A}}_\mathsf{v}^r$ as depicted in eq. (197). This operator acts on the computational basis as

$$^\phi\widetilde{\mathbb{A}}_\mathsf{v}^r : |\mathfrak{q},\mathfrak{m}\rangle \mapsto |\mathfrak{q} + d\mathfrak{x}_\mathsf{v}, \phi_{\mathfrak{x}_\mathsf{v}}(\mathfrak{m})\rangle, \tag{217}$$

where $\mathfrak{x}_\mathsf{v} \in C^0(\Sigma_\triangle, \mathbb{Z}_2)$ is trivial everywhere except at the vertex $\mathsf{v}$. Let us now separately evaluate $^\phi\widetilde{\mathbb{A}}_\mathsf{v}^r \odot \mathcal{U}^{\mathbb{Z}_2,\pm 1}[\Sigma_\triangle]$ and $\mathcal{U}^{\mathbb{Z}_2,\pm 1}[\Sigma_\triangle] \odot {}^\phi\widetilde{\mathbb{A}}_\mathsf{v}^r$. On the one hand, we have

$$\mathcal{U}^{\mathbb{Z}_2,\pm 1}[\Sigma_\triangle] \odot {}^\phi\widetilde{\mathbb{A}}_\mathsf{v}^r = \frac{1}{2^{\#(\Sigma_\triangle)}} \sum_{\substack{\mathfrak{q},\mathfrak{m},\mathfrak{b},\mathfrak{n} \\ \mathfrak{q}',\mathfrak{m}'}} (-1)^{\int_{\Sigma_\triangle} \mathfrak{b} \smile (d\mathfrak{n}+\mathfrak{q})} |\mathfrak{q},\mathfrak{m} \pm \phi_\mathfrak{n}(1)\rangle\langle\mathfrak{q},\mathfrak{m}|\mathfrak{q}' + d\mathfrak{x}_\mathsf{v}, \phi_{\mathfrak{x}_\mathsf{v}}(\mathfrak{m}')\rangle\langle\mathfrak{q}',\mathfrak{m}'|$$

$$= \frac{1}{2^{\#(\Sigma_\triangle)}} \sum_{\mathfrak{q},\mathfrak{m},\mathfrak{b},\mathfrak{n}} (-1)^{\int_{\Sigma_\triangle} \mathfrak{b} \smile (d\mathfrak{n}+\mathfrak{q}+d\mathfrak{x}_\mathsf{v})} |\mathfrak{q} + d\mathfrak{x}_\mathsf{v}, \phi_{\mathfrak{x}_\mathsf{v}}(\mathfrak{m}) \pm \phi_\mathfrak{n}(1)\rangle\langle\mathfrak{q},\mathfrak{m}|$$

$$= \frac{1}{2^{\#(\Sigma_\triangle)}} \sum_{\mathfrak{q},\mathfrak{m},\mathfrak{b},\mathfrak{n}} (-1)^{\int_{\Sigma_\triangle} \mathfrak{b} \smile (d\mathfrak{n}+\mathfrak{q})} |\mathfrak{q} + d\mathfrak{x}_\mathsf{v}, \phi_{\mathfrak{x}_\mathsf{v}}(\mathfrak{m}) \pm \phi_{\mathfrak{n}-\mathfrak{x}_\mathsf{v}}(1)\rangle\langle\mathfrak{q},\mathfrak{m}|, \tag{218}$$

---

[25]Recall that we write group elements in $\mathbb{Z}_3$ as $\{0,1,2\}$ so that the mutliplication is given by the addition modulo 3.

where in the last line we performed the change of variable $\mathfrak{n} \mapsto \mathfrak{n} - \mathfrak{x}_v$. On the other hand, we have

$$
\begin{aligned}
{}^{\phi}\widetilde{\mathbb{A}}_v^r \odot \mathcal{U}^{\mathbb{Z}_2,\pm 1}[\Sigma_\triangle] &= \frac{1}{2^{\#(\Sigma_\triangle)}} \sum_{\substack{q,m,b,n \\ q',m'}} (-1)^{\int_{\Sigma_\triangle} b \smile (dn+q)} |q' + d\mathfrak{x}_v, \phi_{\mathfrak{x}_v}(m')\rangle \langle q', m'|q, m \pm \phi_n(1)\rangle \langle q, m| \\
&= \frac{1}{2^{\#(\Sigma_\triangle)}} \sum_{q,m,b,n} (-1)^{\int_{\Sigma_\triangle} b \smile (dn+q)} |q + d\mathfrak{x}_v, \phi_{\mathfrak{x}_v}(m) \pm \phi_{\mathfrak{x}_v}\phi_n(1)\rangle \langle q, m|. \quad (219)
\end{aligned}
$$

Since $\phi_{\mathfrak{n}-\mathfrak{x}_v}(1) = \phi_{\mathfrak{x}_v}\phi_\mathfrak{n}(1)$, (218) equals (219), establishing that $\mathcal{U}^{\mathbb{Z}_2,\pm 1}$ is indeed a symmetry operator of the model (195).

Guided by the presentation of $2\mathrm{Rep}(\mathbb{G})$ provided in sec. 3.6, topological lines living on the surfaces described above can be constructed mimicking previous examples. Let us rather focus on the fusion rules of these surface operators. The surface operator $\mathcal{U}^{\mathbb{Z}_2,0}$ being identical to that encountered in the study of the transverse-field $\mathbb{Z}_2$-Ising model, we already know that it satisfies fusion rules (19). However, here we present an alternative derivation, which is more suitable to compute fusion rules of the remaining operators:

$$
\begin{aligned}
\left(\mathcal{U}^{\mathbb{Z}_2,0} \odot \mathcal{U}^{\mathbb{Z}_2,0}\right)[\Sigma_\triangle] &= \frac{1}{2^{2\#(\Sigma_\triangle)}} \sum_{\substack{q_{1,2},m_{1,2} \\ n_{1,2},b_{1,2}}} (-1)^{\int_{\Sigma_\triangle} \delta^{I,J} b_I \smile (dn_J + q_J)} |q_1, m_1\rangle \langle q_1, m_1|q_2, m_2\rangle \langle q_2, m_2| \\
&= \frac{1}{2^{2\#(\Sigma_\triangle)}} \sum_{\substack{q,m \\ b_{1,2},n_{1,2}}} (-1)^{\int_{\Sigma_\triangle} b_1 \smile (dn_1 + q) + b_2 \smile (dn_2 + q)} |q, m\rangle \langle q, m| \\
&= \frac{1}{2^{2\chi(\Sigma_\triangle)}} \sum_{q,m,n_{1,2}} \delta_{dn_1,q} \delta_{dn_2,q} |q, m\rangle \langle q, m|.
\end{aligned}
$$

(220)

At this point, notice that the configuration space of virtual degrees of freedom at each vertex is given by the Cartesian product $\mathbb{Z}_2 \times \mathbb{Z}_2$, which can be decomposed into the disjoint union of two $\mathbb{Z}_2$-sets, namely $\mathbb{Z}_2^{(1)} \equiv \{(0,0),(1,1)\}$ and $\mathbb{Z}_2^{(2)} \equiv \{(0,1),(1,0)\}$. Therefore, we can decompose the summation over $\mathfrak{n}_{1,2}$ into

$$
\sum_{\mathfrak{n}_{1,2}} = \sum_{\mathfrak{n} \in C^0(\Sigma_\triangle, \mathbb{Z}_2^{(1)})} \boxplus \sum_{\mathfrak{n}' \in C^0(\Sigma_\triangle, \mathbb{Z}_2^{(2)})}. \quad (221)
$$

Using (221) in (220), we immediately obtain

$$
\left(\mathcal{U}^{\mathbb{Z}_2,0} \odot \mathcal{U}^{\mathbb{Z}_2,0}\right)[\Sigma_\triangle] = \frac{1}{2^{\chi(\Sigma_\triangle)}} \cdot \mathcal{U}_0^{\mathbb{Z}_2}[\Sigma_\triangle] \boxplus \frac{1}{2^{\chi(\Sigma_\triangle)}} \cdot \mathcal{U}_0^{\mathbb{Z}_2}[\Sigma_\triangle] = 2^{1-\chi(\Sigma_\triangle)} \cdot \mathcal{U}_0^{\mathbb{Z}_2}[\Sigma_\triangle]. \quad (222)
$$

Note that the prefactor $2^{1-\chi(\Sigma_\triangle)}$ is precisely the partition function $\mathcal{Z}_{2d}[\Sigma_\triangle]$ of the pure $\mathbb{Z}_2$ gauge theory for a path-connected manifold $\Sigma$. Whenever $\Sigma$ is a torus so that $\chi(\Sigma_\triangle) = 0$, the fusion rules reproduce the monoidal product of $\mathrm{Vec}_{\mathbb{Z}_2}$-module categories, namely

$$
\mathrm{Vec}_{\mathbb{Z}_2} \odot \mathrm{Vec}_{\mathbb{Z}_2} \cong \mathrm{Vec}_{\mathbb{Z}_2} \boxplus \mathrm{Vec}_{\mathbb{Z}_2}, \quad (223)
$$

where the first copy of $\mathrm{Vec}_{\mathbb{Z}_2}$ on the r.h.s. has simple objects $\mathbb{C}_0 \boxtimes \mathbb{C}_0$ and $\mathbb{C}_1 \boxtimes \mathbb{C}_1$, whereas the second copy has simple objects $\mathbb{C}_0 \boxtimes \mathbb{C}_1$ and $\mathbb{C}_1 \boxtimes \mathbb{C}_0$.

Guided by the derivation above, we can compute the fusion of two surface operators $\mathcal{U}^{\mathbb{Z}_2,l_1}$ and $\mathcal{U}^{\mathbb{Z}_2,l_2}$ with $l_1, l_2 \in \{0, \pm 1\}$:

$$\left(\mathcal{U}^{\mathbb{Z}_2,l_1} \odot \mathcal{U}^{\mathbb{Z}_2,l_2}\right)[\Sigma_\triangle] = \frac{1}{2^{2\chi(\Sigma_\triangle)}} \sum_{\mathfrak{q},\mathfrak{m},\mathfrak{n}_{1,2}} \delta_{d\mathfrak{n}_1,\mathfrak{q}} \delta_{d\mathfrak{n}_2,\mathfrak{q}} |\mathfrak{q}, \mathfrak{m} + \phi_{\mathfrak{n}_1}(l_1) + \phi_{\mathfrak{n}_2}(l_2)\rangle\langle\mathfrak{q},\mathfrak{m}|$$

$$= \frac{1}{2^{2\chi(\Sigma_\triangle)}} \sum_{\substack{\mathfrak{q},\mathfrak{m} \\ \mathfrak{n}\in C^0(\Sigma_\triangle,\mathbb{Z}_2^{(1)})}} \delta_{d\mathfrak{n},\mathfrak{q}} |\mathfrak{q}, \mathfrak{m} + \phi_{\mathfrak{n}_1}(l_1) + \phi_{\mathfrak{n}_2}(l_2)\rangle\langle\mathfrak{q},\mathfrak{m}|$$

$$\boxplus \frac{1}{2^{2\chi(\Sigma_\triangle)}} \sum_{\substack{\mathfrak{q},\mathfrak{m} \\ \mathfrak{n}'\in C^0(\Sigma_\triangle,\mathbb{Z}_2^{(2)})}} \delta_{d\mathfrak{n}',\mathfrak{q}_1} |\mathfrak{q}, \mathfrak{m} + \phi_{\mathfrak{n}_1'}(l_1) + \phi_{\mathfrak{n}_2'}(l_2)\rangle\langle\mathfrak{q},\mathfrak{m}|. \tag{224}$$

Let us describe various choices of $l_1, l_2$ in some detail: First of all, choosing $l_1 = l_2 = 0$, we recover the fusion rules (222). Similarly, choosing $l_2 = 0$, we find

$$\left(\mathcal{U}^{\mathbb{Z}_2,l} \odot \mathcal{U}^{\mathbb{Z}_2,0}\right)[\Sigma_\triangle] = \frac{1}{2^{\chi(\Sigma_\triangle)}} \cdot \left(\mathcal{U}^{\mathbb{Z}_2,l}[\Sigma_\triangle] \boxplus \mathcal{U}^{\mathbb{Z}_2,l}[\Sigma_\triangle]\right). \tag{225}$$

Let us now suppose that $l_1 = l_2 = 1$. Given $\mathfrak{n} \in C^0(\Sigma_\triangle, \mathbb{Z}_2^{(1)})$ so that $\mathfrak{n}_1 = \mathfrak{n}_2$, we find that the first operator appearing in the decomposition of the fusion product is a $\mathbb{Z}_2$ condensation defect that additionally acts as $\Sigma_\mathsf{v}^{x\dagger} = (\Sigma_\mathsf{v}^x)^2$ at a vertex $\mathsf{v} \subset \Sigma_\triangle$ if $\mathfrak{n}[\mathsf{v}] = (0,0)$ and $\Sigma_\mathsf{v}^x = (\Sigma_\mathsf{v}^{x\dagger})^2$ if $\mathfrak{n}[\mathsf{v}] = (1,1)$. Conversely, given $\mathfrak{n}' \in C^0(\Sigma_\triangle, \mathbb{Z}_2^{(2)})$ so that $\mathfrak{n}_1' \neq \mathfrak{n}_2'$, we find that the second operator appearing in the decomposition is a plain $\mathbb{Z}_2$ condensation defect since it acts as $\Sigma_\mathsf{v}^x \Sigma_\mathsf{v}^{x\dagger} = \text{id}$ at a vertex $\mathsf{v} \subset \Sigma_\triangle$ if $\mathfrak{n}'[\mathsf{v}] = (0,1)$ and $\Sigma_\mathsf{v}^{x\dagger} \Sigma_\mathsf{v}^x = \text{id}$ if $\mathfrak{n}'[\mathsf{v}] = (1,0)$. Putting everything together, we find

$$\left(\mathcal{U}^{\mathbb{Z}_2,1} \odot \mathcal{U}^{\mathbb{Z}_2,1}\right)[\Sigma_\triangle] = \frac{1}{2^{\chi(\Sigma_\triangle)}} \cdot \left(\mathcal{U}^{\mathbb{Z}_2,-1}[\Sigma_\triangle] \boxplus \mathcal{U}^{\mathbb{Z}_2,0}[\Sigma_\triangle]\right). \tag{226}$$

More generally, fusion rules of arbitrary topological surfaces read

$$\left(\mathcal{U}^{\mathbb{Z}_2,l_1} \odot \mathcal{U}^{\mathbb{Z}_2,l_2}\right)[\Sigma_\triangle] = \frac{1}{2^{\chi(\Sigma_\triangle)}} \cdot \left(\mathcal{U}^{\mathbb{Z}_2,l_1+l_2}[\Sigma_\triangle] \boxplus \mathcal{U}^{\mathbb{Z}_2,l_1-l_2}[\Sigma_\triangle]\right). \tag{227}$$

Choosing $\Sigma$ to be the two-torus, we recover the fusion rules provided by the monoidal structure of $2\text{Rep}(\mathbb{G})$. These can be immediately inferred from the treatment of eq. (223) [45].

# 6 Discussion

*We conclude with a discussion of extensions and generalisations of the results presented in this manuscript.*

## 6.1 Further examples

The focus of this manuscript was on developing a general framework for gauging invertible symmetries of two-dimensional quantum models and studying the resulting higher categorical symmetries. In order to illustrate our constructions, we considered finite group generalisations of the transverse-field Ising model. It will be very interesting to employ the present formalism in order to tackle more challenging as well as physically more relevant models. Indeed, using the framework presented in this manuscript, the entire parameter space of symmetric Hamiltonians generated by the local operators described in sec. 3.1 can be investigated.

Symmetries strongly constrain various aspects of the low-energy or infra-red phase diagrams of symmetric quantum systems. For instance, the kinds of phases realised, the spectrum

of excitations within each phase, universality classes of phase transitions and dualities acting on the parameter space of symmetric models, can all be studied from the lens of the symmetry structure. Therefore it is natural to study the phase diagrams of the quantum spin models introduced in this work from the perspective of their higher categorical symmetries.

It is also possible to extend the current framework so as to consider more general models as well as more general higher categorical symmetries. For instance, given a group $G \simeq Q \ltimes_\phi L$, our framework readily accommodates models with $2\mathsf{Vec}_G^\pi$-symmetry where $\pi$ is a (non-trivial) 4-cocycle in $H^4(G, \mathrm{U}(1))$ characterising the monoidal pentagonator of the fusion 2-category.[26] Such a monoidal pentagonator encapsulates an anomaly revealing an obstruction to gauging the whole symmetry. For certain choices of 4-cocycle $\pi$, gauging the $L$-sub-symmetry would result in a model with a $2\mathsf{Vec}_{\mathbb{G}}$-symmetry, where $\mathbb{G}$ is a 2-group with a non-trivial *Postnikov* class [114], thereby going beyond the examples considered in the current manuscript.

More generally, the framework employed in this manuscript can be extended so as to accommodate arbitrary fusion 2-categories generalising further the one-dimensional framework presented in [89]. For instance, given an input fusion 2-category and a choice of module 2-category over it, local operators can be defined of the form (52) evaluating to matrix entries of the corresponding module pentagonator. In particular, it would be interesting to consider models built from the data of fusion 2-categories obtained by idempotent completions of deloopings of braided fusion 1-categories [47,70]. The corresponding module 2-categories were discussed in ref. [53].

## 6.2 Self-dual models

A celebrated result in low-dimensional condensed matter physics is the exact localisation of the critical point of the (1+1)d transverse-field Ising model by invoking its self-duality [85]. The relevant duality in this case is the Kramers-Wannier duality, which, up to a local unitary, is obtained by gauging its $\mathbb{Z}_2$-symmetry. As we reviewed in this manuscript, this phenomenon is specific to (1+1)d since the (2+1)d transverse-field Ising model that possesses an ordinary $\mathbb{Z}_2$-symmetry is dual to a model that possesses in particular a 1-form $\mathbb{Z}_2^\vee$-symmetry. This begs the question, how to construct (non-trivially) self-dual symmetric spin systems in two spatial dimensions?

It turns out that the mathematical framework developed in this manuscript allows us to rule out many possibilities. Indeed, a necessary condition for self-duality is that the fusion 2-categories of symmetry operators associated with a model and its dual are monoidally equivalent, in addition to being Morita equivalent. This is the case of the (1+1)d Ising model, where the initial $\mathsf{Vec}_{\mathbb{Z}_2}$-symmetry is monoidally equivalent to the dual $\mathsf{Rep}(\mathbb{Z}_2)$-symmetry. In contrast, $2\mathsf{Vec}_{\mathbb{Z}_2}$ and $2\mathsf{Rep}(\mathbb{Z}_2)$ are clearly not monoidally equivalent. As a matter of fact, starting from a $G$-symmetric theory, any duality involving the gauging of a non-trivial subgroup of $G$ would result in a model whose symmetry is not monoidally equivalent to $2\mathsf{Vec}_G$, making it impossible for the model to be self-dual. Recent computations of the *Brauer-Picard group* of $2\mathsf{Rep}(G)$, which informs us about auto-equivalences of the algebraic structure encoding the super-selection sectors of a $G$-symmetric model, suggests that the only candidate dualities may be of the form $2\mathsf{Vec} \to 2\mathsf{Vec}^\lambda$, which only involve a change of module pentagonator amounting to the pasting of a (2+1)d *symmetry-protected* topological phase [54,79]. This special type of duality, and the possibility of defining self-dual models with respect to it, will be investigated elsewhere.

---

[26]Within our framework, this is simply accomplished by choosing $2\mathsf{Vec}_G^\pi$ as a module 2-category over itself when defining the local operators (52), so that the module pentagonator coincides with the monoidal pentagonator.

## 6.3 Symmetry-twisted boundary conditions

Throughout this manuscript, we have purposefully been somewhat vague regarding the role of boundary conditions. For instance, given a two-dimensional surface with the topology of a torus, our results would implicitly assume periodic boundary conditions. But our framework can be extended so as to accommodate *symmetry-twisted* boundary conditions. In the case of a $G$-symmetric model, we expect symmetry-twisted boundary conditions along the pair of non-contractible cycles of the torus to be labelled by commuting group elements in $G$. Commutativity in $G$ should be required so as to preserve translation invariance of the model up to local unitary transformations. Importantly, the original $G$-symmetry interacts with these boundary conditions in such a way that one is typically left with a smaller symmetry in the presence of non-trivial symmetry twists. Concretely, given symmetry twists $(g_1, g_2) \in G^2$ such that $g_1 g_2 = g_2 g_1$, the leftover symmetry group is given by the *stabiliser* subgroup of group elements $x \in G$ satisfying $(x g_1 x^{-1}, x g_2 x^{-1}) = (g_1, g_2)$. It follows in particular that every pair of commuting twists $(g_1, g_2)$ and $(g_1', g_2')$ for which there exists $x \in G$ such that $(g_1', g_2') = (x g_1 x^{-1}, x g_2 x^{-1})$ possess the same stabiliser subgroup, thereby defining equivalence classes of boundary conditions. Given such an equivalence class and one of its representatives, the resulting Hamiltonian would decompose into symmetry charge sectors labelled by irreducible representations of the stabiliser subgroup of the representative.

Interestingly, the same data labelling the super-selection sectors described above, i.e. symmetry-twisted boundary conditions together with twisted symmetry charge sectors, appeared before in a different context. Indeed, these correspond to the simple modules of *tube algebras*, which were first considered in ref. [43] and generalised in ref. [12,13], that classify and characterise *loop-like excitations* in (3+1)d Hamiltonian realisations of *Dijkgraaf-Witten* theory [51]. More specifically, such a simple module labels a loop-like flux, to which a point-like charge may be attached, while being threaded by an auxiliary string-like flux.

A complimentary field-theoretic approach was developed in ref. [38,39,111,112] to extract topological line and surface operators and the braiding phases of (3+1)d topological finite group gauge theories from their gapless surface theories.

A similar interplay between super-selection sectors of symmetric models and topological excitations of a higher-dimensional topological model exist in (1+1)d. Indeed, super-selection sectors of a symmetric model are known to be labelled by simple objects in the monoidal centre of the symmetry fusion 1-category [1,87,90,93,97], and the interplay between dualities and super-selection sectors was recently studied in detail for arbitrary one-dimensional quantum lattice models in ref. [90,97]. But the monoidal center of a fusion 1-category encodes the anyonic excitations of the corresponding Hamiltonian realisation of the *Turaev-Viro-Barrett-Westbury* state-sum invariant [26,96,116]. Interestingly, the notion of monoidal centre of a fusion 2-category also exists [21,36] and was computed explicitly in a few cases [84]. Perhaps surprisingly, given a topological model with input datum a certain (spherical) fusion 2-category, the excitation content encoded into its monoidal centre differs from that described by the tube algebras mentioned above [14]. The precise relation between both algebraic structures together with the corresponding physical implications were clarified in ref. [15]. Concretely, this means that in contrast to the one-dimensional scenario, super-selection sectors of a $G$-symmetric model on a torus are not labelled by simple objects in the monoidal centre of $2\mathsf{Vec}_G$. In light of the results obtained in ref. [14,15], we rather conjecture that the monoidal centre in the higher dimensional case would encode super-selection sectors of a model defined on a cylinder hosting a combination of open and closed boundary conditions.

## Acknowledgments

CD would like to thank Thibault Décoppet for numerous discussions on Morita equivalence of fusion 2-categories, as well as Laurens Lootens, Frank Verstraete and Gerardo Ortiz for collaborations on the lower-dimensional setting. AT would like to thank Lakshya Bhardwaj, Lea Bottini, Heidar Moradi, Faroogh Moosavian and Sakura Schäfer Nameki for numerous discussions on related topics.

**Funding information** This work has received funding from the Research Foundation Flanders (FWO) through postdoctoral fellowship No. 1228522N awarded to CD. AT is supported by the Swedish Research Council (VR) through grants number 2019-04736 and 2020-00214.

## A Two-dimensional $\mathbb{Z}_p$ gauge theory

In this appendix, we collect some basic facts about different presentations of the two-dimensional $\mathbb{Z}_p$ topological gauge theory, which appear in the definition of condensation defects. In particular, the case $p = 2$ appears throughout the manuscript. Given a closed oriented surface $\Sigma$ endowed with a triangulation $\Sigma_\triangle$, the partition function of the theory reads[27]

$$\mathcal{Z}_{2d}^{(p)}[\Sigma_\triangle] = \frac{1}{p^{\#(\Sigma_\triangle)}} \sum_{\mathfrak{b},\mathfrak{n}} \exp\left( \frac{2\pi i}{p} \int_{\Sigma_\triangle} \mathfrak{b} \smile d\mathfrak{n} \right), \tag{A.1}$$

where $\#(\Sigma_\triangle) := |\Sigma_\triangle^0| + |\Sigma_\triangle^2|$ and $|\Sigma_\triangle^j|$ is the number of $j$-simplices in the triangulation $\Sigma_\triangle$. In the above partition sum, $\mathfrak{b} \in C^1(\Sigma_\triangle, \mathbb{Z}_p)$ and $\mathfrak{n} \in C^0(\Sigma_\triangle, \mathbb{Z}_p)$. As in the main text, the symbols $d$ and $\smile$ denote the simplicial codifferential and the cup product, respectively.

Let us begin by listing a couple of important identities that are used in several occurences in the main text

$$\sum_{\mathfrak{n}} \exp\left( \frac{2\pi i}{p} \int_{\Sigma_\triangle} \mathfrak{n} \smile (d\mathfrak{b} - \mathfrak{q}_2) \right) = p^{|\Sigma_\triangle^2|} \delta_{d\mathfrak{b},\mathfrak{q}_2},$$

$$\sum_{\mathfrak{b}} \exp\left( \frac{2\pi i}{p} \int_{\Sigma_\triangle} \mathfrak{b} \smile (d\mathfrak{n} - \mathfrak{q}_1) \right) = p^{|\Sigma_\triangle^1|} \delta_{d\mathfrak{n},\mathfrak{q}_1}, \tag{A.2}$$

where $\mathfrak{q}_j \in C^j(\Sigma_\triangle, \mathbb{Z}_p)$. We can now evaluate the partition function by summing over $\mathfrak{n}$. First we perform an integration by parts such that the codifferential acts on $\mathfrak{b}$. Then summing over $\mathfrak{n}$ imposes a $\mathbb{Z}_p$ delta function on each 2-simplex, giving an overall factor of $p^{|\Sigma_\triangle^2|}$. Putting everything together, one obtains

$$\mathcal{Z}_{2d}^{(p)}[\Sigma_\triangle] = \frac{p^{|\Sigma_\triangle^2|}}{p^{\#(\Sigma_\triangle)}} \sum_{\mathfrak{b}} \delta_{d\mathfrak{b},0} = \frac{|Z^1(\Sigma_\triangle, \mathbb{Z}_p)|}{p^{|\Sigma_\triangle^0|}} = \frac{|H^1(\Sigma_\triangle, \mathbb{Z}_p)| \times |B^1(\Sigma_\triangle, \mathbb{Z}_p)|}{p^{|\Sigma_\triangle^0|}}$$

$$= \frac{|H^1(\Sigma_\triangle, \mathbb{Z}_p)|}{|H^0(\Sigma_\triangle, \mathbb{Z}_p)|} = p^{b_1(\Sigma) - b_0(\Sigma)}. \tag{A.3}$$

---

[27]In the main text, we denote the partition function of the two dimensional $\mathbb{Z}_2$ gauge theory $\mathcal{Z}_{2d}[\Sigma_\triangle] := \mathcal{Z}_{2d}^{(2)}[\Sigma_\triangle]$.

Notice that the sum over $\mathfrak{b}$ gives a factor of $|Z^1(\Sigma_\triangle, \mathbb{Z}_p)|$ due to the $\mathbb{Z}_p$ cocycle constraint. Moreover, we used $H^1(\Sigma_\triangle, \mathbb{Z}_p) = Z^1(\Sigma_\triangle, \mathbb{Z}_p)/B^1(\Sigma_\triangle, \mathbb{Z}_p)$, where $Z^1(\Sigma_\triangle, \mathbb{Z}_p)$ and $B^1(\Sigma_\triangle, \mathbb{Z}_p)$ are the set of 1-cocycles and 1-coboundaries, respectively. Finally we employed the expression

$$B^1(\Sigma_\triangle, \mathbb{Z}_p) \simeq C^0(\Sigma_\triangle, \mathbb{Z}_p)/Z^0(\Sigma_\triangle, \mathbb{Z}_p) \simeq C^0(\Sigma_\triangle, \mathbb{Z}_p)/H^0(\Sigma_\triangle, \mathbb{Z}_p). \tag{A.4}$$

Notice that the final expression in eq. (A.3), which is a topological invariant, can equivalently be expressed in terms of the 1st and 2nd Betti numbers $b_{1,2}(\Sigma)$ of $\Sigma$.

It is also instructive to compute (A.1) by summing over $\mathfrak{b}$ instead of summing over $\mathfrak{n}$:

$$\mathcal{Z}_{2d}^{(p)} = \frac{p^{|\Sigma_\triangle^1|}}{p^{\#(\Sigma_\triangle)}} \sum_{\mathfrak{n}} \delta_{d\mathfrak{n},0} = \frac{1}{p^{\chi(\Sigma)}} \sum_{\mathfrak{n}} \delta_{d\mathfrak{n},0} = \frac{|H^0(\Sigma_\triangle, \mathbb{Z}_p)|}{p^{\chi(\Sigma)}} = p^{b_1(\Sigma)-b_2(\Sigma)} = p^{b_1(\Sigma)-b_0(\Sigma)}. \tag{A.5}$$

We first used eq. (A.2), as well as the expression

$$\chi(\Sigma) := |\Sigma_\triangle^0| - |\Sigma_\triangle^1| + |\Sigma_\triangle^2| = b_0(\Sigma) - b_1(\Sigma) + b_2(\Sigma), \tag{A.6}$$

defining the Euler characteristic $\chi(\Sigma)$ of the surface $\Sigma$. Then, we evaluated the sum over $\mathfrak{n}$, producing a factor $|Z^0(\Sigma_\triangle, \mathbb{Z}_p)| = |H^0(\Sigma_\triangle, \mathbb{Z}_p)| = p^{b_0(\Sigma)}$. Lastly, in going to the final expression, we used that for any 2-manifold $b_2(\Sigma) = b_0(\Sigma)$. As expected, the two ways of computing the partition function give the same topological invariant.

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
