# Peer review of "Higher categorical symmetries and gauging in two-dimensional spin systems"

_SciPost Physics, doi:SciPost Phys. 16, 110 (2024)_

## Round 2 · Referee Report · Anonymous (Referee 2) · 2024-3-24

Report

I am happy with the authors' changes and recommend their paper for publication. Below I reply to one of their responses to my first report to hopefully make my original remark clear

$\underline{\text{Original remark}}$: “Since the authors study Hamiltonian lattice models, mentioning subsystem symmetries may be worthwhile in the introduction. Otherwise, according to the author's discussion in the introduction, string-net models would have no exact symmetries since the string operators commuting with the Hamiltonian are not topological outside of the ground state subspace.”

$\underline{\text{Author's reply}:}$ “We do not fully understand the referee’s comment about subsystem symmetries in this context. All the models we consider in our paper have exact generalized symmetries owing to the local (Gauss) constraints imposed kinematically on the Hilbert space. We fail to see any connection between these local constraints and sub-system symmetries. While the latter are genuine global symmetries, the former are kinematic constraints imposed at the level of the Hilbert space.”

$\underline{\text{My reply}}$: I apologize if my original remark was unclear. In the paper, the authors follow the definition that a topological operator = a symmetry operator. My remark was that in non-relativistic theories (i.e., lattice models), there can be symmetry operators that are not topological or are only partially topological (i.e., subsystem symmetry). The string-net model was an example of a lattice model whose 1-form symmetry operators are not topological (they are only in the ground state sub-Hilbert space).

Since the authors do not require a Hilbert space with a tensor product decomposition, this isn't a huge deal. But, many people I know would regard a lattice model - especially something called a spin model - to have a tensor product Hilbert space by definition. With a tensor product Hilbert space, I believe higher-form symmetry operators will never be topological operators for the entire Hilbert space. Since 99% of CMT folks working with lattice models probably define lattice models to be one with tensor product Hilbert space, my original remark was for the authors to bring this up in their introduction.

---

## Round 2 · Referee Report · Anonymous (Referee 3) · 2024-3-25

Report

Recommend publication

---

## Round 2 · Author Response

We thank both referees for their careful reading of our manuscript, encouraging comments and useful suggestions. We have implemented all requested changes by the referees as well as numerous optional ones as per the referees’ suggestions. We detail below the changes we made and address the referees’ comments:

REFEREE 1:

“One point that I got confused about: when the authors say two-dimensions, do they really mean (2+1)d in all the cases?”

Although the quantum mechanical systems we consider are always (2+1)d, their referee is correct that we sometimes refer to (2+0)d partition function when defining condensation defects. We clarified this aspect wherever there was scope for confusion.

REFEREE 2:

“Since the authors study Hamiltonian lattice models, mentioning subsystem symmetries may be worthwhile in the introduction. Otherwise, according to the author's discussion in the introduction, string-net models would have no exact symmetries since the string operators commuting with the Hamiltonian are not topological outside of the ground state subspace.”

We do not fully understand the referee’s comment about subsystem symmetries in this context. All the models we consider in our paper  have exact generalized symmetries owing to the local (Gauss) constraints imposed kinematically on the Hilbert space. We fail to see any connection between these local constraints and sub-system symmetries. While the latter are genuine global symmetries, the former are kinematic constraints imposed at the level of the Hilbert space.

“A handful of statements made throughout the paper apply only to discrete or finite symmetries but are presented as if generally true. For example, only finite non-invertible symmetries are described by fusion d-categories not all non-invertible symmetries. Another example is dual symmetries and gauging, which only applies to gauging discrete sub-symmetries.”

We agree with the referee that not all generalised symmetries fall within the scope of fusion 2-categories. Wherever there was potential scope for confusion, we clarified  that the fusion 2-category classification pertains to finite semisimple symmetries. Given that we clearly indicate in multiple places that our starting point is an invertible symmetry encoded into a finite group, we believe there should not be any confusion as to the scope of our work.

“When defining the general Hamiltonian $H \equiv \sum_h h_v$ in section 3.1, would it be possible to write the operator $h_v$ in a purely two-dimensional way instead of using the pitched cobordism Eq. (3.3)?”  

One of the motivations for our graphical calculus is the appearance of phase factors evaluating to group 3-cocycles in the case of twisted gauging, which can be conveniently read off the pinched interval cobordism by following a prescription akin to the  Dijkgraaf-Witten state-sum invariant. In the absence of cohomological twist, it is indeed more beneficial to write the operators in a purely two-dimensional manner, as we do in section 3.8.

“It would be interesting to see how the model in section 3.1 can become the Heisenberg model, as stated in the text. This may be demonstrated alongside the transverse field Ising model in that subsection: both models have the same Hilbert spaces, but different choices of coefficients $h_{v,i}$    yield the different models. Also, interestingly, one model has a finite symmetry while the other has a continuous symmetry.”

We now explain at the end of section 3.1 how to obtain for instance the spin-1/2 XY model within this formalism.  

“The starting model before gauging has an ordinary symmetry whose group elements define the Hilbert space. Do the authors expect the generalization to a model with a general fusion 2-category symmetry, which they state they can do in section 6, to be straightforward? I would suspect that the gauging procedure in sec 3.2 would need to be modified since non-invertible symmetries do not have gauge fields. Maybe a more general starting point for gauging is not minimal coupling but instead using symmetry defects. It may be interesting to comment on this in section 6 or elsewhere.”

Yes, the motivation for our formalism is precisely that it generalises fairly straightforwardly to the general case. As a matter of fact, the formalism we employ in this manuscript is a direct higher-dimensional generalisation of that presented in [arXiv:2112.09091 ], which can deal with gauging/duality of arbitrary symmetries encoded into fusion categories in the one-dimensional setting. In this more general context, dualities would simply be performed by changing module 2-categories, which encode a topological boundary condition of the underlying (3+1)d TQFT, the same way we do it in the invertible case. The difficulty with the more general case is to rewrite the operators in a more enlightening manner in terms of spin operators for instance.

“It is common with Hamiltonian lattice models to assume the Hilbert space has a tensor product decomposition. This isn't the case in this paper since the Morita dual model has a flatness condition imposed at the Hilbert space level. Of course, this flatness condition can be implemented energetically, in which case the low-energy Hilbert space would have the dual symmetry studied here. Do the authors have an idea about what the microscopic symmetry affecting the entire Hilbert space is in such a modified Morita dual model?”

The referee is correct that gauging/dualising comes with some kinematical constraints that need to be imposed for the dual model to have the expected dual symmetry encoded in the Morita dual category. We have not considered what symmetry may remain upon promoting these kinematical constraints to dynamical ones.

“When reading Sec. 3.7, I found myself frequently jumping back to Sec. 3.6 to remind me of the definition of the 2-group $\mathbb G$, particularly whether $Q$ or $L$ was the homotopy group of degree $1$ or $2$ suggest the authors change the notation and denote $Q$ and $L$ as $\pi_1$ and $\pi_2$ so the group $G$ becomes $\pi_\ltimes_\phi \pi_2$ in Sec. 3.7. It would make it much smoother to read.”

Although we sympathise with the referee’s comment, using $Q$ and $L$ for the group allows us to use different notations for group elements, namely $q$ and $l$, which we believe greatly improve readability.

“Could the authors comment on any challenges arising from considering a more general $G$ in section 3.7, such as one that is a nontrivial group extension? The authors do briefly remark in Section 6 that the semidirect product $G$is not the most general and could be anomalous. But I am curious why they didn't explore such a more general $G$ given how general other parts of the paper are and, therefore, wonder if there are some technical hurdles.”

The referee is correct that our framework is very general and can readily accommodate the case of symmetries encoded into fusion 2-categories and in particular the case of a group $G$ with a non-trivial group extension. The sole reason we do not consider this case in our manuscript is our wish to be able to write down Hamiltonians as explicitly as possible in terms of spin operators. However, including a non-trivial extension means dealing with dual models with anomalous symmetries that are difficult to write in an enlightening way in terms of spin operators beyond very simple cases.

“I only have one requested change. To me, two theories are dual if they are two different-looking theories that describe the same physical system (e.g., particle vortex dualities). The authors use the term "duality" in a much weaker way to mean two theories whose local symmetric operators can be mapped to one another. I am okay with that since it is used this way else way in the literature (e.g., Wen's work since 2018). Still, I think it would be nice for the authors to explicitly remark somewhere that when they say two models are "dual," it does not mean that their partition functions are equivalent nor that they describe the same physical system. Rather, their spectrum and correlation functions match only in the symmetric sub-Hilbert space.”

The referee is correct that we employ a somewhat weak notion of duality, where two models are defined to be dual if an isometric mapping between the corresponding Hamiltonians can be found. A comment regarding this point was added in the introduction. However, this is does not necessarily mean that it only holds in the singlet sectors of the symmetries. Indeed, by inserting appropriate symmetry-twisted boundary conditions, isometric mappings can be found between all possible compatible sectors.

---

## Editorial Decision

published